# Mitigating Data Heterogeneity Effect in Client-Reshuffling-Based Federated Learning

## Abstract

Data heterogeneity and low client participation have been the key challenges in federated learning. Client-reshuffling-based federated learning methods were recently introduced to improve the client participation efficiency. However, the client-reshuffling-based methods still suffer from the data heterogeneity issue. To fill in this gap, we propose a new algorithm, FedCDR, to mitigate the data heterogeneity challenge in client-reshuffling-based federated learning. Our algorithm achieves the state-of-the-art $O(\epsilon^{-2})$ convergence rate for finding an $\epsilon$-approximate stationary point under standard assumptions. Unlike previous works, our method achieves convergence **independent** of the degree of data heterogeneity, *i.e.* our algorithm converges fast in highly heterogeneous data environments, whereas previous methods suffer from non-convergence or slow convergence rates. Moreover, our algorithm uses inexact local solvers, which are essential for practical implementation and requirements. In our theoretical analysis, client-reshuffling-based approaches introduce a new technical challenge: non-i.i.d. sampling bias, which complicates the convergence analysis. We design a novel potential function and adopt advanced analytical techniques to address this challenge. Our experimental results demonstrate the advantages of our method over existing algorithms on both synthetic and benchmark datasets.

## 1 Introduction

Federated learning (FL) research has gained increasing attention in the machine learning community in recent years. Modern applications often require training deep neural networks with billions of parameters on large datasets (Brown et al., 2020). To accelerate such large-scale model training, federated learning algorithms have become increasingly important. Beyond computational scalability, a key motivation for FL is that data is often naturally distributed across multiple devices and may be private. Leveraging these advantages, FL has been deployed in various real-world applications (Hard et al., 2018).

FL often faces several key challenges in practical applications, such as the data heterogeneity challenge (Li et al., 2020a; Karimireddy et al., 2020) and the low client participation issue (Malinovsky et al., 2023a). The data heterogeneity challenge arises because clients' local datasets are non-IID, varying in distribution, size, and noise. This leads to biased updates, slow or unstable convergence, and poor generalization. The low client participation issue is inefficiency and unfairness in client participation that results from naive or poorly designed client sampling strategies in large-scale FL. When FL has a large number of clients, due to communication, computation, and network constraints, only a small subset of clients can participate in each communication round. If the client sampling strategies are not carefully designed, this often leads to inefficient and unfair client participation (detailed discussed in Appendix D): some clients may have a very low chance of being selected or may only be chosen a few times, while others could be repeatedly selected.

Traditional FL methods (Li et al., 2020a; Karimireddy et al., 2020; Tran-Dinh et al., 2021) adopt naive client sampling strategies and suffer from the low client participation issue. Recent works (Malinovsky et al., 2023a) propose *client reshuffling* to mitigate the low client participation issue. Unlike naive client sampling methods, client reshuffling groups $R$ communication rounds into a meta-

epoch. At the beginning of each meta-epoch, a random permutation of all clients is generated and divided into batches $(S_0, \ldots, S_{R-1})$. In the $r$-th communication round, only the clients in batch $S_r$ participate in training. This strategy avoids overlapping client selections within the same meta-epoch and ensures that each client is selected exactly once per meta-epoch, thereby improving participation efficiency and fairness. Moreover, *client reshuffling* also satisfies certain practical requirements (e.g., some devices may limit the number of times they can participate).

However, the data heterogeneity challenge under client-reshuffling-based FL settings[1] has not yet been resolved. This notable gap in the literature naturally leads to the following question:

> *Is it possible to design a federated learning algorithm that addresses the data heterogeneity challenge under the client-reshuffling-based FL setting, without relying on additional restrictive assumptions?*

In this paper, we provide a positive answer to this question with rigorous theoretical analysis.

### 1.1 OUR MAIN CONTRIBUTIONS

**New Method.** We propose a new method called **FedCDR** (**Fed**erated **C**lient Reshuffling **D**ouglas– **R**achford Method) (*Algorithm 1*). This method integrates client reshuffling with the Douglas-Rachford (DR) splitting technique in nonconvex federated learning. Moreover, our method employs an inexact evaluation of the proximal operator in the local solver, which is essential for practical implementation and requirements (discussed in Section 4 and 6.3). FedCDR effectively addresses the data heterogeneity challenge in the client-reshuffling-based FL setting.

**Convergence Analysis.** Our method achieves the best-known $O(\epsilon^{-2})$ communication complexity for finding an $\epsilon$-approximate stationary point under standard assumptions in nonconvex FL. Moreover, our analysis shows that the upper bound of the proposed method is *independent of data heterogeneity* across different clients. Specifically, the convergence rate of our method remains unaffected even in high heterogeneity settings.

**Technical Novelty.** A key challenge in analyzing the convergence rate of our method is that, unlike naive client sampling methods, the client-reshuffling-based approach does not produce independent and identically distributed (i.i.d.) samples across different inner communication rounds within the same meta-epoch. As a result, we cannot directly compute the expectation of variables associated with a single inner communication round. To address this technical challenge, we treat the entire meta-epoch as an ensemble and rely heavily on conditional expectations for calculation. Firstly, we bound several key terms, such as the error between aggregated updates and local updates. Secondly, we construct a new potential function and establish a bound on its descent across successive rounds. Finally, we compute the expectation of the sum of key terms over the meta-epoch to eliminate randomness and derive the convergence theorem.

**Experimental Validation.** The theoretical analysis of our method is supported by experimental evaluations. We conduct numerical experiments on both synthetic and benchmark datasets. The results demonstrate that our method outperforms the baselines and effectively mitigating data heterogeneity under the client-reshuffling-based FL setting. Moreover, we discuss the communication–computation trade-off in FedCDR and provide ablation studies showing that client reshuffling accelerates FL training.

## 2 RELATED WORK

**Federated Learning and Data Heterogeneity.** Federated Averaging (FedAvg) is one of the earliest methods used in FL (McMahan et al., 2017a). In FedAvg, clients perform stochastic gradient descent (SGD) updates for a number of epochs and then send the updated models to the server for aggregation. The practical performance of FedAvg has been demonstrated in many early studies

---

[1] By *client-reshuffling-based FL*, we mean federated learning algorithms that adopt client reshuffling as the client participation scheme.

(Konečný et al., 2016; McMahan et al., 2017b). However, analyzing the convergence of FedAvg was challenging in its early stages due to data heterogeneity. For the i.i.d. setting, Stich (2019) presents one of the earliest works to analyze FedAvg in convex settings. Yu et al. (2019) extended this analysis to nonconvex settings. In the non-i.i.d. setting, Khaled et al. (2019) studied the convergence properties of local gradient descent (GD) in FL. Additionally, Li et al. (2020b) analyzed the convergence of SGD in convex FL settings. Moreover, Pathak & Wainwright (2020) and Zhang et al. (2020) demonstrated that FedAvg may fail to converge even in convex settings due to data heterogeneity.

Several studies have addressed the data heterogeneity problem in FL. FedProx (Li et al., 2020a) introduces a proximal term in local updates to improve the stability of the training process, demonstrating superior performance to FedAvg in heterogeneous settings. SCAFFOLD (Karimireddy et al., 2020) is another approach that mitigates data heterogeneity by using a control variate to correct client drift in local updates. FedDC (Gao et al., 2022) introduces correction variables in local updates to decouple client drift from global optimization, thereby improving stability and fairness, and demonstrating superior performance to FedAvg in heterogeneous settings. FedDR (Tran-Dinh et al., 2021) employs a nonconvex Douglas-Rachford splitting method to address the client drift problem in FL. However, in terms of client sampling strategies, previous algorithms typically select clients using naive sampling methods. Typically, a subset of clients is sampled from the entire client set *with replacement* in each communication round. Each client $k$ is selected with probability $p_k$. This naive strategy leads to inefficient and unfair client participation and suffers from the low client participation issue. A detailed discussion of why this naive client sampling strategy suffers from the low client participation issue is provided in Appendix D.

Unlike these prior works, our method operates under client-reshuffling-based FL settings, which improve client utilization efficiency and better satisfy practical requirements. Our goal is to address the data heterogeneity challenge within this specific framework.

**Federated Learning with Client Reshuffling.** In FL, most existing methods (Li et al., 2020a; Karimireddy et al., 2020; McMahan et al., 2017a) sample clients via naive random sampling, which leads to low effective client participation and practical limitations. To address the limitations of naive sampling, researchers (Malinovsky et al., 2023a; Cho et al., 2023; Horváth et al., 2022) introduced *client reshuffling* in FL, which shuffles the full client list once per meta-epoch and visits clients sequentially so that each client participates exactly once per epoch, thereby leveraging the finite-sum structure and improving client attendance. Moreover, Wang & Ji (2022) extend client reshuffling to a unified analysis for arbitrary client participation. However, client reshuffling introduces sampling bias, since each step no longer uses an unbiased estimate of the average client gradient, which complicates convergence analysis. The analysis in (Malinovsky et al., 2023a) is restricted to convex and strongly convex objectives, while Demidovich et al. (2024) extend client-reshuffling-based analysis to the nonconvex setting under generalized smoothness assumptions.

Nevertheless, the data heterogeneity challenge under client-reshuffling-based FL settings has not yet been resolved. All the aforementioned methods are only effective under low data heterogeneity settings. In their convergence analysis, a heterogeneity-induced error term appears, which is positively correlated with the degree of data heterogeneity and does not decay as the number of iterations increases. As a result, their algorithms do not converge in highly heterogeneous environments when using a constant learning rate. To guarantee convergence, they adopt a very small learning rate, on the same order as the target accuracy $\epsilon$, which is often impractical and slows down the convergence of the algorithm.

In contrast to prior work, our method achieves a convergence rate that is *independent of data heterogeneity*. Specifically, the convergence rate depends only on the objective function value and the algorithm's hyperparameters. This key distinction demonstrates that our method successfully addresses the data heterogeneity issue within the client-reshuffling-based FL framework.

## 3 PRELIMINARIES

We denote $\mathbb{R}^n$ as the $n$-dimensional Euclidean space with inner product $\langle \cdot, \cdot \rangle$ and Euclidean norm $\| \cdot \|$. An extended-real-valued function $f : \mathbb{R}^n \to [-\infty, \infty]$ is said to be proper if $f := \{x \in \mathbb{R}^n :$

$f(x) < \infty\}$ is not empty and $f$ never equals $-\infty$. We say a proper function $f$ is closed if it is lower semicontinuous.

In this paper, we consider the following nonconvex optimization problem:

$$\min_{x \in \mathbb{R}^d} \left[ f(x) := \frac{1}{n} \sum_{i=1}^n f_i(x) \right], \tag{1}$$

where $f_i$ is differentiable. In federated learning, $n$ is the total number of clients. $x \in \mathbb{R}^d$ corresponds to the parameters of the model. $f_i : \mathbb{R}^d \to \mathbb{R}$ is empirical loss of the model $x$ defined by the local training data stored by the $i_{\text{th}}$ client.

To analyze the optimzation problem (1), we make the following assumption.

**Assumption 1.** *(Well-defined Optimzation Problem)*
$\operatorname{dom}(f) \neq \emptyset$ *and* $f^* := \inf_{x \in \mathbb{R}^d} f(x) > -\infty$

All of our theoretical results rely on the smoothness property of $f_i$.

**Definition 1.** *(L-smoothness). Function* $h : \mathbb{R}^d \to \mathbb{R}$ *is called L-smooth if it is continuously differentiable and its gradient is L-Lipschitz continuous for some* $L > 0$

$$\|\nabla h(x) - \nabla h(y)\| \leq L\|x - y\| \ \ \forall x, y \in \mathbb{R}^d. \tag{2}$$

For (1), we made the following assumption.

**Assumption 2.** *The* $f_1, f_2, .., f_n$ *in (1) are L-smooth functions.*

Assumptions 1 and 2 are widely used in the study of non-convex optimization problems.

Then, we define the $\epsilon$-stationary point for (1).

**Definition 2.** *($\epsilon$-approximate stationary point) Consider (1). If* $\tilde{x} \in \operatorname{dom}(f)$ *satisfies* $\mathbb{E}[\|\nabla f(\tilde{x})\|^2] \leq \epsilon^2$, *then* $\tilde{x}$ *is called $\epsilon$-approximate stationary point, where we use expectation to eliminate the randomness of the algorithm.*

Our method uses the proximal operator in DR iterations. Therefore, we provide its definition below.

**Definition 3.** *(Proximal Operator). For a proper function* $f : \mathbb{R}^n \to [-\infty, \infty]$, *we denote the proximal operator of f as*

$$\operatorname{prox}_{\beta f}(x) := \arg \min_{z \in \mathbb{R}^n} \left\{ f(z) + \frac{1}{2\beta} \|z - x\|^2 \right\}.$$

Even if $f$ is nonconvex, under Assumption 2, choosing $0 < \beta < \frac{1}{L}$ ensures that the proximal operator $\operatorname{prox}_{\beta f}$ is well-defined and single-valued. Moreover, evaluating $\operatorname{prox}_{\beta f}$ amounts to solving a strongly convex optimization problem.

However, in some cases, $\operatorname{prox}_{\beta f}$ can only be computed approximately up to an accuracy $\epsilon > 0$. That is, if $x^* := \operatorname{prox}_{\beta f}(x)$ denotes the exact solution, then an approximate solution $\tilde{x} \approx \operatorname{prox}_{\beta f}(x)$ satisfies $\|\tilde{x} - x^*\| \leq \epsilon$. Approximating $\operatorname{prox}_{\beta f}$ can be efficiently performed using various existing methods, including stochastic gradient descent (SGD) and accelerated gradient-based algorithms.

## 4 ALGORITHM

In FL, splitting methods (Pathak & Wainwright, 2020; Zhang et al., 2020) have proven effective in addressing the challenge of data heterogeneity, as they are capable of preserving consistency between local minimizers and the global minimizer. In contrast, traditional FL methods often suffer from client drift, where local minimizers diverge significantly from the global minimizer due to non-i.i.d. data distributions. Recent works (Tran-Dinh et al., 2021) demonstrates that the Douglas-Rachford (DR) splitting method can effectively address data heterogeneity in FL settings that employ naive client sampling methods. However, the heterogeneity issue under client-reshuffling-based FL remains unresolved, *i.e.*, previous DR algorithm cannot be directly applied, primarily due to the

technical challenges introduced by client reshuffling, such as sampling bias and interdependence across iterations. To address this issue, we propose the **Fed**erated **C**lient Reshuffling Douglas–**R**achford Method (**FedCDR**) and derive new rigorous theoretical analysis to support our algorithm.

Our FL algorithm supports both exact (Theorem 2) and inexact (Theorem 1) variants, depending on how the proximal subproblems are evaluated. We focus on the inexact variant, motivated by recent studies (Li et al., 2020a; Zhang et al., 2020), which demonstrate that solving the proximal subproblem inexactly is key to achieving practical and flexible performance. In particular, inexact solutions allow for a trade-off between communication and computation by adjusting the number of local iterations. This trade-off is crucial in practical federated learning systems, where computation and communication costs must be carefully balanced. Detailed discussions of this trade-off are presented in Section 6.3 and Appendix F. Therefore, we adopt the inexact variant of our method in both theory and practice.

Our new algorithm is summarized in *Algorithm 1*, and its complete derivation is provided in Appendix A. We explain the details of FedCDR as follows:

---

**Algorithm 1 Fed**erated **C**lient Reshuffling **D**ouglas–**R**achford Method (**FedCDR**)

---

1: Choose initial point $x_0 \in \mathrm{dom}(f)$, proximal coefficient $\eta > 0$, and relaxation coefficient $\alpha > 0$.

2: Initialize the server and all clients $i \in [n]$.
3: **for** meta-epoch $t = 0, \ldots, T-1$ **do**
4:    **Client Reshuffling:** sample a permutation $S^t = (S_0^t, S_1^t, \ldots, S_{R-1}^t)$ of $[n]$.
5:    **for** communication round $r = 0, \ldots, R-1$ **do**
6:       Each participating client $i \in S_r^t$ receives the current iterate $x_r^t$ from the server.
7:       Each participating client $i \in S_r^t$ performs the following updates:

$$y_{i,r+1}^t = y_{i,r}^t + \alpha(x_r^t - x_{i,r}^t), \quad x_{i,r+1}^t \approx \mathrm{prox}_{\eta f_i}(y_{i,r+1}^t),$$
$$\widehat{x}_{i,r+1}^t = 2x_{i,r+1}^t - y_{i,r+1}^t, \quad g_{i,r}^t = \widehat{x}_{i,r+1}^t - \widehat{x}_{i,r}^t.$$

8:       The server aggregates gradients: $g_r^t = \frac{1}{n} \sum_{i \in S_r^t} g_{i,r}^t$.
9:       The server updates: $x_{r+1}^t = x_r^t + g_r^t$.
10:   **end for**
11: **end for**

---

Firstly, we initialize both the server and clients at the same point $x^0$.

At each meta-epoch $t$, suppose we have a permutation $(\pi_1, \ldots, \pi_n)$ of $\{1, 2, \ldots, n\}$. We then divide $(\pi_1, \ldots, \pi_n)$ into batches, each containing $C$ clients. Specifically, we define the batch $S_r^t$ as

$$S_r^t := \{\pi_{rC}, \ldots, \pi_{(r+1)C-1}\}.$$

Let $R := \frac{n}{C}$ denote the total number of batches, and define $S^t := (S_0^t, \ldots, S_{R-1}^t)$.

Now, we perform $R$ rounds of inner training within meta-epoch $t$, where only the clients in batch $S_r^t$ participate in training at round $r$. For each communication round $r$, the server sends the current aggregated model parameter $x_r^t$ to the clients in $S_r^t$. Then, each client $i \in S_r^t$ perform Douglas-Rachford updates as follows:

$$\begin{cases} y_{i,r+1}^t := y_{i,r}^t + \alpha(x_r^t - x_{i,r}^t) \\ x_{i,r+1}^t :\approx \mathrm{prox}_{\eta f_i}(y_{i,r+1}^t) \\ \hat{x}_{i,r+1}^t := 2x_{i,r+1}^t - y_{i,r+1}^t \end{cases},$$

In the DR updates of clients, the proximal subproblem ($x_{i,r+1}^t :\approx \mathrm{prox}_{\eta f_i}(y_{i,r+1}^t)$) is solved approximately up to an absolute accuracy of $\epsilon_{t,i,r} \geq 0$.

Once all the local updates are finished, each participating client $i$ send $g_{i,r}^t := \hat{x}_{i,r+1}^t - \hat{x}_{i,r}^t$ back to the server. Let $g_r^t = \frac{1}{n} \sum_{i \in S_r^t} g_{i,r}^t$. Then, the server updates the global model as follows:

$$x_{r+1}^t = x_r^t + g_r^t \tag{3}$$

and start the next round.

## 5  CONVERGENCE ANALYSIS

In this section, we analyze the convergence of *Algorithm 1*. We begin by establishing the convergence of its inexact variant.

**Theorem 1.** *Consider problem (1) and suppose that Assumptions 1 and 2 hold. Let the sequence $\{(x_r^t, x_{i,r}^t, y_{i,r}^t, \hat{x}_{i,r}^t)\}$ be generated by Algorithm 1 using coefficients $\alpha$, $\eta$. Let $\tilde{x}$ be a random variable selected uniformly at random from the set $\{x_0^0, x_1^0, \ldots, x_{R-1}^0, x_0^1, \ldots, x_{R-1}^{T-1}\}$ and returned as the output of Algorithm 1. Assume the coefficients satisfy $0 < \eta < \eta_0$, where $\eta_0$ is defined in (59). Then, the following bound holds:*

$$\mathbb{E}[\|\nabla f(\tilde{x})\|^2] \leq \frac{\mathcal{A}_1[f(x^0) - f^*]}{T} + \frac{1}{nT} \sum_{t=0}^{T-1} \sum_{r=0}^{R-1} \sum_{i=1}^{n} \left( \mathcal{A}_1 \mathcal{B}_1 \epsilon_{t,i,r}^2 + \mathcal{A}_1 \mathcal{B}_2 \epsilon_{t,i,r+1}^2 \right), \quad (4)$$

*where the constants $\mathcal{A}_1$, $\mathcal{B}_1$, and $\mathcal{B}_2$ are defined in (58).*

*Let the accuracies $\epsilon_{t,i,r}$ for all $i \in [n]$, $0 \leq r \leq R$, and $t \geq 0$ be chosen such that*

$$\frac{1}{n} \sum_{t=0}^{T-1} \sum_{r=0}^{R} \sum_{i=1}^{n} \epsilon_{t,i,r}^2 \leq M,$$

*for some constant $M > 0$ and for all $R \geq 0$ and $T \geq 0$. Then, if Algorithm 1 is run for at most*

$$K := \left\lceil \frac{\mathcal{A}_1(f(x^0) - f^*)}{\epsilon^2} + \frac{(\mathcal{A}_1 \mathcal{B}_1 + \mathcal{A}_1 \mathcal{B}_2)M}{\epsilon^2} \right\rceil$$

*communication rounds, then $\tilde{x}$ is an $\epsilon$-approximate stationary point of (1), as defined in Definition 2.*

**Remark 1.** *(Choice of accuracies $\epsilon_{t,i,r}$) To guarantee the condition $\frac{1}{n} \sum_{i=1}^{n} \sum_{t=0}^{T-1} \sum_{r=0}^{R} \epsilon_{t,i,r}^2 \leq M$ in Theorem 1 for a given constant $M$ and all $R \geq 0$, $T \geq 0$, one possible choice is, e.g., $\epsilon_{t,i,r}^2 := \frac{M}{2[r+t(R+1)+1]^2}$ for all $i \in [n]$, $r \geq 0$, and $t \geq 0$. With this choice, and using the well-known bound $\sum_{k=1}^{\infty} \frac{1}{k^2} = \frac{\pi^2}{6} < 2$, we have*

$$\frac{1}{n} \sum_{i=1}^{n} \sum_{t=0}^{T-1} \sum_{r=0}^{R} \epsilon_{t,i,r}^2 = \frac{M}{2} \sum_{t=0}^{T-1} \sum_{r=0}^{R} \frac{1}{[r+t(R+1)+1]^2} = \frac{M}{2} \sum_{k=1}^{T(R+1)} \frac{1}{k^2} \leq M.$$

**Remark 2.** *The proof of this theorem is presented in Appendix B.*

- (Optimal complexity) *We achieve the state-of-the-art communication complexity of $O(\epsilon^{-2})$ for solving the nonconvex optimization problem (1) using federated learning methods. This result matches the lower-bound complexity up to a constant factor.*

- (Heterogeneity bound free) *Our convergence rate in the upper bound is **NOT** dependent on client heterogeneity. This is a key merit of our method compared with previous work. For example, in Theorem 6.2 of Malinovsky et al. (2023a), the convergence rate is related to the data heterogeneity, i.e. $\tilde{\sigma}_*^2 := \frac{1}{n} \sum_{i=1}^{n} \|\nabla f_i(x^*)\|^2$. Moreover, the error term containing $\tilde{\sigma}_*^2$ grows **quadratically** with the number of clients. This implies that as the number of clients increases, their method requires an extremely small step size to guarantee convergence, slowing down the training process. In contrast, our method does not suffer from this issue. This finding provides theoretical justification for why our method effectively mitigates data heterogeneity problems under client-reshuffling-based FL settings.*

- (Approximate local solver) *Our algorithm employs an approximate evaluation of the proximal operator in the local solver, and this approximation does **NOT** degrade the convergence rate in terms of big-$\mathcal{O}$ complexity. Specifically, the associated error term diminishes as the number of iterations increases. Approximate evaluation is essential for practical implementations and computational efficiency. In Corollaries 1 and 2, we demonstrate that the use of inexact evaluations influences only the constant factor in the convergence bound, without altering the overall asymptotic rate.*

The following corollary gives a specific choice of coefficients in *Algorithm 1*.

**Corollary 1.** *Consider (1). Suppose that Assumptions 1 and 2 hold. Let the sequence $\{(x_r^t, x_{i,r}^t, y_{i,r}^t, \hat{x}_{i,r}^t)\}$ be generated by Algorithm 1 using coefficients $\alpha$ and $\eta$. Let $\tilde{x}$ be a random variable chosen uniformly at random from $\{x_0^0, x_1^0, ..., x_{R-1}^0, x_0^1, ..., x_{R-1}^{T-1}\}$ and returned as the output of Algorithm 1. Assume $\alpha = 1$, $\eta = \frac{1}{4L}$ and $\gamma = \frac{1}{25}$. Then, the following bound holds*

$$\mathbb{E}[\|\nabla f(\tilde{x})\|^2] \leq \frac{625L[(f(x^0) - f^*) + 50000LM]}{4T}. \tag{5}$$

*Moreover, if Algorithm 1 is run for at most*

$$K := \left\lceil \frac{625L[(f(x^0) - f^*) + 50000LM]}{4\epsilon^2} \right\rceil$$

*communication rounds, then $\tilde{x}$ is an $\epsilon$-approximate stationary point of (1), as defined in Definition 2.*

If we consider the exact evaluation of the proximal subproblem, i.e., let the accuracies $\epsilon_{t,i,r} = 0$ for all $i \in [n]$, $0 \leq r \leq R$, and $t \geq 0$, the analysis can be significantly simplified. In this case, we obtain following convergence result for the exact variant of *Algorithm 1*.

**Theorem 2.** *Consider problem (1) and suppose that Assumptions 1 and 2 hold. Let the sequence $\{(x_r^t, x_{i,r}^t, y_{i,r}^t, \hat{x}_{i,r}^t)\}$ be generated by Algorithm 1 using coefficients $\alpha$ and $\eta$. Let $\tilde{x}$ be a random variable chosen uniformly at random from the set $\{x_0^0, x_1^0, \ldots, x_{R-1}^{T-1}\}$ and returned as the output of Algorithm 1. Assume the coefficients satisfy $0 < \eta < \eta_1$, where $\eta_1$ is defined in (99). Then, the following bound holds:*

$$\mathbb{E}[\|\nabla f(\tilde{x})\|^2] \leq \frac{\mathcal{A}[f(x^0) - f^*]}{T}, \tag{6}$$

*where constant $\mathcal{A}$ is defined in (98).*

*Moreover, if Algorithm 1 is run for at most $K := \left\lceil \frac{\mathcal{A}[f(x^0) - f^*]}{\epsilon^2} \right\rceil$ communication rounds, then $\tilde{x}$ is an $\epsilon$-approximate stationary point of (1), as defined in Definition 2.*

**Remark 3.** *The proof of this theorem (exact variant of Algorithm 1) is presented in Appendix C.*

The following corollary gives a specific choice of coefficients in the exact variant of *Algorithm 1*.

**Corollary 2.** *Consider problem (1) and suppose that Assumptions 1 and 2 hold. Let the sequence $\{(x_r^t, x_{i,r}^t, y_{i,r}^t, \hat{x}_{i,r}^t)\}$ be generated by Algorithm 1 using coefficients $\alpha$ and $\eta$. Let $\tilde{x}$ be a random variable selected uniformly at random from the set $\{x_0^0, x_1^0, \ldots, x_{R-1}^0, x_0^1, \ldots, x_{R-1}^{T-1}\}$ and returned as the output of Algorithm 1. Assume $\alpha = 1$ and $\eta = \frac{1}{4L}$. Then, the following holds*

$$\mathbb{E}[\|\nabla f(\tilde{x})\|^2] \leq \frac{125L[f(x^0) - f^*]}{4T}. \tag{7}$$

*Moreover, if Algorithm 1 is run for at most $K := \left\lceil \frac{125L[f(x^0) - f^*]}{4\epsilon^2} \right\rceil$ communication rounds, then $\tilde{x}$ is an $\epsilon$-approximate stationary point of (1), as defined in Definition 2.*

**Remark 4.** *Corollary 2 adopts the same proximal coefficient $\eta$ and relaxation coefficient $\alpha$ as in Corollary 1. However, the constant in the convergence rate is significantly smaller in Corollary 2, which is attributed to the exact evaluation of the proximal subproblem. This theoretical result highlights that exact computation of the proximal subproblem leads to improved convergence performance of Algorithm 1.*

## 6 EXPERIMENTS

To study the ability of FedCDR to address the data heterogeneity challenge in client-reshuffling-based FL settings, we compare our method with client-reshuffling variants of state-of-the-art FL algorithms on both synthetic and benchmark datasets. For each dataset, we apply these algorithms under client data distributions with varying degrees of heterogeneity. Our results show that FedCDR significantly outperforms existing methods (Section 6.2). In addition, we evaluate the communication-computation trade-off in our method (Section 6.3), and conduct ablation studies (Section 6.4) to demonstrate that client reshuffling can accelerate FL training.

Table 1: Top-1 accuracy on synthetic datasets. Reported values are mean $\pm$ standard deviation.

| Methods | Synthetic-(0,0) | Synthetic-(1,1) | Synthetic-(5,5) |
|---|---|---|---|
| CR+FedAvg | $88.12 \pm 0.21$ | $77.54 \pm 0.42$ | $46.15 \pm 1.32$ |
| CR+FedProx | $88.45 \pm 0.15$ | $80.34 \pm 0.24$ | $65.80 \pm 0.45$ |
| CR+SCAFFOLD | $91.34 \pm 0.25$ | $88.15 \pm 0.19$ | $77.92 \pm 0.32$ |
| CR+FedDC | $92.25 \pm 0.14$ | $90.04 \pm 0.21$ | $79.78 \pm 0.26$ |
| FedCDR | $\mathbf{93.00 \pm 0.24}$ | $\mathbf{92.02 \pm 0.17}$ | $\mathbf{85.76 \pm 0.37}$ |

Table 2: Top-1 accuracy on benchmark datasets. Reported values are mean $\pm$ standard deviation.

| Methods | $\alpha = 0.1$ | | | $\alpha = 1.0$ | | |
|---|---|---|---|---|---|---|
| | MNIST | CIFAR10 | CIFAR100 | MNIST | CIFAR10 | CIFAR100 |
| CR+FedAvg | $97.45 \pm 0.14$ | $69.79 \pm 0.70$ | $55.68 \pm 0.70$ | $97.79 \pm 0.07$ | $78.29 \pm 0.50$ | $65.89 \pm 0.41$ |
| CR+FedProx | $97.68 \pm 0.09$ | $70.34 \pm 0.56$ | $58.34 \pm 0.62$ | $97.72 \pm 0.05$ | $79.34 \pm 0.34$ | $65.45 \pm 0.43$ |
| CR+SCAFFOLD | $98.06 \pm 0.05$ | $74.45 \pm 0.40$ | $63.47 \pm 0.34$ | $98.02 \pm 0.06$ | $82.35 \pm 0.30$ | $69.67 \pm 0.25$ |
| CR+FedDC | $98.00 \pm 0.03$ | $75.24 \pm 0.25$ | $64.56 \pm 0.35$ | $98.29 \pm 0.03$ | $83.56 \pm 0.24$ | $69.28 \pm 0.34$ |
| FedCDR | $\mathbf{98.17 \pm 0.05}$ | $\mathbf{76.98 \pm 0.31}$ | $\mathbf{67.89 \pm 0.32}$ | $\mathbf{98.40 \pm 0.04}$ | $\mathbf{85.29 \pm 0.26}$ | $\mathbf{71.90 \pm 0.28}$ |

## 6.1 EXPERIMENTAL SETUP

**Datasets and data assignment**  We use both synthetic and benchmark datasets to evaluate our method. For the synthetic datasets, we generate multiple heterogeneous variants of synthetic datasets by using different pairs of $(\alpha, \beta)$ to control the degree of heterogeneity, denoted as Synthetic-$(\alpha, \beta)$. Larger values of $\alpha$ and $\beta$ correspond to higher levels of heterogeneity. In our experiments, we consider three levels of heterogeneity: Synthetic-(0,0), Synthetic-(1,1), and Synthetic-(5,5). We generate 500 clients for training. For the benchmark datasets, we use MNIST, CIFAR-10, and CIFAR-100 (Krizhevsky, 2009). To create heterogeneous client data splits, we adopt Dirichlet-based partitioning strategies (Wang et al., 2020). Specifically, we sample $p_c \sim \mathrm{Dir}_C(\alpha)$ from a $C$-dimensional symmetric Dirichlet distribution and assign a $p_{k,c}$-fraction of class $c$ samples to client $k$. ($C$ is the number of class.) A smaller $\alpha$ corresponds to greater heterogeneity. We generate 500 clients for training. Further details on datasets and partitioning strategies are provided in Appendix G.1.

**Baselines**  We compare our method with client-reshuffling variants of state-of-the-art FL algorithms, denoted as CR(client reshuffling)+Algorithm. Specifically, we consider the widely used algorithms FedAvg (McMahan et al., 2017a), FedProx (Li et al., 2020a), SCAFFOLD (Karimireddy et al., 2020), and FedDC (Gao et al., 2022).

**Models and training schemes**  For the synthetic datasets and MNIST, we use a fully connected neural network. For CIFAR-10 and CIFAR-100, we use ResNet-20 (He et al., 2016). For the inexact proximal solver of FedCDR, we follow the experimental setup in (Li et al., 2020a) and adopt SGD as the approximate local solver. Specifically, each client performs local updates for 10 epochs, where each epoch consists of multiple mini-batch SGD iterations. Due to the strong convex property of the proximal subproblem, the local solution can be regarded as an approximation of the exact solution. We tune all the hyperparameters using grid search. We select $10\%$ of the clients to participate in each communication round to simulate a low client participation setting and run FL algorithms for $400$ communication rounds. We run all algorithms five times and report the mean and standard deviation. Additional details regarding model architectures and training schemes are provided in Appendix G.2. Detailed discussions of the communication and computation overhead of all methods are presented in Appendix E.

## 6.2 MAIN RESULTS

Table 1 displays the top-1 accuracy of all algorithms on synthetic datasets with three degrees of data heterogeneity. Larger values of $\alpha$ and $\beta$ in table 1 correspond to higher levels of heterogeneity. Table 2 presents the top-1 accuracy of all algorithms on three benchmark datasets with two degrees of heterogeneity, where smaller $\alpha$ indicates greater heterogeneity. We observe that our algorithm consistently achieves strong performance, regardless of the degree of heterogeneity. These results

Table 3: Communication-computation trade-off of FedCDR. Comm. rounds indicates communication rounds required to achieve the target top-1 accuracy of $98\%$ in MNIST. Communication and computation speedups are measured relative to the setting with a local epoch of 10.

| Local epoch | $\alpha = 0.1$ | | | $\alpha = 1.0$ | | |
|---|---|---|---|---|---|---|
| | Comm. rounds | Comm. speedup | Comp. speedup | Comm. rounds | Comm. speedup | Comp. speedup |
| 3 | 161 | $\times 0.373$ | $\times 3.333$ | 125 | $\times 0.392$ | $\times 3.333$ |
| 5 | 112 | $\times 0.536$ | $\times 2.500$ | 89 | $\times 0.551$ | $\times 2.500$ |
| 10 | 60 | – | – | 49 | – | – |
| 20 | 43 | $\times 1.395$ | $\times 0.500$ | 32 | $\times 1.531$ | $\times 0.500$ |

Table 4: Comparison of FedDR and FedCDR under different participation levels. Comm. rounds indicates the communication rounds required to achieve the target top-1 accuracy of $90\%$.

| Methods | Metrics | Extreme Low Participation (5%) | | Low Participation (10%) | | High Participation (50%) | |
|---|---|---|---|---|---|---|---|
| | | Synthetic-(0,0) | Synthetic-(1,1) | Synthetic-(0,0) | Synthetic-(1,1) | Synthetic-(0,0) | Synthetic-(1,1) |
| FedDR | Top-1 accuracy | 91.67 | 90.85 | 92.46 | 91.43 | 93.01 | 92.02 |
| | Comm. rounds | 235 | 347 | 104 | 134 | 35 | 44 |
| FedCDR | Top-1 accuracy | 92.47 | 91.89 | 92.97 | 91.96 | 93.78 | 93.01 |
| | Comm. rounds | 57 | 68 | 39 | 47 | 28 | 34 |
| | Comm. speedup | $\times 4.123$ | $\times 5.103$ | $\times 2.667$ | $\times 2.851$ | $\times 1.250$ | $\times 1.294$ |

confirm that FedCDR effectively mitigates the data heterogeneity challenge in client-reshuffling-based FL settings.

### 6.3 Communication-computation Trade-off

To illustrate the communication-computation trade-off in FedCDR, we vary the number of local epochs used by the inexact solver and compare the resulting communication and computation speedups. Table 3 reports the number of communication rounds each method requires to achieve the target top-1 accuracy ($98\%$) on MNIST, along with the corresponding communication and computation speedups. We observe that increasing the number of local epochs (i.e., more local computation) reduces the communication cost, whereas decreasing the number of local epochs (i.e., less local computation) increases the communication cost. This communication-computation trade-off is useful for applying the algorithm in scenarios with different hardware capabilities. Further discussion and simulations of scenarios with different hardware capabilities are presented in Appendix F.

### 6.4 Ablation Studies

To demonstrate that client reshuffling accelerates FL training, we compare FedCDR with FedDR (Tran-Dinh et al., 2021) under three client participation levels: extreme low, low, and high. FedDR is an FL baseline that employs DR splitting but does not incorporate client reshuffling. In the extreme low-participation setting, only $5\%$ of clients participate in each communication round. In the low-participation setting, $10\%$ of clients participate, while in the high-participation setting, $50\%$ of clients participate. Table 4 reports the top-1 accuracy across all settings, the number of communication rounds required to reach the target accuracy ($90\%$), and the corresponding speedup of FedCDR. We find that FedCDR consistently converges faster than FedDR in all scenarios, while both algorithms achieve similar top-1 accuracy after the full training procedure. These results show that client reshuffling can accelerate training, whereas DR splitting mitigates the data heterogeneity challenge. Moreover, FedCDR achieves greater training speedup under the extreme low-participation setting, highlighting the advantage of client reshuffling when client participation is severely limited.

## 7 Conclusion

In this work, we propose a novel method to address the data heterogeneity challenge in client-reshuffling-based federated learning. We theoretically demonstrate that our method achieve the state-of-the-art $O(\epsilon^{-2})$ convergence rate, and importantly, this convergence is *independent of the degree of data heterogeneity* across clients. Our numerical experiments demonstrate that our method consistently outperforms existing algorithms on both synthetic and benchmark datasets.

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

# Part I

# Appendix

## Table of Contents

## A   Derivation of *Agorithm 1* (FedCDR)

**Distributed Douglas-Rachford.**   We begin by reformulating the standard federated learning problem with a consensus constraint into an equivalent optimization problem. To solve this problem efficiently in a decentralized manner, we apply the Douglas–Rachford (DR) splitting method, which allows us to decompose the centralized formulation into a distributed algorithm amenable to federated optimization. We further enhance this framework by integrating client reshuffling and allowing for inexact local updates, which are practical considerations in real-world federated settings. The resulting method is formalized in *Algorithm 1*.

Since the standard federated learning problem is formulated as (1), we can equivalently express it as the following constrained optimization problem by adding $x_1 = x_2 = \cdots = x_n$:

$$\min_{x_1, x_2, \ldots, x_n} f(\mathbf{x}) = \frac{1}{n} \sum_{i=1}^{n} f_i(x_i) \quad \text{s.t.} \quad x_1 = x_2 = \cdots = x_n, \tag{8}$$

where $\mathbf{x} := [x_1; x_2; \ldots; x_n]$ represents the local parameters of all $n$ clients.

Note that $\mathbf{x} \in \mathbb{R}^{nd}$. The constraint $x_1 = x_2 = \cdots = x_n$ can be viewed as defining a solution space corresponding to a particular linear equation. Equivalently, we can express this constraint using a linear subspace $\mathcal{H} \subseteq \mathbb{R}^{nd}$, where

$$\mathcal{H} := \left\{ \mathbf{x} \in \mathbb{R}^{nd} \mid x_1 = x_2 = \cdots = x_n \right\}.$$

To eliminate the explicit constraint in (8), we introduce the indicator function $I_{\mathcal{H}}$ associated with the subspace $\mathcal{H}$, defined as

$$I_{\mathcal{H}}(\mathbf{x}) := \begin{cases} 0, & \text{if } \mathbf{x} \in \mathcal{H}, \\ +\infty, & \text{otherwise.} \end{cases}$$

Using this indicator function, we can reformulate the constrained optimization problem as an unconstrained problem:

$$\min_{\mathbf{x} \in \mathbb{R}^{nd}} F(\mathbf{x}) := f(\mathbf{x}) + I_{\mathcal{H}}(\mathbf{x}) = \frac{1}{n} \sum_{i=1}^{n} f_i(x_i) + I_{\mathcal{H}}(\mathbf{x}), \tag{9}$$

where $f$ is differentiable while $I_{\mathcal{H}}$ is non-differentiable. In this setting, we apply the Dougla-Rachford (DR) splitting method to decouple the optimization of the differentiable function $f$ and the non-differentiable indicator function $I_{\mathcal{H}}$, thereby simplifying the overall optimization process. The Douglas-Rachford iterations for solving problem (9) are given by:

$$\begin{cases} \mathbf{y}^{k+1} := \mathbf{y}^k + \alpha(\mathbf{w}^k - \mathbf{x}^k), \\ \mathbf{x}^{k+1} := \text{prox}_{n\eta f}(\mathbf{y}^{k+1}), \\ \mathbf{w}^{k+1} := \text{prox}_{n\eta I_{\mathcal{H}}}(2\mathbf{x}^{k+1} - \mathbf{y}^{k+1}), \end{cases} \tag{10}$$

where $k$ denotes the iteration index, $\eta > 0$ is the proximal coefficient, and $\alpha \in (0, 2]$ is a relaxation coefficient as discussed in Themelis & Patrinos (2020). In classical Douglas-Rachford literature (Lions & Mercier, 1979), the coefficient $\alpha$ is typically set to $\alpha = 1$. To clarify the definitions of $\text{prox}_{n\eta f}$ and $\text{prox}_{n\eta I_{\mathcal{H}}}$, we have $\text{prox}_{n\eta f}(x) = \arg\min_{z \in \mathbb{R}^n} \left\{ f(z) + \frac{1}{2n\eta}\|z - x\|^2 \right\}$, and $\text{prox}_{n\eta I_{\mathcal{H}}}(x) = \arg\min_{z \in \mathbb{R}^n} \left\{ I_{\mathcal{H}}(z) + \frac{1}{2n\eta}\|z - x\|^2 \right\}$.

Since $I_{\mathcal{H}}$ is the indicator function of a linear subspace, we can simplify the second proximal step in the Douglas-Rachford iteration as follows:

$$\mathbf{w}^{k+1} = \text{prox}_{n\eta I_{\mathcal{H}}}(2\mathbf{x}^{k+1} - \mathbf{y}^{k+1})$$

$$= \arg\min_{\mathbf{w}} \left[ n\eta I_{\mathcal{H}}(\mathbf{w}) + \frac{1}{2} \left\| \mathbf{w} - (2\mathbf{x}^{k+1} - \mathbf{y}^{k+1}) \right\|^2 \right]$$

$$\overset{(a)}{=} \arg\min_{w_1, w_2, \ldots, w_n} \left\| \mathbf{w} - (2\mathbf{x}^{k+1} - \mathbf{y}^{k+1}) \right\|^2 \quad \text{s.t.} \quad w_1 = w_2 = \cdots = w_n \tag{11}$$

$$\overset{(b)}{=} \arg\min_{w_1, w_2, \ldots, w_n} \sum_{i=1}^{n} \left\| w_i - (2x_i^{k+1} - y_i^{k+1}) \right\|^2 \quad \text{s.t.} \quad w_1 = w_2 = \cdots = w_n,$$

where step (a) rewrites the indicator as an explicit constraint, and step (b) decomposes the norm coordinate-wise.

Defining $w := w_1$, the solution to (11) is the vector $\underbrace{(w; \ldots; w)}_{n \text{ copies}}$, where

$$w = \arg\min_w \sum_{i=1}^n \left\| w - (2x_i^{k+1} - y_i^{k+1}) \right\|^2 \quad \Rightarrow \quad w = \frac{1}{n} \sum_{i=1}^n \left( 2x_i^{k+1} - y_i^{k+1} \right). \quad (12)$$

We give the detailed proof of (12):

*Proof.* Notate the superscript $w^{(a)}$ means the $a$-th element of parameter $w$. Since the parameter have $d$ dimensions, i.e., $w \in \mathbb{R}^d$, then we have

$$\min_w \sum_{i=1}^n \left\| w - (2x_i^{k+1} - y_i^{k+1}) \right\|^2 = \min_w \sum_{i=1}^n \sum_{a=1}^d (w^{(a)} - (2x_i^{k+1} - y_i^{k+1})^{(a)})^2$$

$$= \min_{w^{(1)}, \cdots, w^{(d)}} \sum_{a=1}^d \sum_{i=1}^n (w^{(a)} - (2x_i^{k+1} - y_i^{k+1})^{(a)})^2$$

We define $h_a(x) = \sum_{i=1}^n (x - (2x_i^{k+1} - y_i^{k+1})^{(a)})^2$. Therefore, we have

$$\min_w \sum_{i=1}^n \left\| w - (2x_i^{k+1} - y_i^{k+1}) \right\|^2 = \min_{w^{(1)}, \cdots, w^{(d)}} \sum_{a=1}^d h_d(w^{(a)})$$

i.e.

$$w^{(a)} = \arg\min_x h_a(x)$$

To solve this problem, we view $h_a(x)$ as a quadratic function of $x$. Therefore, we can use quadratic minimum formula $x^* = -\frac{B}{2A}$ for $f(x) = Ax^2 + Bx + C(A \neq 0)$ to get the minimum. That is

$$w^{(a)} = \arg\min_x h_a(x) = \frac{1}{n} \sum_{i=1}^n (2x_i^{k+1} - y_i^{k+1})^{(a)}$$

Therefore, we get the conclusion:

$$w = \frac{1}{n} \sum_{i=1}^n \left( 2x_i^{k+1} - y_i^{k+1} \right).$$

$\square$

Furthermore, since $f(\mathbf{x}) = \frac{1}{n} \sum_{i=1}^n f_i(x_i)$, we can simplify the first proximal step as follows:

$$\mathbf{x}^{k+1} = \text{prox}_{n\eta f}(\mathbf{y}^{k+1})$$

$$= \arg\min_{\mathbf{x}} \left[ f(\mathbf{x}) + \frac{1}{2n\eta} \|\mathbf{x} - \mathbf{y}^{k+1}\|^2 \right]$$

$$\stackrel{(a)}{=} \arg\min_{\mathbf{x}} \left\{ \frac{1}{n} \sum_{i=1}^n \left[ f_i(x_i) + \frac{1}{2\eta} \|x_i - y_i^{k+1}\|^2 \right] \right\}$$

$$= \left( \arg\min_{x_1} \left[ f_1(x_1) + \frac{1}{2\eta} \|x_1 - y_1^{k+1}\|^2 \right]; \ldots; \arg\min_{x_n} \left[ f_n(x_n) + \frac{1}{2\eta} \|x_n - y_n^{k+1}\|^2 \right] \right)$$

$$= \left( \text{prox}_{\eta f_1}(y_1^{k+1}); \ldots; \text{prox}_{\eta f_n}(y_n^{k+1}) \right),$$

(13)

where step (a) follows from the separability of $f$.

Therefore, we can implement the centralized Douglas-Rachford iteration (10) in a distributed manner, particularly within the federated learning framework. Specifically, each distributed client can

compute the local update $x_i^{k+1} = \text{prox}_{\eta f_i}(y_i^{k+1})$ according to (13), while the central server aggregates the pseudo-gradients to compute the step in (11).

To streamline the iteration process, we introduce an auxiliary variable $\hat{x}_i^k := 2x_i^k - y_i^k$, and reformulate the centralized Douglas–Rachford method (10) into the following decentralized form:

$$
\begin{cases}
y_i^{k+1} := y_i^k + \alpha(w^k - x_i^k), & \text{for } i = 1, \ldots, n, \\
x_i^{k+1} := \text{prox}_{\eta f_i}(y_i^{k+1}), & \text{for } i = 1, \ldots, n, \\
\hat{x}_i^{k+1} := 2x_i^{k+1} - y_i^{k+1}, & \text{for } i = 1, \ldots, n, \\
w^{k+1} := \dfrac{1}{n} \sum_{i=1}^{n} \hat{x}_i^{k+1}.
\end{cases}
\tag{14}
$$

The first three updates in (14) are performed independently by each client in a fully parallel and distributed fashion. The final step is executed by the central server to aggregate the intermediate results and update the global variable $w^{k+1}$. This decentralized procedure is mathematically equivalent to the centralized Douglas–Rachford iteration (10), and is well-suited for implementation in federated learning systems.

**Clients Reshuffling** In our algorithm, we incorporate client reshuffling to improve client participation efficiency. At the beginning of each meta-epoch $t$, we generate a random permutation of all clients, denoted by $(\pi_1, \ldots, \pi_n)$, where $\pi_i \in \{1, 2, \ldots, n\}$. The permuted client list is then partitioned into batches of size $C$. Specifically, the batch of clients selected at round $r$ within meta-epoch $t$ is defined as
$$
S_r^t := \{\pi_{rC}, \ldots, \pi_{(r+1)C-1}\}.
$$
Let $R := \frac{n}{C}$ denote the total number of batches per meta-epoch, and define the complete reshuffling schedule as $S^t := (S_0^t, \ldots, S_{R-1}^t)$. This reshuffling procedure is repeated at the beginning of each meta-epoch $t$, ensuring that the selection order varies across training stages.

We now introduce the necessary notation and adapt Algorithm (14) to incorporate client reshuffling. Each meta-epoch $t$ consists of $R$ inner communication rounds, indexed by $r \in \{0, 1, \ldots, R-1\}$. In the $t$-th meta-epoch and $r$-th inner round, the participating client set is denoted by $S_r^t \subseteq [n]$. For each client $m \in S_r^t$, we perform the local update steps as described in Algorithm (14).

Using this notation, we replace the global iteration index $k$ with the pair $(t, r)$, where the transformation between indices is given by:
$$
k = tR + r,
$$
where $t \geq 0$ and $0 \leq r \leq R-1$. This provides a one-to-one correspondence between the global iteration index and the client reshuffling schedule.

Then, we introduce our client-reshuffling-based algorithm.

First, within a given meta-epoch $t$, at round $r$, each selected client $i \in S_r^t$ performs the following local updates:

$$
\begin{cases}
y_{i,r+1}^t := y_{i,r}^t + \alpha(x_r^t - x_{i,r}^t), \\
x_{i,r+1}^t := \text{prox}_{\eta f_i}(y_{i,r+1}^t), \\
\hat{x}_{i,r+1}^t := 2x_{i,r+1}^t - y_{i,r+1}^t.
\end{cases}
\tag{15}
$$

For the inactive clients $i \notin S_r^t$, no computation is performed, and their states remain unchanged:

$$
\begin{cases}
y_{i,r+1}^t := y_{i,r}^t, \\
x_{i,r+1}^t := x_{i,r}^t, \\
\hat{x}_{i,r+1}^t := \hat{x}_{i,r}^t.
\end{cases}
\tag{16}
$$

After all participating clients complete their local updates, the server performs the aggregation step:

$$
x_{r+1}^t := \frac{1}{n} \sum_{i=1}^{n} \hat{x}_{i,r+1}^t.
\tag{17}
$$

At the beginning of the next meta-epoch $t + 1$, the initialization of state variables is given by:

$$
\begin{aligned}
x_0^{t+1} &:= x_R^t, \\
x_{i,0}^{t+1} &:= x_{i,R}^t, \\
y_{i,0}^{t+1} &:= y_{i,R}^t, \\
\hat{x}_{i,0}^{t+1} &:= \hat{x}_{i,R}^t.
\end{aligned}
\tag{18}
$$

Therefore, the overall process of our algorithm can be described as follows.

**Initialization:** Choose an initial parameter $x^0 \in \mathrm{dom}(f)$, and coefficients $\eta > 0$ and $\alpha > 0$.
Initialize the server with $x_0^0 := x^0$.
Initialize each client $i \in [n]$ with:

$$
y_{i,0}^0 := x^0, \quad x_{i,0}^0 := \mathrm{prox}_{\eta f_i}(y_{i,0}^0), \quad \hat{x}_{i,0}^0 := 2x_{i,0}^0 - y_{i,0}^0.
$$

**Meta-Epoch $t$:** Sample a permutation of clients and partition them into batches:
$S^t = (S_0^t, S_1^t, \ldots, S_{R-1}^t)$, where each $S_r^t \subset [n]$ is of size $C$.

**Inner Communication Round $r \in \{0, \ldots, R-1\}$:**

- **Communication:** Each active client $i \in S_r^t$ receives $x_r^t$ from the server.
- **Active Client Update:** Each active client $i \in S_r^t$ performs the following updates:
$$
\begin{aligned}
y_{i,r+1}^t &:= x_{i,r}^t + \alpha(x_r^t - x_{i,r}^t), \\
x_{i,r+1}^t &:= \mathrm{prox}_{\eta f_i}(y_{i,r+1}^t), \\
\hat{x}_{i,r+1}^t &:= 2x_{i,r+1}^t - y_{i,r+1}^t.
\end{aligned}
$$
- **Inactive Client Update:** Each inactive client $i \notin S_r^t$ maintains its previous values:
$$
\begin{aligned}
y_{i,r+1}^t &:= y_{i,r}^t, \\
x_{i,r+1}^t &:= x_{i,r}^t, \\
\hat{x}_{i,r+1}^t &:= \hat{x}_{i,r}^t.
\end{aligned}
$$
- **Communication:** Each active client $i \in S_r^t$ sends $\hat{x}_{i,r+1}^t$ to the server.
- **Server Aggregation:** The server aggregates the updates as:
$$
x_{r+1}^t := \frac{1}{n} \sum_{i=1}^n \hat{x}_{i,r+1}^t.
$$

To simplify the server aggregation step of our algorithm, we derive the following:

$$
\begin{aligned}
x_{r+1}^t &= \frac{1}{n} \sum_{i=1}^n \hat{x}_{i,r+1}^t \\
&= \frac{1}{n} \sum_{i \in S_r^t} \hat{x}_{i,r+1}^t + \frac{1}{n} \sum_{i \notin S_r^t} \hat{x}_{i,r+1}^t \\
&\overset{(16)}{=} \frac{1}{n} \sum_{i \in S_r^t} \hat{x}_{i,r+1}^t + \frac{1}{n} \sum_{i \notin S_r^t} \hat{x}_{i,r}^t \\
&= \frac{1}{n} \sum_{i \in S_r^t} \left( \hat{x}_{i,r}^t + (\hat{x}_{i,r+1}^t - \hat{x}_{i,r}^t) \right) + \frac{1}{n} \sum_{i \notin S_r^t} \hat{x}_{i,r}^t \\
&= \frac{1}{n} \sum_{i=1}^n \hat{x}_{i,r}^t + \frac{1}{n} \sum_{i \in S_r^t} (\hat{x}_{i,r+1}^t - \hat{x}_{i,r}^t) \\
&\overset{(17)}{=} x_r^t + \frac{1}{n} \sum_{i \in S_r^t} g_{i,r}^t = x_r^t + g_r^t,
\end{aligned}
$$

where we define the update increments as $g_{i,r}^t := \hat{x}_{i,r+1}^t - \hat{x}_{i,r}^t$, and their aggregate as $g_r^t := \frac{1}{n} \sum_{i \in S_r^t} g_{i,r}^t$. This simplified update rule is implemented in *Algorithm 1*.

Furthermore, we assume that we can only approximate the proximal operator $\text{prox}_{\eta f_i}$ up to a given accuracy for all $i \in [n]$. In this case, we replace the exact proximal step $x_{i,r}^t := \text{prox}_{\eta f_i}(y_{i,r}^t)$ with its approximation $x_{i,r}^t \approx \text{prox}_{\eta f_i}(y_{i,r}^t)$ such that

$$\|x_{i,r}^t - \text{prox}_{\eta f_i}(y_{i,r}^t)\| \leq \epsilon_{t,i,r}, \tag{19}$$

for some given accuracy $\epsilon_{t,i,r} \geq 0$.

Since $x_{i,r}^t$ is approximately computed from $\text{prox}_{\eta f_i}(y_{i,r}^t)$ as in (19), we have the decomposition:

$$x_{i,r}^t = z_{i,r}^t + e_{i,r}^t, \quad \text{where} \quad z_{i,r}^t := \text{prox}_{\eta f_i}(y_{i,r}^t) \quad \text{and} \quad \|e_{i,r}^t\| \leq \epsilon_{t,i,r}, \tag{20}$$

where $e_{i,r}^t$ denotes the error vector. Note that $z_{i,r}^t$ does not exist in the actual implementation of *Algorithm 1*, and we only have access to its approximation $x_{i,r}^t$. Nonetheless, we will use $x_{i,r}^t$ and $z_{i,r}^t$ in our theoretical analysis.

We also conceptually introduce $z_{i,0}^0$, $z_{i,0}^t$, and $z_{i,r+1}^t$ for $i \in [n]$ as follows:

$$z_{i,0}^0 := \text{prox}_{\eta f_i}(y_{i,0}^0), \quad z_{i,0}^t := z_{i,R}^{t-1}, \quad z_{i,r+1}^t := \begin{cases} \text{prox}_{\eta f_i}(y_{i,r+1}^t) & \text{if } i \in S_r^t, \\ z_{i,r}^t & \text{if } i \notin S_r^t. \end{cases} \tag{21}$$

# B PROOFS OF CONVERGENCE ANALYSIS FOR THE INEXACT VARIANT OF *Algorithm 1*

**Lemma 1.** *Let* $\{(x_{i,r}^t, y_{i,r}^t, z_{i,r}^t, \hat{x}_{i,r}^t)\}$ *be the sequence generated by Algorithm 1, starting from the initial state* $z_{i,0}^0 = \text{prox}_{\eta f_i}(y_{i,0}^0)$ *for all* $i \in [n]$. *Then, for all* $i \in [n]$, $r \geq 0$, *and* $t \geq 0$, *the following equalities hold:*

$$y_{i,r}^t = z_{i,r}^t + \eta \nabla f_i(z_{i,r}^t) \quad \text{and} \quad \hat{x}_{i,r}^t = 2x_{i,r}^t - y_{i,r}^t. \tag{22}$$

*Proof.* We prove (22) by induction on $t$.

**Base case** ($t = 0$):

For $r = 0$, due to the optimality condition of the proximal operator in (21), the initial state of $z_{i,0}^0$, and the initialization of *Algorithm 1*, we have both

$$y_{i,0}^0 = z_{i,0}^0 + \eta \nabla f_i(z_{i,0}^0) \quad \text{and} \quad \hat{x}_{i,0}^0 = 2x_{i,0}^0 - y_{i,0}^0,$$

which verifies (22) for $r = 0$.

Now suppose that (22) holds for $t = 0$ and $r \geq 0$, i.e.,

$$y_{i,r}^0 = z_{i,r}^0 + \eta \nabla f_i(z_{i,r}^0) \quad \text{and} \quad \hat{x}_{i,r}^0 = 2x_{i,r}^0 - y_{i,r}^0.$$

We will show that it also holds for $r + 1$. There are two cases:

- **Case 1:** If $i \in S_r^0$, then by the optimality condition of the proximal operator in equation 21, we have

$$\nabla f_i(z_{i,r+1}^0) + \frac{1}{\eta}(z_{i,r+1}^0 - y_{i,r+1}^0) = 0,$$

which implies

$$y_{i,r+1}^0 = z_{i,r+1}^0 + \eta \nabla f_i(z_{i,r+1}^0).$$

Moreover, from (15), we have

$$\hat{x}_{i,r+1}^0 = 2x_{i,r+1}^0 - y_{i,r+1}^0.$$

- **Case 2:** If $i \notin S_r^0$, then by (16) and the induction hypothesis, we have

$$y_{i,r+1}^0 = z_{i,r+1}^0 + \eta \nabla f_i(z_{i,r+1}^0) \quad \text{and} \quad \hat{x}_{i,r+1}^0 = 2x_{i,r+1}^0 - y_{i,r+1}^0.$$

In both cases, we conclude that (22) holds for all $i \in [n]$ when $t = 0$ and all $r \geq 0$.

**Inductive step:**

Now suppose that (22) holds for some $t \geq 0$ and all $r \geq 0$, i.e.,

$$y_{i,r}^t = z_{i,r}^t + \eta \nabla f_i(z_{i,r}^t) \quad \text{and} \quad \hat{x}_{i,r}^t = 2x_{i,r}^t - y_{i,r}^t.$$

We will show that it holds for $t + 1$.

From the inductive hypothesis (in particular, for $r = R$), we have

$$y_{i,R}^t = z_{i,R}^t + \eta \nabla f_i(z_{i,R}^t) \quad \text{and} \quad \hat{x}_{i,R}^t = 2x_{i,R}^t - y_{i,R}^t.$$

Thus, by (18), we have

$$y_{i,0}^{t+1} = z_{i,0}^{t+1} + \eta \nabla f_i(z_{i,0}^{t+1}) \quad \text{and} \quad \hat{x}_{i,0}^{t+1} = 2x_{i,0}^{t+1} - y_{i,0}^{t+1}.$$

Finally, applying the same argument as in the base case $t = 0$, we can extend the result to all $r \geq 0$ for $t + 1$. This completes the inductive proof. $\qquad \square$

**Lemma 2.** *Let* $\{(x_{i,r}^t, y_{i,r}^t, z_{i,r}^t, x_r^t)\}$ *be the sequence generated by Algorithm 1, and let* $\alpha > 0$. *Then, for all* $i \in S_t^T$, *we have:*

$$\|x_r^t - x_{i,r}^t\|^2 \leq \frac{(1 + \eta L)^2}{\alpha^2} \left[ 2\|x_{i,r}^t - x_{i,r+1}^t\|^2 + 4 \left( \|e_{i,r}^t\|^2 + \|e_{i,r+1}^t\|^2 \right) \right], \tag{23}$$

*Proof.* Since the update step of $y_{i,r+1}^t$ and Lemma 1 hold, for $i \in S_r^t$, we have

$$x_r^t - x_{i,r}^t = \frac{1}{\alpha}(y_{i,r+1}^t - y_{i,r}^t) = \frac{1}{\alpha}(z_{i,r+1}^t - z_{i,r}^t) + \frac{\eta}{\alpha}(\nabla f_i(z_{i,r+1}^t) - \nabla f_i(z_{i,r}^t)). \quad (24)$$

Therefore, we have:

$$
\begin{aligned}
\|x_r^t - x_{i,r}^t\|^2 &\overset{(24)}{=} \left\| \frac{1}{\alpha}(z_{i,r+1}^t - z_{i,r}^t) + \frac{\eta}{\alpha}(\nabla f_i(z_{i,r+1}^t) - \nabla f_i(z_{i,r}^t)) \right\|^2 \\
&\overset{(a)}{\leq} \left[ \left\| \frac{1}{\alpha}(z_{i,r+1}^t - z_{i,r}^t) \right\| + \left\| \frac{\eta}{\alpha}(\nabla f_i(z_{i,r+1}^t) - \nabla f_i(z_{i,r}^t)) \right\| \right]^2 \\
&\overset{(b)}{\leq} \left[ \frac{1}{\alpha}\|z_{i,r+1}^t - z_{i,r}^t\| + \frac{\eta L}{\alpha}\|z_{i,r+1}^t - z_{i,r}^t\| \right]^2 \\
&= \frac{(1+\eta L)^2}{\alpha^2}\|z_{i,r+1}^t - z_{i,r}^t\|^2 \\
&\overset{(20)}{=} \frac{(1+\eta L)^2}{\alpha^2}\|x_{i,r}^t - e_{i,r}^t - x_{i,r+1}^t + e_{i,r+1}^t\|^2 \\
&\overset{(c)}{\leq} \frac{(1+\eta L)^2}{\alpha^2} \left[ 2\|x_{i,r}^t - x_{i,r+1}^t\|^2 + 2\|e_{i,r}^t - e_{i,r+1}^t\|^2 \right] \\
&\overset{(c)}{\leq} \frac{(1+\eta L)^2}{\alpha^2} \left[ 2\|x_{i,r}^t - x_{i,r+1}^t\|^2 + 4(\|e_{i,r}^t\|^2 + \|e_{i,r+1}^t\|^2) \right],
\end{aligned}
\quad (25)
$$

where

(a) uses the triangle inequality: $\|a + b\| \leq \|a\| + \|b\|$,

(b) uses the $L$-smoothness of $f_i$: $\|\nabla f_i(x) - \nabla f_i(y)\| \leq L\|x - y\|$,

(c) uses the inequality: $\|a + b\|^2 \leq 2\|a\|^2 + 2\|b\|^2$.

$\square$

**Lemma 3.** *Let $\{(x_{i,r}^t, y_{i,r}^t, z_{i,r}^t, \hat{x}_{i,r}^t, x_r^t)\}$ be the sequence generated by Algorithm 1, and let $\alpha > 0$. Then, the following inequality holds:*

$$\|\nabla f(x_r^t)\|^2 \leq \frac{(1+\eta L)^2}{n\eta^2} \sum_{i=1}^n \left[ 2\|x_{i,r}^t - x_r^t\|^2 + 2\|e_{i,r}^t\|^2 \right], \quad (26)$$

*Proof.* From the aggregation step of *Algorithm 1* and Lemma 1, we have

$$x_r^t = \frac{1}{n}\sum_{i=1}^n \hat{x}_{i,r}^t = \frac{1}{n}\sum_{i=1}^n (2x_{i,r}^t - y_{i,r}^t) = \frac{1}{n}\sum_{i=1}^n (2x_{i,r}^t - z_{i,r}^t - \eta\nabla f_i(z_{i,r}^t)). \quad (27)$$

Therefore, we can write the gradient at the aggregated point as:

$$
\begin{aligned}
\nabla f(x_r^t) &= \frac{1}{n}\sum_{i=1}^n \nabla f_i(x_r^t) \\
&= \frac{1}{n}\sum_{i=1}^n \left[ \nabla f_i(x_r^t) - \nabla f_i(z_{i,r}^t) + \nabla f_i(z_{i,r}^t) \right] \\
&= \frac{1}{n}\sum_{i=1}^n \left[ \nabla f_i(x_r^t) - \nabla f_i(z_{i,r}^t) + \frac{1}{\eta}(2x_{i,r}^t - z_{i,r}^t - x_r^t) \right],
\end{aligned}
\quad (28)
$$

where the last equality uses the identity from (27).

Now we bound the squared norm of the gradient:

$$\|\nabla f(x_r^t)\|^2 \overset{(28)}{\leq} \frac{1}{n^2} \left\| \sum_{i=1}^n \left( \nabla f_i(x_r^t) - \nabla f_i(z_{i,r}^t) + \frac{1}{\eta}(2x_{i,r}^t - z_{i,r}^t - x_r^t) \right) \right\|^2$$

$$\overset{(a)}{\leq} \frac{1}{n^2\eta^2} \left[ \eta \sum_{i=1}^n \|\nabla f_i(x_r^t) - \nabla f_i(z_{i,r}^t)\| + \sum_{i=1}^n \|2x_{i,r}^t - z_{i,r}^t - x_r^t\| \right]^2$$

$$\overset{(b)}{\leq} \frac{1}{n\eta^2} \sum_{i=1}^n \left( \eta\|\nabla f_i(x_r^t) - \nabla f_i(z_{i,r}^t)\| + \|2x_{i,r}^t - z_{i,r}^t - x_r^t\| \right)^2$$

$$\overset{(c)}{\leq} \frac{1}{n\eta^2} \sum_{i=1}^n \left( \eta L\|x_r^t - z_{i,r}^t\| + \|2x_{i,r}^t - z_{i,r}^t - x_r^t\| \right)^2$$

$$\overset{(20)}{=} \frac{1}{n\eta^2} \sum_{i=1}^n \left( \eta L\|x_r^t - x_{i,r}^t + e_{i,r}^t\| + \|x_{i,r}^t + e_{i,r}^t - x_r^t\| \right)^2$$

$$\overset{(d)}{\leq} \frac{1}{n\eta^2} \sum_{i=1}^n \left[ (1+\eta L)\|x_r^t - x_{i,r}^t\| + (1+\eta L)\|e_{i,r}^t\| \right]^2$$

$$\overset{(e)}{\leq} \frac{(1+\eta L)^2}{n\eta^2} \sum_{i=1}^n \left[ 2\|x_r^t - x_{i,r}^t\|^2 + 2\|e_{i,r}^t\|^2 \right],$$

where

(a) uses the triangle inequality: $\|a+b\| \leq \|a\| + \|b\|$,

(b) uses Jensen's inequality: $\left(\sum_{i=1}^n A_i\right)^2 \leq n\sum_{i=1}^n A_i^2$,

(c) uses the $L$-smoothness of $f_i$,

(d) again applies the triangle inequality and groups terms.

(e) uses the inequality: $\|a+b\|^2 \leq 2\|a\|^2 + 2\|b\|^2$.

$\square$

**Definition 4.** *Define the potential function $\mathcal{D}_r^t$ for $t \geq 0$ and $0 \leq r \leq R-1$ as follows:*

$$\mathcal{D}_r^t := \frac{1}{n} \sum_{i=1}^n \left[ f_i(x_{i,r}^t) + \langle \nabla f_i(x_{i,r}^t), x_r^t - x_{i,r}^t \rangle + \frac{1}{2\eta}\|x_r^t - x_{i,r}^t\|^2 \right].$$

*For continuity of definition and notational convenience, we define $\mathcal{D}_R^t := \mathcal{D}_0^{t+1}$.*

**Lemma 4.** *Let $\{(x_{i,r}^t, y_{i,r}^t, z_{i,r}^t, \hat{x}_{i,r}^t, x_r^t)\}$ be the sequence generated by Algorithm 1, and let $\mathcal{D}_r^t$ be the potential function defined in Definition 4. Then, the following inequality holds:*

$$\mathcal{D}_{r+1}^t \leq \mathcal{D}_r^t - \mathcal{C}_0 \sum_{i \in S_r^t} \|x_{i,r+1}^t - x_{i,r}^t\|^2 + \frac{\mathcal{C}_1}{n} \sum_{i \in S_r^t} \|e_{i,r}^t\|^2 + \frac{\mathcal{C}_2}{n} \sum_{i \in S_r^t} \|e_{i,r+1}^t\|^2 + \frac{\mathcal{C}_3}{n} \sum_{i \notin S_r^t} \|e_{i,r}^t\|^2,$$

(29)

*where $\gamma > 0$ is a tunable coefficient, and the constants are defined as follows:*

$$\mathcal{C}_0 = \frac{2 - \alpha - \alpha\eta L - 2\eta^2 L^2 - 2\alpha\gamma(1+\eta L)^2}{2n\alpha\eta}, \quad \mathcal{C}_1 = \frac{(1+\eta L)^2}{\gamma\eta\alpha^2} + \frac{(1+\eta L)^2}{2\eta},$$

$$\mathcal{C}_2 = \frac{(1+\eta L)^2}{\gamma\eta\alpha^2}, \quad \mathcal{C}_3 = \frac{(1+\eta L)^2}{2\eta}.$$

*Proof.* Since Lemma 1 and equation (20) hold, we have

$$x_{i,r+1}^t + \eta\nabla f_i(x_{i,r+1}^t) = z_{i,r+1}^t + e_{i,r+1}^t + \eta\nabla f_i(x_{i,r+1}^t)$$
$$= y_{i,r+1}^t + e_{i,r+1}^t + \eta\left(\nabla f_i(x_{i,r+1}^t) - \nabla f_i(z_{i,r+1}^t)\right).$$

(30)

Therefore, we obtain

$$\mathcal{D}_{r+1}^t = \frac{1}{n}\sum_{i=1}^{n}\left[f_i(x_{i,r+1}^t) + \langle\nabla f_i(x_{i,r+1}^t), x_{r+1}^t - x_{i,r+1}^t\rangle + \frac{1}{2\eta}\|x_{i,r+1}^t - x_{r+1}^t\|^2\right]$$

$$= \frac{1}{n}\sum_{i=1}^{n}\left[f_i(x_{i,r+1}^t) + \langle\nabla f_i(x_{i,r+1}^t), x_r^t - x_{i,r+1}^t\rangle + \frac{1}{2\eta}\|x_{i,r+1}^t - x_r^t\|^2\right]$$

$$+ \frac{1}{n}\sum_{i=1}^{n}\left[\langle\nabla f_i(x_{i,r+1}^t), x_{r+1}^t - x_r^t\rangle + \frac{1}{2\eta}\left(\|x_{i,r+1}^t - x_{r+1}^t\|^2 - \|x_{i,r+1}^t - x_r^t\|^2\right)\right].$$

(31)

By combining (31) with (30), we get

$$\mathcal{D}_{r+1}^t \stackrel{(a)}{=} \frac{1}{n}\sum_{i=1}^{n}\left[f_i(x_{i,r+1}^t) + \langle\nabla f_i(x_{i,r+1}^t), x_r^t - x_{i,r+1}^t\rangle + \frac{1}{2\eta}\|x_{i,r+1}^t - x_r^t\|^2\right]$$

$$+ \frac{1}{n\eta}\sum_{i=1}^{n}\langle y_{i,r+1}^t + e_{i,r+1}^t + \eta\left(\nabla f_i(x_{i,r+1}^t) - \nabla f_i(z_{i,r+1}^t)\right) - x_{i,r+1}^t,\, x_{r+1}^t - x_r^t\rangle$$

$$+ \frac{1}{2n\eta}\sum_{i=1}^{n}\left[\|x_r^t - x_{r+1}^t\|^2 + 2\langle x_r^t - x_{i,r+1}^t, x_{r+1}^t - x_r^t\rangle\right]$$

$$\stackrel{(30)}{=} \frac{1}{n}\sum_{i=1}^{n}\left[f_i(x_{i,r+1}^t) + \langle\nabla f_i(x_{i,r+1}^t), x_r^t - x_{i,r+1}^t\rangle + \frac{1}{2\eta}\|x_{i,r+1}^t - x_r^t\|^2 + \frac{1}{2\eta}\|x_r^t - x_{r+1}^t\|^2\right]$$

$$+ \frac{1}{n\eta}\sum_{i=1}^{n}\langle y_{i,r+1}^t + e_{i,r+1}^t + \eta\left(\nabla f_i(x_{i,r+1}^t) - \nabla f_i(z_{i,r+1}^t)\right) + x_r^t - 2x_{i,r+1}^t,\, x_{r+1}^t - x_r^t\rangle$$

$$= \frac{1}{n}\sum_{i=1}^{n}\left[f_i(x_{i,r+1}^t) + \langle\nabla f_i(x_{i,r+1}^t), x_r^t - x_{i,r+1}^t\rangle + \frac{1}{2\eta}\|x_{i,r+1}^t - x_r^t\|^2\right]$$

$$+ \frac{1}{2\eta}\|x_r^t - x_{r+1}^t\|^2 + \frac{1}{n\eta}\sum_{i=1}^{n}\langle y_{i,r+1}^t + x_r^t - 2x_{i,r+1}^t,\, x_{r+1}^t - x_r^t\rangle$$

$$+ \frac{1}{n\eta}\sum_{i=1}^{n}\langle e_{i,r+1}^t + \eta(\nabla f_i(x_{i,r+1}^t) - \nabla f_i(z_{i,r+1}^t)),\, x_{r+1}^t - x_r^t\rangle$$

$$\stackrel{(15)}{=} \frac{1}{n}\sum_{i=1}^{n}\left[f_i(x_{i,r+1}^t) + \langle\nabla f_i(x_{i,r+1}^t), x_r^t - x_{i,r+1}^t\rangle + \frac{1}{2\eta}\|x_{i,r+1}^t - x_r^t\|^2\right]$$

$$+ \frac{1}{2\eta}\|x_r^t - x_{r+1}^t\|^2 + \frac{1}{n\eta}\sum_{i=1}^{n}\langle x_r^t - \hat{x}_{i,r+1}^t,\, x_{r+1}^t - x_r^t\rangle$$

$$+ \frac{1}{n\eta}\sum_{i=1}^{n}\langle e_{i,r+1}^t + \eta(\nabla f_i(x_{i,r+1}^t) - \nabla f_i(z_{i,r+1}^t)),\, x_{r+1}^t - x_r^t\rangle$$

$$\stackrel{(17)}{=} \frac{1}{n}\sum_{i=1}^{n}\left[f_i(x_{i,r+1}^t) + \langle\nabla f_i(x_{i,r+1}^t), x_r^t - x_{i,r+1}^t\rangle + \frac{1}{2\eta}\|x_{i,r+1}^t - x_r^t\|^2\right]$$

$$- \frac{1}{2\eta}\|x_r^t - x_{r+1}^t\|^2 + \frac{1}{n\eta}\sum_{i=1}^{n}\langle e_{i,r+1}^t + \eta(\nabla f_i(x_{i,r+1}^t) - \nabla f_i(z_{i,r+1}^t)),\, x_{r+1}^t - x_r^t\rangle,$$

(32)

where (a) uses the identity

$$\|x_{r+1}^t - x_{i,r+1}^t\|^2 - \|x_r^t - x_{i,r+1}^t\|^2 - \|x_r^t - x_{r+1}^t\|^2 = 2\langle x_r^t - x_{i,r+1}^t, x_{r+1}^t - x_r^t\rangle.$$

Notate

$$\sum_{i \in S_r^t} h_i = \sum_{i=1}^{n} h_i \, \mathbf{1}_{i \in S_r^t}, \quad \sum_{i \notin S_r^t} h_i = \sum_{i=1}^{n} h_i \, \mathbf{1}_{i \notin S_r^t},$$

where $\mathbf{1}_{\{\cdot\}}$ denotes the indicator function.

Moreover, we can bound the remaining inner product as

$$\frac{1}{n\eta} \sum_{i=1}^{n} \left\langle e_{i,r+1}^t + \eta \left( \nabla f_i(x_{i,r+1}^t) - \nabla f_i(z_{i,r+1}^t) \right), x_{r+1}^t - x_r^t \right\rangle$$

$$\overset{(a)}{\leq} \frac{1}{2n\eta} \sum_{i=1}^{n} \left[ \|x_{r+1}^t - x_r^t\|^2 + \left\| e_{i,r+1}^t + \eta \left( \nabla f_i(x_{i,r+1}^t) - \nabla f_i(z_{i,r+1}^t) \right) \right\|^2 \right]$$

$$\overset{(b)}{\leq} \frac{1}{2\eta} \|x_{r+1}^t - x_r^t\|^2 + \frac{1}{2n\eta} \sum_{i=1}^{n} \left( \|e_{i,r+1}^t\| + \eta \|\nabla f_i(x_{i,r+1}^t) - \nabla f_i(z_{i,r+1}^t)\| \right)^2 \qquad (33)$$

$$\overset{(c)}{\leq} \frac{1}{2\eta} \|x_{r+1}^t - x_r^t\|^2 + \frac{(1+\eta L)^2}{2n\eta} \sum_{i=1}^{n} \|e_{i,r+1}^t\|^2$$

$$= \frac{1}{2\eta} \|x_{r+1}^t - x_r^t\|^2 + \frac{(1+\eta L)^2}{2n\eta} \left( \sum_{i \in S_r^t} \|e_{i,r+1}^t\|^2 + \sum_{i \notin S_r^t} \|e_{i,r}^t\|^2 \right),$$

where

(a) uses $\langle a, b \rangle \leq \frac{1}{2}(\|a\|^2 + \|b\|^2)$,

(b) uses the triangle inequality $\|a + b\| \leq \|a\| + \|b\|$,

(c) follows from the $L$-smoothness of $f_i$ and the definition $z_{i,r+1}^t = x_{i,r+1}^t - e_{i,r+1}^t$.

Therefore, we have

$$\mathcal{D}_{r+1}^t \leq \frac{1}{n} \sum_{i=1}^{n} \left[ f_i(x_{i,r+1}^t) + \left\langle \nabla f_i(x_{i,r+1}^t), x_r^t - x_{i,r+1}^t \right\rangle + \frac{1}{2\eta} \|x_{i,r+1}^t - x_r^t\|^2 \right]$$

$$+ \frac{(1+\eta L)^2}{2n\eta} \left[ \sum_{i \in S_r^t} \|e_{i,r+1}^t\|^2 + \sum_{i \notin S_r^t} \|e_{i,r}^t\|^2 \right]$$

$$= \frac{1}{n} \sum_{i \in S_r^t} f_i(x_{i,r+1}^t) + \frac{1}{n} \sum_{i \in S_r^t} \left\langle \nabla f_i(x_{i,r+1}^t), x_{i,r}^t - x_{i,r+1}^t \right\rangle + \frac{1}{n} \sum_{i \in S_r^t} \left\langle \nabla f_i(x_{i,r+1}^t), x_r^t - x_{i,r}^t \right\rangle$$

$$+ \frac{1}{2n\eta} \sum_{i \in S_r^t} \|x_r^t - x_{i,r+1}^t\|^2 + \frac{1}{n} \sum_{i \notin S_r^t} f_i(x_{i,r}^t) + \frac{1}{n} \sum_{i \notin S_r^t} \left\langle \nabla f_i(x_{i,r}^t), x_r^t - x_{i,r}^t \right\rangle$$

$$+ \frac{1}{2n\eta} \sum_{i \notin S_r^t} \|x_r^t - x_{i,r}^t\|^2 + \frac{(1+\eta L)^2}{2n\eta} \left[ \sum_{i \in S_r^t} \|e_{i,r+1}^t\|^2 + \sum_{i \notin S_r^t} \|e_{i,r}^t\|^2 \right].$$

$$(34)$$

According to the $L$-smoothness property of $f_i$, we apply the inequality

$$f_i(x_{i,r+1}^t) + \left\langle \nabla f_i(x_{i,r+1}^t), x_{i,r}^t - x_{i,r+1}^t \right\rangle \leq f_i(x_{i,r}^t) + \frac{L}{2} \|x_{i,r}^t - x_{i,r+1}^t\|^2, \qquad (35)$$

and obtain

$$
\begin{aligned}
\mathcal{D}_{r+1}^t \leq{} & \frac{1}{n} \sum_{i \in S_r^t} f_i(x_{i,r}^t) + \frac{L}{2n} \sum_{i \in S_r^t} \|x_{i,r}^t - x_{i,r+1}^t\|^2 + \frac{1}{n} \sum_{i \in S_r^t} \langle \nabla f_i(x_{i,r+1}^t), x_r^t - x_{i,r}^t \rangle \\
& + \frac{1}{2n\eta} \sum_{i \in S_r^t} \|x_r^t - x_{i,r+1}^t\|^2 + \frac{1}{n} \sum_{i \notin S_r^t} f_i(x_{i,r}^t) + \frac{1}{n} \sum_{i \notin S_r^t} \langle \nabla f_i(x_{i,r}^t), x_r^t - x_{i,r}^t \rangle \\
& + \frac{1}{2n\eta} \sum_{i \notin S_r^t} \|x_r^t - x_{i,r}^t\|^2 + \frac{(1+\eta L)^2}{2n\eta} \left[ \sum_{i \in S_r^t} \|e_{i,r+1}^t\|^2 + \sum_{i \notin S_r^t} \|e_{i,r}^t\|^2 \right] \\
={} & \frac{1}{n} \sum_{i \in S_r^t} f_i(x_{i,r}^t) + \frac{1}{n} \sum_{i \in S_r^t} \langle \nabla f_i(x_{i,r}^t), x_r^t - x_{i,r}^t \rangle + \frac{1}{2n\eta} \sum_{i \in S_r^t} \|x_r^t - x_{i,r+1}^t\|^2 \quad (36) \\
& + \frac{L}{2n} \sum_{i \in S_r^t} \|x_{i,r}^t - x_{i,r+1}^t\|^2 + \frac{1}{n} \sum_{i \in S_r^t} \langle \nabla f_i(x_{i,r+1}^t) - \nabla f_i(x_{i,r}^t), x_r^t - x_{i,r}^t \rangle \\
& + \frac{1}{n} \sum_{i \notin S_r^t} f_i(x_{i,r}^t) + \frac{1}{n} \sum_{i \notin S_r^t} \langle \nabla f_i(x_{i,r}^t), x_r^t - x_{i,r}^t \rangle + \frac{1}{2n\eta} \sum_{i \notin S_r^t} \|x_r^t - x_{i,r}^t\|^2 \\
& + \frac{(1+\eta L)^2}{2n\eta} \left[ \sum_{i \in S_r^t} \|e_{i,r+1}^t\|^2 + \sum_{i \notin S_r^t} \|e_{i,r}^t\|^2 \right].
\end{aligned}
$$

Therefore, we have

$$
\begin{aligned}
\mathcal{D}_{r+1}^t \overset{(a)}{\leq}{} & \frac{1}{n} \sum_{i \in S_r^t} f_i(x_{i,r}^t) + \frac{1}{n} \sum_{i \in S_r^t} \langle \nabla f_i(x_{i,r}^t), x_r^t - x_{i,r}^t \rangle \\
& + \frac{1}{2n\eta} \sum_{i \in S_r^t} [\|x_r^t - x_{i,r}^t\|^2 + 2\langle x_r^t - x_{i,r}^t, x_{i,r}^t - x_{i,r+1}^t \rangle + \|x_{i,r}^t - x_{i,r+1}^t\|^2] \\
& + \frac{L}{2n} \sum_{i \in S_r^t} \|x_{i,r}^t - x_{i,r+1}^t\|^2 + \frac{1}{n} \sum_{i \in S_r^t} \langle \nabla f_i(x_{i,r+1}^t) - \nabla f_i(x_{i,r}^t), x_r^t - x_{i,r}^t \rangle \\
& + \frac{1}{n} \sum_{i \notin S_r^t} f_i(x_{i,r}^t) + \frac{1}{n} \sum_{i \notin S_r^t} \langle \nabla f_i(x_{i,r}^t), x_r^t - x_{i,r}^t \rangle + \frac{1}{2n\eta} \sum_{i \notin S_r^t} \|x_r^t - x_{i,r}^t\|^2 \\
& + \frac{(1+\eta L)^2}{2n\eta} [\sum_{i \in S_r^t} \|e_{i,r+1}^t\|^2 + \sum_{i \notin S_r^t} \|e_{i,r}^t\|^2] \\
={} & \frac{1}{n} \sum_{i \in S_r^t} [f_i(x_{i,r}^t) + \langle \nabla f_i(x_{i,r}^t), x_r^t - x_{i,r}^t \rangle + \frac{1}{2\eta} \|x_r^t - x_{i,r}^t\|^2] \\
& + \frac{1}{n} \sum_{i \notin S_r^t} [f_i(x_{i,r}^t) + \langle \nabla f_i(x_{i,r}^t), x_r^t - x_{i,r}^t \rangle + \frac{1}{2\eta} \|x_r^t - x_{i,r}^t\|^2] \\
& + \frac{1}{n\eta} \sum_{i \in S_r^t} \langle x_r^t - x_{i,r}^t, x_{i,r}^t - x_{i,r+1}^t \rangle + \frac{\eta L + 1}{2n\eta} \sum_{i \in S_r^t} \|x_{i,r}^t - x_{i,r+1}^t\|^2 \\
& + \frac{1}{n} \sum_{i \in S_r^t} \langle \nabla f_i(x_{i,r+1}^t) - \nabla f_i(x_{i,r}^t), x_r^t - x_{i,r}^t \rangle + \frac{(1+\eta L)^2}{2n\eta} [\sum_{i \in S_r^t} \|e_{i,r+1}^t\|^2 + \sum_{i \notin S_r^t} \|e_{i,r}^t\|^2] \\
={} & \mathcal{D}_r^t + \frac{1}{n\eta} \sum_{i \in S_r^t} \langle x_r^t - x_{i,r}^t, x_{i,r}^t - x_{i,r+1}^t \rangle + \frac{\eta L + 1}{2n\eta} \sum_{i \in S_r^t} \|x_{i,r}^t - x_{i,r+1}^t\|^2 \\
& + \frac{1}{n} \sum_{i \in S_r^t} \langle \nabla f_i(x_{i,r+1}^t) - \nabla f_i(x_{i,r}^t), x_r^t - x_{i,r}^t \rangle + \frac{(1+\eta L)^2}{2n\eta} [\sum_{i \in S_r^t} \|e_{i,r+1}^t\|^2 + \sum_{i \notin S_r^t} \|e_{i,r}^t\|^2],
\end{aligned}
$$

$$(37)$$

where (a) uses the identity

$$\|x_r^t - x_{i,r+1}^t\|^2 = \|x_r^t - x_{i,r}^t\|^2 + 2\left\langle x_r^t - x_{i,r}^t,\, x_{i,r}^t - x_{i,r+1}^t\right\rangle + \|x_{i,r}^t - x_{i,r+1}^t\|^2.$$

Since we have

$$
\begin{aligned}
x_r^t - x_{i,r}^t &= \frac{1}{\alpha}(y_{i,r+1}^t - y_{i,r}^t) \\
&\overset{(22)}{=} \frac{1}{\alpha}\left(z_{i,r+1}^t - z_{i,r}^t + \eta\nabla f_i(z_{i,r+1}^t) - \eta\nabla f_i(z_{i,r}^t)\right) \\
&\overset{(20)}{=} \frac{1}{\alpha}(x_{i,r+1}^t - e_{i,r+1}^t - x_{i,r}^t + e_{i,r}^t) + \frac{\eta}{\alpha}\left(\nabla f_i(x_{i,r+1}^t) - \nabla f_i(x_{i,r}^t)\right) \\
&\quad + \frac{\eta}{\alpha}\left[\nabla f_i(z_{i,r+1}^t) - \nabla f_i(x_{i,r+1}^t) - \left(\nabla f_i(z_{i,r}^t) - \nabla f_i(x_{i,r}^t)\right)\right] \\
&= \frac{1}{\alpha}(x_{i,r+1}^t - x_{i,r}^t) + \frac{\eta}{\alpha}(\nabla f_i(x_{i,r+1}^t) - \nabla f_i(x_{i,r}^t)) \\
&\quad + \frac{1}{\alpha}(e_{i,r}^t - e_{i,r+1}^t) + \frac{\eta}{\alpha}\left[\nabla f_i(z_{i,r+1}^t) - \nabla f_i(x_{i,r+1}^t) - \nabla f_i(z_{i,r}^t) + \nabla f_i(x_{i,r}^t)\right],
\end{aligned}
$$
(38)

we obtain

$$
\begin{aligned}
\mathcal{D}_{r+1}^t \overset{(38)}{\le}\;& \mathcal{D}_r^t + \frac{\eta L + 1}{2n\eta}\sum_{i\in S_r^t}\|x_{i,r}^t - x_{i,r+1}^t\|^2 - \frac{1}{n\eta\alpha}\sum_{i\in S_r^t}\|x_{i,r+1}^t - x_{i,r}^t\|^2 \\
&+ \frac{1}{n\alpha}\sum_{i\in S_r^t}\left\langle\nabla f_i(x_{i,r+1}^t) - \nabla f_i(x_{i,r}^t),\, x_{i,r}^t - x_{i,r+1}^t\right\rangle \\
&+ \frac{1}{n\alpha}\sum_{i\in S_r^t}\left\langle\nabla f_i(x_{i,r+1}^t) - \nabla f_i(x_{i,r}^t),\, x_{i,r+1}^t - x_{i,r}^t\right\rangle \\
&+ \frac{1}{n\eta}\sum_{i\in S_r^t}\left\langle\omega_{i,r}^t,\, x_{i,r}^t - x_{i,r+1}^t\right\rangle + \frac{1}{n}\sum_{i\in S_r^t}\left\langle\omega_{i,r}^t,\, \nabla f_i(x_{i,r+1}^t) - \nabla f_i(x_{i,r}^t)\right\rangle \\
&+ \frac{\eta}{n\alpha}\sum_{i\in S_r^t}\|\nabla f_i(x_{i,r+1}^t) - \nabla f_i(x_{i,r}^t)\|^2 + \frac{(1+\eta L)^2}{2n\eta}\left[\sum_{i\in S_r^t}\|e_{i,r+1}^t\|^2 + \sum_{i\notin S_r^t}\|e_{i,r}^t\|^2\right],
\end{aligned}
$$
(39)

where we define

$$\omega_{i,r}^t := \frac{1}{\alpha}(e_{i,r}^t - e_{i,r+1}^t) + \frac{\eta}{\alpha}\left[\nabla f_i(z_{i,r+1}^t) - \nabla f_i(x_{i,r+1}^t) - \nabla f_i(z_{i,r}^t) + \nabla f_i(x_{i,r}^t)\right]. \quad (40)$$

Finally, we have

$$\mathcal{D}_{r+1}^t \overset{(40)}{\leq} \mathcal{D}_r^t - \frac{2 - \alpha - \alpha\eta L - 2\eta^2 L^2}{2n\alpha\eta} \sum_{i \in S_r^t} \|x_{i,r+1}^t - x_{i,r}^t\|^2$$

$$+ \frac{(1+\eta L)^2}{2n\eta} \left[ \sum_{i \in S_r^t} \|e_{i,r+1}^t\|^2 + \sum_{i \notin S_r^t} \|e_{i,r}^t\|^2 \right]$$

$$+ \frac{1}{n\eta} \sum_{i \in S_r^t} \langle \omega_{i,r}^t, \, x_{i,r}^t - x_{i,r+1}^t + \eta(\nabla f_i(x_{i,r+1}^t) - \nabla f_i(x_{i,r}^t)) \rangle$$

$$\overset{(a)}{\leq} \mathcal{D}_r^t - \frac{2 - \alpha - \alpha\eta L - 2\eta^2 L^2}{2n\alpha\eta} \sum_{i \in S_r^t} \|x_{i,r+1}^t - x_{i,r}^t\|^2$$

$$+ \frac{(1+\eta L)^2}{2n\eta} \left[ \sum_{i \in S_r^t} \|e_{i,r+1}^t\|^2 + \sum_{i \notin S_r^t} \|e_{i,r}^t\|^2 \right]$$

$$+ \frac{1}{2n\eta} \sum_{i \in S_r^t} \left( \frac{1}{\gamma} \|\omega_{i,r}^t\|^2 + \gamma \|x_{i,r}^t - x_{i,r+1}^t + \eta(\nabla f_i(x_{i,r+1}^t) - \nabla f_i(x_{i,r}^t))\|^2 \right) \quad (41)$$

$$\overset{(b)}{\leq} \mathcal{D}_r^t - \frac{2 - \alpha - \alpha\eta L - 2\eta^2 L^2}{2n\alpha\eta} \sum_{i \in S_r^t} \|x_{i,r+1}^t - x_{i,r}^t\|^2$$

$$+ \frac{(1+\eta L)^2}{2n\eta} \left[ \sum_{i \in S_r^t} \|e_{i,r+1}^t\|^2 + \sum_{i \notin S_r^t} \|e_{i,r}^t\|^2 \right]$$

$$+ \frac{\gamma(1+\eta L)^2}{2n\eta} \sum_{i \in S_r^t} \|x_{i,r}^t - x_{i,r+1}^t\|^2 + \frac{1}{2n\eta\gamma} \sum_{i \in S_r^t} \|\omega_{i,r}^t\|^2$$

$$\leq \mathcal{D}_r^t - \frac{2 - \alpha - \alpha\eta L - 2\eta^2 L^2 - 2\alpha\gamma(1+\eta L)^2}{2n\alpha\eta} \sum_{i \in S_r^t} \|x_{i,r+1}^t - x_{i,r}^t\|^2$$

$$+ \frac{(1+\eta L)^2}{2n\eta} \left[ \sum_{i \in S_r^t} \|e_{i,r+1}^t\|^2 + \sum_{i \notin S_r^t} \|e_{i,r}^t\|^2 \right] + \frac{1}{2n\eta\gamma} \sum_{i \in S_r^t} \|\omega_{i,r}^t\|^2,$$

where

(a) uses the inequality $\langle a, b \rangle \leq \frac{1}{2} \left( \gamma \|a\|^2 + \frac{1}{\gamma} \|b\|^2 \right)$,

(b) uses the $L$-smoothness of $f_i$ and the inequality $\|a + b\| \leq \|a\| + \|b\|$.

Moreover, we can bound the remaining norm as

$$\sum_{i \in S_r^t} \|\omega_{i,r}^t\|^2 = \sum_{i \in S_r^t} \left\| \frac{1}{\alpha}(e_{i,r}^t - e_{i,r+1}^t) + \frac{\eta}{\alpha} \left[ \nabla f_i(z_{i,r+1}^t) - \nabla f_i(x_{i,r+1}^t) - \nabla f_i(z_{i,r}^t) + \nabla f_i(x_{i,r}^t) \right] \right\|^2$$

$$\leq \sum_{i \in S_r^t} \left[ \frac{1}{\alpha}\|e_{i,r}^t\| + \frac{1}{\alpha}\|e_{i,r+1}^t\| + \frac{\eta}{\alpha}\|\nabla f_i(z_{i,r+1}^t) - \nabla f_i(x_{i,r+1}^t)\| + \frac{\eta}{\alpha}\|\nabla f_i(z_{i,r}^t) - \nabla f_i(x_{i,r}^t)\| \right]^2$$

$$\overset{(a)}{\leq} \sum_{i \in S_r^t} \frac{1}{\alpha^2} \left[ (1+\eta L)\left(\|e_{i,r}^t\| + \|e_{i,r+1}^t\|\right) \right]^2$$

$$\overset{(b)}{\leq} \sum_{i \in S_r^t} \frac{2(1+\eta L)^2}{\alpha^2} \left( \|e_{i,r}^t\|^2 + \|e_{i,r+1}^t\|^2 \right),$$

$$(42)$$

where

    (a) uses the $L$-smoothness of $f_i$.

    (b) uses the inequality: $\|a + b\|^2 \leq 2\|a\|^2 + 2\|b\|^2$.

Therefore, we conclude

$$\mathcal{D}_{r+1}^t \leq \mathcal{D}_r^t - \frac{2 - \alpha - \alpha\eta L - 2\eta^2 L^2 - 2\alpha\gamma(1 + \eta L)^2}{2n\alpha\eta} \sum_{i \in S_r^t} \|x_{i,r+1}^t - x_{i,r}^t\|^2$$

$$+ \frac{(1 + \eta L)^2}{2n\eta} \left[ \sum_{i \in S_r^t} \|e_{i,r+1}^t\|^2 + \sum_{i \notin S_r^t} \|e_{i,r}^t\|^2 \right] + \frac{(1 + \eta L)^2}{n\eta\gamma\alpha^2} \sum_{i \in S_r^t} \left( \|e_{i,r}^t\|^2 + \|e_{i,r+1}^t\|^2 \right)$$

$$= \mathcal{D}_r^t - \mathcal{C}_0 \sum_{i \in S_r^t} \|x_{i,r+1}^t - x_{i,r}^t\|^2 + \frac{\mathcal{C}_1}{n} \sum_{i \in S_r^t} \|e_{i,r}^t\|^2 + \frac{\mathcal{C}_2}{n} \sum_{i \in S_r^t} \|e_{i,r+1}^t\|^2 + \frac{\mathcal{C}_3}{n} \sum_{i \notin S_r^t} \|e_{i,r}^t\|^2.$$

$$\tag{43}$$

$\square$

**Lemma 5.** *Let $\{(x_{i,r}^t, y_{i,r}^t, z_{i,r}^t, \hat{x}_{i,r}^t, x_r^t)\}$ be the sequence generated by Algorithm 1, and let $\mathcal{D}_r^t$ be the potential function defined in Definition 4. Then the following inequality holds:*

$$\mathbb{E}_{(S_0^t, \dots, S_r^t)}[\mathcal{D}_{r+1}^t] \leq \mathbb{E}_{(S_0^t, \dots, S_{r-1}^t)}[\mathcal{D}_r^t] - \frac{C\mathcal{C}_0\alpha^2}{2n(1 + \eta L)^2} \sum_{i=1}^n \|x_r^t - x_{i,r}^t\|^2$$

$$+ \frac{C(\mathcal{C}_1 + 2n\mathcal{C}_0) + (n - C)\mathcal{C}_3}{n^2} \sum_{i=1}^n \|e_{i,r}^t\|^2 + \frac{C(\mathcal{C}_2 + 2n\mathcal{C}_0)}{n^2} \sum_{i=1}^n \|e_{i,r+1}^t\|^2.$$

$$\tag{44}$$

*Proof.* According to Lemma 2 and Lemma 4, we have

$$\mathcal{D}_{r+1}^t \leq \mathcal{D}_r^t - \mathcal{C}_0 \sum_{i \in S_r^t} \|x_{i,r+1}^t - x_{i,r}^t\|^2 + \frac{\mathcal{C}_1}{n} \sum_{i \in S_r^t} \|e_{i,r}^t\|^2 + \frac{\mathcal{C}_2}{n} \sum_{i \in S_r^t} \|e_{i,r+1}^t\|^2 + \frac{\mathcal{C}_3}{n} \sum_{i \notin S_r^t} \|e_{i,r}^t\|^2$$

$$\leq \mathcal{D}_r^t - \mathcal{C}_0 \sum_{i \in S_r^t} \left[ \frac{\alpha^2}{2(1 + \eta L)^2} \|x_r^t - x_{i,r}^t\|^2 - 2\|e_{i,r}^t\|^2 - 2\|e_{i,r+1}^t\|^2 \right]$$

$$+ \frac{\mathcal{C}_1}{n} \sum_{i \in S_r^t} \|e_{i,r}^t\|^2 + \frac{\mathcal{C}_2}{n} \sum_{i \in S_r^t} \|e_{i,r+1}^t\|^2 + \frac{\mathcal{C}_3}{n} \sum_{i \notin S_r^t} \|e_{i,r}^t\|^2$$

$$= \mathcal{D}_r^t - \frac{\mathcal{C}_0\alpha^2}{2(1 + \eta L)^2} \sum_{i \in S_r^t} \|x_r^t - x_{i,r}^t\|^2 + \frac{\mathcal{C}_1 + 2n\mathcal{C}_0}{n} \sum_{i \in S_r^t} \|e_{i,r}^t\|^2$$

$$+ \frac{\mathcal{C}_2 + 2n\mathcal{C}_0}{n} \sum_{i \in S_r^t} \|e_{i,r+1}^t\|^2 + \frac{\mathcal{C}_3}{n} \sum_{i \notin S_r^t} \|e_{i,r}^t\|^2.$$

$$\tag{45}$$

Before taking expectations, we clarify the source of randomness in our algorithm and the random variables over which the expectation is taken. The only randomness arises from the random permutation of clients formed at the beginning of each meta-epoch. Accordingly, expectations are taken with respect to this permutation (or its prefix).

Let $S^t = (S_0^t, \dots, S_{R-1}^t)$ denote the without-replacement partition in meta-epoch $t$, where $S_i^t$ is the $i$-th batch. Then $\mathbb{E}_{S_0^t, \dots, S_r^t}[\cdot]$ denotes expectation over the random choice of the first $r+1$ batches (i.e., first the 0-th batch, then the 1-st, ..., up to the $r$-th). Likewise, $\mathbb{E}_{S_r^t}[\cdot \mid S_0^t, \dots, S_{r-1}^t]$ is the conditional expectation over the $r$-th batch given the previous $r$ batches. Note that batches are

sets (no order within a batch), but the sequence of batches $(S_0^t, \ldots, S_{R-1}^t)$ is ordered. For example, if $n = 4$, $C = 2$, all the possible partition: $((1,2),(3,4)),((3,4),(1,2)),((1,3),(2,4)),((2,4),(1,3)),$ $((1,4),(2,3)),((2,3),(1,4))$.

Taking the expectation over $(S_0^t, \ldots, S_r^t)$ yields

$$
\begin{aligned}
\mathbb{E}_{(S_0^t,\ldots,S_r^t)}[\mathcal{D}_{r+1}^t] \leq \ &\mathbb{E}_{(S_0^t,\ldots,S_r^t)}[\mathcal{D}_r^t] - \frac{\mathcal{C}_0 \alpha^2}{2(1+\eta L)^2} \mathbb{E}_{(S_0^t,\ldots,S_r^t)}\left[\sum_{i \in S_r^t} \|x_r^t - x_{i,r}^t\|^2\right] \\
&+ \frac{\mathcal{C}_1 + 2n\mathcal{C}_0}{n} \mathbb{E}_{(S_0^t,\ldots,S_r^t)}\left[\sum_{i \in S_r^t} \|e_{i,r}^t\|^2\right] \\
&+ \frac{\mathcal{C}_2 + 2n\mathcal{C}_0}{n} \mathbb{E}_{(S_0^t,\ldots,S_r^t)}\left[\sum_{i \in S_r^t} \|e_{i,r+1}^t\|^2\right] \\
&+ \frac{\mathcal{C}_3}{n} \mathbb{E}_{(S_0^t,\ldots,S_r^t)}\left[\sum_{i \notin S_r^t} \|e_{i,r}^t\|^2\right].
\end{aligned}
\tag{46}
$$

Since $\mathcal{D}_r^t$ depends only on $(S_0^t, \ldots, S_{r-1}^t)$, it follows that

$$
\mathbb{E}_{(S_0^t,\ldots,S_r^t)}[\mathcal{D}_r^t] = \mathbb{E}_{(S_0^t,\ldots,S_{r-1}^t)}[\mathcal{D}_r^t].
\tag{47}
$$

We first focus on the uniform term $\mathbb{E}_{(S_0^t,\ldots,S_r^t)}\left[\sum_{i \in S_r^t} \mathcal{F}_{i,r}^t\right]$. Using the law of total expectation conditioned on $(S_0^t, \ldots, S_{r-1}^t)$, we have

$$
\begin{aligned}
\mathbb{E}_{(S_0^t,\ldots,S_r^t)}\left[\sum_{i \in S_r^t} \mathcal{F}_{i,r}^t\right] &= \mathbb{E}_{(S_0^t,\ldots,S_{r-1}^t)}\left\{\mathbb{E}_{S_r^t}\left[\sum_{i \in S_r^t} \mathcal{F}_{i,r}^t \,\middle|\, S_0^t, \ldots, S_{r-1}^t\right]\right\} \\
&\overset{(a)}{=} \mathbb{E}_{(S_0^t,\ldots,S_{r-1}^t)}\left[\frac{C}{n - rC} \sum_{i \notin (S_0^t,\ldots,S_{r-1}^t)} \mathcal{F}_{i,r}^t\right] \\
&\overset{(b)}{=} \frac{(C!)^r (n - rC)!}{n!} \sum_{(S_0^t,\ldots,S_{r-1}^t)}\left[\frac{C}{n - rC} \sum_{i \notin (S_0^t,\ldots,S_{r-1}^t)} \mathcal{F}_{i,r}^t\right] \\
&= \frac{(C!)^r (n - rC)! \cdot C}{n!(n - rC)} \sum_{(S_0^t,\ldots,S_{r-1}^t)} \sum_{i \notin (S_0^t,\ldots,S_{r-1}^t)} \mathcal{F}_{i,r}^t,
\end{aligned}
\tag{48}
$$

where (a) follows from the first equation in Lemma 1 of Malinovsky et al. (2023b), and (b) is obtained using combinatorial counting.

Equation (48) is not straightforward to interpret. We first clarify the meaning of $\sum_{(S_0^t,\ldots,S_{r-1}^t)}$ and $\sum_{i \notin (S_0^t,\ldots,S_{r-1}^t)}$. Then, we provide the detailed proofs of steps (a) and (b).

First, we define

$$
\sum_{i \notin (S_0^t,\ldots,S_{r-1}^t)} f_i := \sum_{i=1}^{n} f_i \mathbf{1}_{i \notin (S_0^t,\ldots,S_{r-1}^t)},
$$

where $\mathbf{1}_{\{\cdot\}}$ is the indicator function.

$\sum_{(S_0^t,\ldots,S_{r-1}^t)}$ denotes the summation over all ordered $r$-tuples of disjoint batches of size $C$ (drawn without replacement from $\{1, \ldots, n\}$).

Second, we demonstrate the detailed proof of (a) as follows:

*Proof.* (a) is applied to the conditional expectation

$$\mathbb{E}_{S_r^t}\left[\sum_{i \in S_r^t} \mathcal{F}_{i,r}^t \;\middle|\; S_0^t, \ldots, S_{r-1}^t\right].$$

In meta-epoch $t$, $(S_0^t, \ldots, S_{R-1}^t)$ is a partition of $\{1, \ldots, n\}$ into batches of size $C$. We generate it in two steps: first choose $(S_0^t, \ldots, S_{r-1}^t)$, then uniformly permute the remaining clients and remove the replicate inside the batches to obtain $(S_r^t, \ldots, S_{R-1}^t)$. Conditioning on $S_0^t, \ldots, S_{r-1}^t$ fixes the first step, so $S_r^t$ is a uniformly random size-$C$ subset of $U_r^t := \{1, \ldots, n\} \setminus \bigcup_{q=0}^{r-1} S_q^t$ with $|U_r^t| = n - rC$. By (a) or symmetry,

$$\mathbb{E}_{S_r^t}\left[\sum_{i \in S_r^t} \mathcal{F}_{i,r}^t \;\middle|\; S_0^t, \ldots, S_{r-1}^t\right] = \frac{C}{n - rC} \sum_{i \in U_r^t} \mathcal{F}_{i,r}^t,$$

i.e., a $C$-term sum taken uniformly from a permutation of length $n - rC$. $\qquad\square$

Third, we demonstrate the detailed proof of (b) as follows:

*Proof.* We compute $\mathbb{E}_{(S_0^t, \ldots, S_{r-1}^t)}$ by averaging over all ordered $r$-tuples of disjoint batches of size $C$ (drawn without replacement from $\{1, \ldots, n\}$). The count is

$$N_{\text{part}} = \frac{n!}{(n - rC)! \, (C!)^r}.$$

Indeed, first choose an ordered length-$rC$ sequence: $n!/(n - rC)!$ choices; then quotient out the $C!$ internal permutations within each batch (no $r!$ factor, since batches are ordered). Hence the averaging coefficient is

$$\frac{1}{N_{\text{part}}} = \frac{(n - rC)! \, (C!)^r}{n!}.$$

(Assumes $0 \le rC \le n$.) $\qquad\square$

To simplify the summation, fix an index $i = k$. The term $\mathcal{F}_{k,r}^t$ appears in the summation if and only if $k \notin (S_0^t, \ldots, S_{r-1}^t)$. In other words, we need to count the number of permutations of size $rC$ drawn from $\{0, 1, \ldots, k-1, k+1, \ldots, n-1\}$ (excluding $k$), divided by the repetition factor due to internal ordering in each $S_\ell^t$. The number of such permutations is

$$\frac{(n-1)!}{(n - rC - 1)!}, \quad \text{and the repetition due to internal client permutations is } (C!)^r.$$

Thus, the number of times $\mathcal{F}_{k,r}^t$ appears is $\frac{(n-1)!}{(n-rC-1)!(C!)^r}$. Therefore,

$$\sum_{(S_0^t, \ldots, S_{r-1}^t)} \sum_{i \notin (S_0^t, \ldots, S_{r-1}^t)} \mathcal{F}_{i,r}^t = \frac{(n-1)!}{(n - rC - 1)!(C!)^r} \sum_{i=1}^{n} \left[\mathcal{F}_{i,r}^t\right]. \tag{49}$$

Therefore, we have

$$\mathbb{E}_{(S_0^t, \ldots, S_r^t)}\left[\sum_{i \in S_r^t} \mathcal{F}_{i,r}^t\right] = \frac{(C!)^r (n - rC)! \cdot C}{n!(n - rC)} \sum_{i=1}^{n} \left[\frac{(n-1)!}{(n - rC - 1)!(C!)^r} \mathcal{F}_{i,r}^t\right]$$

$$= \frac{C}{n} \sum_{i=1}^{n} \mathcal{F}_{i,r}^t. \tag{50}$$

As a consequence, the following expectations hold:

$$\mathbb{E}_{(S_0^t, \ldots, S_r^t)}\left[\sum_{i \in S_r^t} \|x_r^t - x_{i,r}^t\|^2\right] = \frac{C}{n} \sum_{i=1}^{n} \|x_r^t - x_{i,r}^t\|^2,$$

$$\mathbb{E}_{(S_0^t,...,S_r^t)} \left[ \sum_{i \in S_r^t} \|e_{i,r}^t\|^2 \right] = \frac{C}{n} \sum_{i=1}^n \|e_{i,r}^t\|^2,$$

$$\mathbb{E}_{(S_0^t,...,S_r^t)} \left[ \sum_{i \in S_r^t} \|e_{i,r+1}^t\|^2 \right] = \frac{C}{n} \sum_{i=1}^n \|e_{i,r+1}^t\|^2,$$

$$\mathbb{E}_{(S_0^t,...,S_r^t)} \left[ \sum_{i \notin S_r^t} \|e_{i,r}^t\|^2 \right] = \sum_{i=1}^n \|e_{i,r}^t\|^2 - \mathbb{E}_{(S_0^t,...,S_r^t)} \left[ \sum_{i \in S_r^t} \|e_{i,r}^t\|^2 \right] = \frac{n-C}{n} \sum_{i=1}^n \|e_{i,r}^t\|^2.$$

Therefore, combining all terms, we obtain the expected inequality:

$$\mathbb{E}_{(S_0^t,...,S_r^t)}[\mathcal{D}_{r+1}^t] \le \mathbb{E}_{(S_0^t,...,S_{r-1}^t)}[\mathcal{D}_r^t] - \frac{C\mathcal{C}_0 \alpha^2}{2n(1+\eta L)^2} \sum_{i=1}^n \|x_r^t - x_{i,r}^t\|^2$$

$$+ \frac{C(\mathcal{C}_1 + 2n\mathcal{C}_0) + (n-C)\mathcal{C}_3}{n^2} \sum_{i=1}^n \|e_{i,r}^t\|^2 + \frac{C(\mathcal{C}_2 + 2n\mathcal{C}_0)}{n^2} \sum_{i=1}^n \|e_{i,r+1}^t\|^2.$$

$\square$

**Lemma 6.** *Let $\{(x_{i,r}^t, y_{i,r}^t, z_{i,r}^t, \hat{x}_{i,r}^t, x_r^t)\}$ be the sequence generated by Algorithm 1. Let $\mathcal{D}_r^t$ be the potential function defined in Definition 4. Then, we have:*

$$\frac{1}{\mathcal{A}_1 R} \mathbb{E}_{(S^0,S^1,...,S^{t-1})} \left[ \sum_{r=0}^{R-1} \|\nabla f(x_r^t)\|^2 \right] \le \mathbb{E}_{(S^0,S^1,...,S^{t-1})}[\mathcal{D}_0^t] - \mathbb{E}_{(S^0,S^1,...,S^t)}[\mathcal{D}_0^{t+1}]$$

$$+ \frac{\mathcal{B}_1}{n} \sum_{r=0}^{R-1} \sum_{i=1}^n \|e_{i,r}^t\|^2 + \frac{\mathcal{B}_2}{n} \sum_{r=0}^{R-1} \sum_{i=1}^n \|e_{i,r+1}^t\|^2. \tag{51}$$

*The constants are defined as:*

$$\mathcal{A}_1 := \frac{4(1+\eta L)^4}{\eta^2 \mathcal{C}_0 \alpha^2 n}, \quad \mathcal{B}_1 := \frac{C(\mathcal{C}_1 + 2n\mathcal{C}_0) + (n-C)\mathcal{C}_3}{n} + \frac{C\alpha^2 \mathcal{C}_0}{2(1+\eta L)^2}, \quad \mathcal{B}_2 := \frac{C(\mathcal{C}_2 + 2n\mathcal{C}_0)}{n}.$$

*Proof.* Using Lemma 3 and Lemma 5, we have

$$\frac{\mathcal{C}_0 \alpha^2 C}{2n(1+\eta L)^2} \left[ \frac{n\eta^2}{2(1+\eta L)^2} \|\nabla f(x_r^t)\|^2 - \sum_{i=1}^n \|e_{i,r}^t\|^2 \right] \le \mathbb{E}_{(S_0^t,...,S_{r-1}^t)}[\mathcal{D}_r^t] - \mathbb{E}_{(S_0^t,...,S_r^t)}[\mathcal{D}_{r+1}^t]$$

$$+ \frac{C(\mathcal{C}_1 + 2n\mathcal{C}_0) + (n-C)\mathcal{C}_3}{n^2} \sum_{i=1}^n \|e_{i,r}^t\|^2$$

$$+ \frac{C(\mathcal{C}_2 + 2n\mathcal{C}_0)}{n^2} \sum_{i=1}^n \|e_{i,r+1}^t\|^2. \tag{52}$$

Rewriting the inequality gives

$$\frac{\eta^2 \mathcal{C}_0 \alpha^2 C}{4(1+\eta L)^4} \|\nabla f(x_r^t)\|^2 \le \mathbb{E}_{(S_0^t,...,S_{r-1}^t)}[\mathcal{D}_r^t] - \mathbb{E}_{(S_0^t,...,S_r^t)}[\mathcal{D}_{r+1}^t]$$

$$+ \left( \frac{C(\mathcal{C}_1 + 2n\mathcal{C}_0) + (n-C)\mathcal{C}_3}{n^2} + \frac{C\alpha^2 \mathcal{C}_0}{2n(1+\eta L)^2} \right) \sum_{i=1}^n \|e_{i,r}^t\|^2 \quad (53)$$

$$+ \frac{C(\mathcal{C}_2 + 2n\mathcal{C}_0)}{n^2} \sum_{i=1}^n \|e_{i,r+1}^t\|^2.$$

Summing (53) over $r = 0$ to $R - 1$, we get

$$\frac{\eta^2 \mathcal{C}_0 \alpha^2 C}{4(1 + \eta L)^4} \sum_{r=0}^{R-1} \|\nabla f(x_r^t)\|^2 \leq \mathcal{D}_0^t - \mathbb{E}_{(S_0^t, \ldots, S_{R-1}^t)}[\mathcal{D}_R^t]$$

$$+ \left( \frac{C(\mathcal{C}_1 + 2n\mathcal{C}_0) + (n - C)\mathcal{C}_3}{n^2} + \frac{C\alpha^2 \mathcal{C}_0}{2n(1 + \eta L)^2} \right) \sum_{r=0}^{R-1} \sum_{i=1}^{n} \|e_{i,r}^t\|^2$$

$$+ \frac{C(\mathcal{C}_2 + 2n\mathcal{C}_0)}{n^2} \sum_{r=0}^{R-1} \sum_{i=1}^{n} \|e_{i,r+1}^t\|^2$$

$$\overset{(a)}{=} \mathcal{D}_0^t - \mathbb{E}_{(S_0^t, \ldots, S_{R-1}^t)}[\mathcal{D}_0^{t+1}]$$

$$+ \left( \frac{C(\mathcal{C}_1 + 2n\mathcal{C}_0) + (n - C)\mathcal{C}_3}{n^2} + \frac{C\alpha^2 \mathcal{C}_0}{2n(1 + \eta L)^2} \right) \sum_{r=0}^{R-1} \sum_{i=1}^{n} \|e_{i,r}^t\|^2$$

$$+ \frac{C(\mathcal{C}_2 + 2n\mathcal{C}_0)}{n^2} \sum_{r=0}^{R-1} \sum_{i=1}^{n} \|e_{i,r+1}^t\|^2, \tag{54}$$

where (a) uses Definition 4.

Define

$$\mathcal{A}_1 := \frac{4(1 + \eta L)^4}{\eta^2 \mathcal{C}_0 \alpha^2 n}, \quad \mathcal{B}_1 := \frac{C(\mathcal{C}_1 + 2n\mathcal{C}_0) + (n - C)\mathcal{C}_3}{n} + \frac{C\alpha^2 \mathcal{C}_0}{2(1 + \eta L)^2}, \quad \mathcal{B}_2 := \frac{C(\mathcal{C}_2 + 2n\mathcal{C}_0)}{n}.$$

Using Lemma 3 and $R = \frac{n}{C}$, we get

$$\frac{1}{\mathcal{A}_1 R} \sum_{r=0}^{R-1} \|\nabla f(x_r^t)\|^2 \leq \mathcal{D}_0^t - \mathbb{E}_{S^t}[\mathcal{D}_0^{t+1}] + \frac{\mathcal{B}_1}{n} \sum_{r=0}^{R-1} \sum_{i=1}^{n} \|e_{i,r}^t\|^2 + \frac{\mathcal{B}_2}{n} \sum_{r=0}^{R-1} \sum_{i=1}^{n} \|e_{i,r+1}^t\|^2. \tag{55}$$

Finally, taking the expectation over $(S^0, \ldots, S^{t-1})$, we obtain

$$\frac{1}{\mathcal{A}_1 R} \mathbb{E}_{(S^0, S^1, \ldots, S^{t-1})} \left[ \sum_{r=0}^{R-1} \|\nabla f(x_r^t)\|^2 \right] \leq \mathbb{E}_{(S^0, S^1, \ldots, S^{t-1})}[\mathcal{D}_0^t] - \mathbb{E}_{(S^0, S^1, \ldots, S^t)}[\mathcal{D}_0^{t+1}]$$

$$+ \frac{\mathcal{B}_1}{n} \sum_{r=0}^{R-1} \sum_{i=1}^{n} \|e_{i,r}^t\|^2 + \frac{\mathcal{B}_2}{n} \sum_{r=0}^{R-1} \sum_{i=1}^{n} \|e_{i,r+1}^t\|^2. \tag{56}$$

$$\square$$

**Definition 5.** *(Meta-epoch Average Gradient Norm)*

$$\mathcal{G}^t = \frac{1}{n} \sum_{r=0}^{R-1} \|\nabla f(x_r^t)\|^2 \tag{57}$$

***Proof of Theorem 1.*** First, we define the constants $\mathcal{A}_1$, $\mathcal{B}_1$, and $\mathcal{B}_2$ as follows:

$$\mathcal{A}_1 := \frac{4(1 + \eta L)^4}{\eta^2 \mathcal{C}_0 \alpha^2 n},$$

$$\mathcal{B}_1 := \frac{C(\mathcal{C}_1 + 2n\mathcal{C}_0) + (n - C)\mathcal{C}_3}{n} + \frac{C\alpha^2 \mathcal{C}_0}{2(1 + \eta L)^2}, \tag{58}$$

$$\mathcal{B}_2 := \frac{C(\mathcal{C}_2 + 2n\mathcal{C}_0)}{n}.$$

Moreover, we define the constant $\eta_0$ as follows:

$$\eta_0 = \frac{-(\alpha L + 4\alpha\gamma L) + \sqrt{(\alpha L + 4\alpha\gamma L)^2 + 4(2 - \alpha)(2L^2 + 2\alpha\gamma L^2)}}{4L^2 + 4\alpha\gamma L^2} > 0, \tag{59}$$

where $\eta_0$ is the larger root of the quadratic equation

$$(2L^2 + 2\alpha\gamma L^2)\eta^2 + (\alpha L + 4\alpha\gamma L)\eta + (\alpha - 2) = 0.$$

The smaller root is given by

$$\eta_{0,-} = \frac{-(\alpha L + 4\alpha\gamma L) - \sqrt{(\alpha L + 4\alpha\gamma L)^2 + 4(2-\alpha)(2L^2 + 2\alpha\gamma L^2)}}{4L^2 + 4\alpha\gamma L^2} < 0.$$

By the properties of quadratic functions, if $0 < \eta < \eta_0$, then $\eta_{0,-} < \eta < \eta_0$. Therefore,

$$(2L^2 + 2\alpha\gamma L^2)\eta^2 + (\alpha L + 4\alpha\gamma L)\eta + (\alpha - 2) < 0,$$

i.e.,

$$2 - \alpha - \alpha\eta L - 2\eta^2 L^2 - 2\alpha\gamma(1 + \eta L)^2 > 0. \tag{60}$$

Using Lemma 6, the constant definitions in (58), and Definition 5, we obtain:

$$\frac{1}{\mathcal{A}_1}\mathbb{E}_{(S^0,\dots,S^{t-1})}[\mathcal{G}^t] \leq \mathbb{E}_{(S^0,\dots,S^{t-1})}[\mathcal{D}_0^t] - \mathbb{E}_{(S^0,\dots,S^t)}[\mathcal{D}_0^{t+1}]$$

$$+ \frac{\mathcal{B}_1}{n}\sum_{r=0}^{R-1}\sum_{i=1}^{n}\|e_{i,r}^t\|^2 + \frac{\mathcal{B}_2}{n}\sum_{r=0}^{R-1}\sum_{i=1}^{n}\|e_{i,r+1}^t\|^2. \tag{61}$$

Summing (61) from $t = 0$ to $t = T - 1$, we obtain:

$$\frac{1}{\mathcal{A}_1}\sum_{t=0}^{T-1}\mathbb{E}_{(S^0,\dots,S^{t-1})}[\mathcal{G}^t] \leq \mathcal{D}_0^0 - \mathbb{E}_{(S^0,\dots,S^{T-1})}[\mathcal{D}_0^T]$$

$$+ \frac{\mathcal{B}_1}{n}\sum_{t=0}^{T-1}\sum_{r=0}^{R-1}\sum_{i=1}^{n}\|e_{i,r}^t\|^2 + \frac{\mathcal{B}_2}{n}\sum_{t=0}^{T-1}\sum_{r=0}^{R-1}\sum_{i=1}^{n}\|e_{i,r+1}^t\|^2. \tag{62}$$

Due to the initialization in *Algorithm 1*, where $x_{i,0}^0 = x^0$ and $x_0^0 = x^0$, we have:

$$\mathcal{D}_0^0 = \frac{1}{n}\sum_{i=1}^{n}\left[f_i(x_{i,0}^0) + \langle\nabla f_i(x_{i,0}^0), x_0^0 - x_{i,0}^0\rangle + \frac{1}{2\eta}\|x_0^0 - x_{i,0}^0\|^2\right]$$

$$= \frac{1}{n}\sum_{i=1}^{n}f_i(x^0) \overset{(1)}{=} f(x^0). \tag{63}$$

Moreover, from the $L$-smoothness of $f_i$, we have the bound:

$$\mathcal{D}_r^t = \frac{1}{n}\sum_{i=1}^{n}\left[f_i(x_{i,r}^t) + \langle\nabla f_i(x_{i,r}^t), x_r^t - x_{i,r}^t\rangle + \frac{1}{2\eta}\|x_r^t - x_{i,r}^t\|^2\right]$$

$$\overset{(35)}{\geq} \frac{1}{n}\sum_{i=1}^{n}\left[f_i(x_r^t) - \frac{L}{2}\|x_r^t - x_{i,r}^t\|^2 + \frac{1}{2\eta}\|x_r^t - x_{i,r}^t\|^2\right]$$

$$= \frac{1}{n}\sum_{i=1}^{n}\left[f_i(x_r^t) + \left(\frac{1}{2\eta} - \frac{L}{2}\right)\|x_r^t - x_{i,r}^t\|^2\right]$$

$$\overset{(a)}{\geq} \frac{1}{n}\sum_{i=1}^{n}f_i(x_r^t) = f(x_r^t) \geq f^*, \tag{64}$$

where (a) uses the assumption $\eta L \leq 1$, which is implied by (60):

$$2 - \alpha - \alpha\eta L - 2\eta^2 L^2 - 2\alpha\gamma(1 + \gamma L)^2 > 0 \quad \Rightarrow \quad 2 - 2\eta^2 L^2 > 0 \quad \Rightarrow \quad \eta L \leq 1.$$

Therefore, we have

$$\mathbb{E}_{(S^0,\dots,S^t)}[\mathcal{D}_0^T] \geq f^*. \tag{65}$$

Using (61), (63), (65), and (20), we have

$$\frac{1}{\mathcal{A}_1} \sum_{t=0}^{T-1} \mathbb{E}_{(S^0,...,S^{t-1})}[\mathcal{G}^t] \leq f(x^0) - f^* + \frac{\mathcal{B}_1}{n} \sum_{t=0}^{T-1} \sum_{r=0}^{R-1} \sum_{i=1}^{n} \epsilon_{t,i,r}^2 + \frac{\mathcal{B}_2}{n} \sum_{t=0}^{T-1} \sum_{r=0}^{R-1} \sum_{i=1}^{n} \epsilon_{t,i,r+1}^2,$$

i.e.,

$$\frac{1}{T} \sum_{t=0}^{T-1} \mathbb{E}[\mathcal{G}^t] \leq \frac{\mathcal{A}_1[f(x^0) - f^*]}{T} + \frac{1}{nT} \sum_{t=0}^{T-1} \sum_{r=0}^{R-1} \sum_{i=1}^{n} (\mathcal{A}_1 \mathcal{B}_1 \epsilon_{t,i,r}^2 + \mathcal{A}_1 \mathcal{B}_2 \epsilon_{t,i,r+1}^2).$$

Finally, Let $\tilde{x}$ be selected uniformly at random from $\{x_0^0, x_1^0, ..., x_{R-1}^0, x_0^1, ..., x_{R-1}^{T-1}\}$ as the output of *Algorithm 1*. We have

$$\mathbb{E}[\|\nabla f(\tilde{x})\|^2] = \mathbb{E}\{\frac{1}{T} \sum_{t=0}^{T-1} \frac{1}{R} \sum_{r=0}^{R-1} [\|\nabla f(x_r^t)\|^2]\}$$

$$\overset{(a)}{=} \frac{1}{T} \sum_{t=0}^{T-1} \mathbb{E}[\mathcal{G}^t] \leq \frac{\mathcal{A}_1[f(x^0) - f^*]}{T} + \frac{1}{nT} \sum_{t=0}^{T-1} \sum_{r=0}^{R-1} \sum_{i=1}^{n} (\mathcal{A}_1 \mathcal{B}_1 \epsilon_{t,i,r}^2 + \mathcal{A}_1 \mathcal{B}_2 \epsilon_{t,i,r+1}^2),$$

where (a) uses Definition 5.

Since $\frac{1}{n} \sum_{i=1}^{n} \sum_{t=0}^{T-1} \sum_{r=0}^{R} \epsilon_{t,i,r}^2 \leq M$,

$$\frac{1}{n} \sum_{t=0}^{T-1} \sum_{r=0}^{R-1} \sum_{i=1}^{n} \epsilon_{t,i,r}^2 \leq \frac{1}{n} \sum_{t=0}^{T-1} \sum_{r=0}^{R} \sum_{i=1}^{n} \epsilon_{t,i,r}^2 \leq M, \tag{66}$$

$$\frac{1}{n} \sum_{t=0}^{T-1} \sum_{r=0}^{R-1} \sum_{i=1}^{n} \epsilon_{t,i,r+1}^2 \leq \frac{1}{n} \sum_{t=0}^{T-1} \sum_{r=-1}^{R-1} \sum_{i=1}^{n} \epsilon_{t,i,r+1}^2 \leq M. \tag{67}$$

Therefore, we have

$$\mathbb{E}[\|\nabla f(\tilde{x})\|^2] \leq \frac{\mathcal{A}_1[f(x^0) - f^*]}{T} + \frac{(\mathcal{A}_1 \mathcal{B}_1 + \mathcal{A}_1 \mathcal{B}_2)M}{T}.$$

Therefore, to ensure $\tilde{x}$ is a $\epsilon$-approximate stationary point, according to Definition 2, we need to guarantee $\mathbb{E}[\|\nabla f(\tilde{x})\|^2] \leq \epsilon^2$, if $T$ satisfies

$$T \geq \frac{\mathcal{A}_1(f(x^0) - f^*) + (\mathcal{A}_1 \mathcal{B}_1 + \mathcal{A}_1 \mathcal{B}_2)M}{\epsilon^2}. \tag{68}$$

Hence, we can select $K := \lceil \frac{\mathcal{A}_1(f(x^0) - f^*) + (\mathcal{A}_1 \mathcal{B}_1 + \mathcal{A}_1 \mathcal{B}_2)M}{\epsilon^2} \rceil = O(\epsilon^{-2})$ as its lower bound. $\qquad \square$

## C  PROOFS OF CONVERGENCE ANALYSIS FOR THE EXACT VARIANT OF *Algorithm 1*

**Lemma 7.** *Let* $\{(x_{i,r}^t, y_{i,r}^t, \hat{x}_{i,r}^t)\}$ *be the sequence generated by Algorithm 1, starting from the initial state* $x_i^0 = \mathrm{prox}_{\eta f_i}(y_i^0)$ *for all* $i \in [n]$. *Then, for all* $i \in [n]$, $r \geq 0$, *and* $t \geq 0$, *we have*

$$y_{i,r}^t = x_{i,r}^t + \eta \nabla f_i(x_{i,r}^t) \quad \text{and} \quad \hat{x}_{i,r}^t = 2x_{i,r}^t - y_{i,r}^t. \tag{69}$$

*Proof.* We prove (69) by induction on $t$.

**Base case** $(t = 0)$.

For $r = 0$, by the optimality condition of the proximal operator, the initialization of *Algorithm 1* ensures that

$$x_i^0 = \mathrm{prox}_{\eta f_i}(y_i^0), \quad \text{for all } i \in [n].$$

This implies

$$y_{i,0}^0 = x_{i,0}^0 + \eta \nabla f_i(x_{i,0}^0), \quad \text{and} \quad \hat{x}_{i,0}^0 = 2x_{i,0}^0 - y_{i,0}^0.$$

Assume that (69) holds for $t = 0$ and all $r \geq 0$, i.e.,

$$y_{i,r}^0 = x_{i,r}^0 + \eta \nabla f_i(x_{i,r}^0), \quad \hat{x}_{i,r}^0 = 2x_{i,r}^0 - y_{i,r}^0.$$

We show that it also holds for $r + 1$. We consider two cases:

- **Case 1:** $i \in S_r^0$. By the optimality condition of the proximal update,

$$\nabla f_i(x_{i,r+1}^0) + \frac{1}{\eta}(x_{i,r+1}^0 - y_{i,r+1}^0) = 0,$$

  which implies

$$y_{i,r+1}^0 = x_{i,r+1}^0 + \eta \nabla f_i(x_{i,r+1}^0).$$

  Furthermore, by (15),

$$\hat{x}_{i,r+1}^0 = 2x_{i,r+1}^0 - y_{i,r+1}^0.$$

- **Case 2:** $i \notin S_r^0$. By (16) and the induction hypothesis, we have

$$y_{i,r+1}^0 = x_{i,r+1}^0 + \eta \nabla f_i(x_{i,r+1}^0), \quad \hat{x}_{i,r+1}^0 = 2x_{i,r+1}^0 - y_{i,r+1}^0.$$

Thus, in both cases, (69) holds for all $i \in [n]$ at $t = 0$ and all $r \geq 0$.

**Induction step.**

Assume that (69) holds for a fixed $t \geq 0$ and all $r \geq 0$, i.e.,

$$y_{i,r}^t = x_{i,r}^t + \eta \nabla f_i(x_{i,r}^t), \quad \hat{x}_{i,r}^t = 2x_{i,r}^t - y_{i,r}^t.$$

We will show that (69) holds for $t + 1$.

By (18) and the induction assumption, we have

$$y_{i,R}^t = x_{i,R}^t + \eta \nabla f_i(x_{i,R}^t), \quad \hat{x}_{i,R}^t = 2x_{i,R}^t - y_{i,R}^t.$$

Then, by the initialization of the next epoch,

$$y_{i,0}^{t+1} = x_{i,0}^{t+1} + \eta \nabla f_i(x_{i,0}^{t+1}), \quad \hat{x}_{i,0}^{t+1} = 2x_{i,0}^{t+1} - y_{i,0}^{t+1}.$$

Using the same argument as in the base case for $t = 0$, we conclude that equation 69 holds for $r \geq 0$ at $t + 1$.

This completes the proof. $\qquad \square$

**Lemma 8.** *Let* $\{(x_r^t, x_{i,r}^t, y_{i,r}^t)\}$ *be the sequence generated by Algorithm 1, and let* $\alpha > 0$. *Then, for all* $i \in S_r^t$, *we have*

$$\|x_r^t - x_{i,r}^t\|^2 \leq \frac{(1 + \eta L)^2}{\alpha^2} \|x_{i,r}^t - x_{i,r+1}^t\|^2. \tag{70}$$

*Proof.* By the update rule of $y_{i,r+1}^t$ and Lemma 7, for $i \in S_r^t$, we have

$$
\begin{aligned}
x_r^t - x_{i,r}^t &= \frac{1}{\alpha}(y_{i,r+1}^t - y_{i,r}^t) \\
&= \frac{1}{\alpha}(x_{i,r+1}^t - x_{i,r}^t) + \frac{\eta}{\alpha}\left(\nabla f_i(x_{i,r+1}^t) - \nabla f_i(x_{i,r}^t)\right).
\end{aligned}
\tag{71}
$$

Using this expression and the triangle inequality $\|a + b\| \le \|a\| + \|b\|$, we obtain

$$
\begin{aligned}
\|x_r^t - x_{i,r}^t\|^2 &= \left\|\frac{1}{\alpha}(x_{i,r+1}^t - x_{i,r}^t) + \frac{\eta}{\alpha}(\nabla f_i(x_{i,r+1}^t) - \nabla f_i(x_{i,r}^t))\right\|^2 \\
&\le \frac{1}{\alpha^2}\left(\|x_{i,r+1}^t - x_{i,r}^t\| + \eta\|\nabla f_i(x_{i,r+1}^t) - \nabla f_i(x_{i,r}^t)\|\right)^2 \\
&\overset{(a)}{\le} \frac{1}{\alpha^2}\left(\|x_{i,r+1}^t - x_{i,r}^t\| + \eta L\|x_{i,r+1}^t - x_{i,r}^t\|\right)^2 \\
&= \frac{(1 + \eta L)^2}{\alpha^2}\|x_{i,r+1}^t - x_{i,r}^t\|^2,
\end{aligned}
\tag{72}
$$

where step (a) follows from the $L$-smoothness of $f_i$. $\qquad\square$

**Lemma 9.** *Let $\{(x_r^t, x_{i,r}^t, \hat{x}_{i,r}^t)\}$ be the sequence generated by Algorithm 1, and let $\alpha > 0$. Then, for any $t \ge 0$ and $r \ge 0$, the following bound holds:*

$$
\|\nabla f(x_r^t)\|^2 \le \frac{(1 + \eta L)^2}{n\eta^2}\sum_{i=1}^n \|x_{i,r}^t - x_r^t\|^2.
\tag{73}
$$

*Proof.* From the aggregation step of *Algorithm 1* and Lemma 7, we have

$$
x_r^t = \frac{1}{n}\sum_{i=1}^n \hat{x}_{i,r}^t = \frac{1}{n}\sum_{i=1}^n(2x_{i,r}^t - y_{i,r}^t) = \frac{1}{n}\sum_{i=1}^n(x_{i,r}^t - \eta\nabla f_i(x_{i,r}^t)).
\tag{74}
$$

Using this identity, we analyze the norm of the gradient:

$$
\begin{aligned}
\|\nabla f(x_r^t)\|^2 &= \left\|\frac{1}{n}\sum_{i=1}^n \nabla f_i(x_r^t)\right\|^2 \\
&= \left\|\frac{1}{\eta}\left(x_r^t - \frac{1}{n}\sum_{i=1}^n x_{i,r}^t + \eta\nabla f_i(x_r^t) - \eta\nabla f_i(x_{i,r}^t)\right)\right\|^2 \\
&= \frac{1}{\eta^2}\left\|\frac{1}{n}\sum_{i=1}^n\left(x_{i,r}^t - x_r^t + \eta\nabla f_i(x_r^t) - \eta\nabla f_i(x_{i,r}^t)\right)\right\|^2 \\
&\le \frac{1}{\eta^2}\left(\frac{1}{n}\sum_{i=1}^n \left\|x_{i,r}^t - x_r^t + \eta\left(\nabla f_i(x_r^t) - \nabla f_i(x_{i,r}^t)\right)\right\|\right)^2 \\
&\overset{(a)}{\le} \frac{1}{\eta^2}\left(\frac{1}{n}\sum_{i=1}^n(1 + \eta L)\|x_{i,r}^t - x_r^t\|\right)^2 \\
&\overset{(b)}{\le} \frac{(1 + \eta L)^2}{n\eta^2}\sum_{i=1}^n \|x_{i,r}^t - x_r^t\|^2,
\end{aligned}
\tag{75}
$$

where

(a) uses the $L$-smoothness of each $f_i$, i.e., $\|\nabla f_i(x_r^t) - \nabla f_i(x_{i,r}^t)\| \le L\|x_r^t - x_{i,r}^t\|$, and the triangle inequality $\|a + b\| \le \|a\| + \|b\|$,

(b) uses Jensen's inequality: $\left(\sum_{i=1}^n A_i\right)^2 \le n\sum_{i=1}^n A_i^2$.

$\square$

**Lemma 10.** *Let $\{(x_r^t, x_{i,r}^t, y_{i,r}^t, \hat{x}_{i,r}^t)\}$ be the sequence generated by Algorithm 1, and let $\mathcal{D}_r^t$ be the potential function defined in Definition 4. Then, for all $t \geq 0$ and $r \geq 0$, the following descent inequality holds:*

$$\mathcal{D}_{r+1}^t \leq \mathcal{D}_r^t - \frac{2 - \alpha - \alpha\eta L - 2\eta^2 L^2}{2n\alpha\eta} \sum_{i \in S_r^t} \|x_{i,r+1}^t - x_{i,r}^t\|^2. \tag{76}$$

*Proof.* We begin with the identity:

$$\|x_{r+1}^t - x_{i,r+1}^t\|^2 - \|x_r^t - x_{i,r+1}^t\|^2 - \|x_{r+1}^t - x_r^t\|^2 = 2\left\langle x_r^t - x_{i,r+1}^t, \, x_{r+1}^t - x_r^t \right\rangle. \tag{77}$$

Using this identity, we expand the potential function $\mathcal{D}_{r+1}^t$:

$$\mathcal{D}_{r+1}^t = \frac{1}{n} \sum_{i=1}^n \left[ f_i(x_{i,r+1}^t) + \left\langle \nabla f_i(x_{i,r+1}^t), \, x_{r+1}^t - x_{i,r+1}^t \right\rangle + \frac{1}{2\eta} \|x_{i,r+1}^t - x_{r+1}^t\|^2 \right]$$

$$= \frac{1}{n} \sum_{i=1}^n \left[ f_i(x_{i,r+1}^t) + \left\langle \nabla f_i(x_{i,r+1}^t), \, x_r^t - x_{i,r+1}^t \right\rangle + \frac{1}{2\eta} \|x_{i,r+1}^t - x_r^t\|^2 \right]$$

$$+ \frac{1}{n} \sum_{i=1}^n \left\langle \nabla f_i(x_{i,r+1}^t), \, x_{r+1}^t - x_r^t \right\rangle + \frac{1}{2\eta} \left( \|x_{r+1}^t - x_{i,r+1}^t\|^2 - \|x_{i,r+1}^t - x_r^t\|^2 \right)$$

$$\overset{(77)}{=} \frac{1}{n} \sum_{i=1}^n \left[ f_i(x_{i,r+1}^t) + \left\langle \nabla f_i(x_{i,r+1}^t), \, x_r^t - x_{i,r+1}^t \right\rangle + \frac{1}{2\eta} \|x_{i,r+1}^t - x_r^t\|^2 \right]$$

$$+ \frac{1}{n} \sum_{i=1}^n \left\langle \nabla f_i(x_{i,r+1}^t), \, x_{r+1}^t - x_r^t \right\rangle + \frac{1}{2\eta} \left( \|x_{r+1}^t - x_r^t\|^2 + 2\langle x_r^t - x_{i,r+1}^t, x_{r+1}^t - x_r^t\rangle \right)$$

$$= \frac{1}{n} \sum_{i=1}^n \left[ f_i(x_{i,r+1}^t) + \left\langle \nabla f_i(x_{i,r+1}^t), \, x_r^t - x_{i,r+1}^t \right\rangle + \frac{1}{2\eta} \|x_{i,r+1}^t - x_r^t\|^2 \right]$$

$$+ \frac{1}{n\eta} \sum_{i=1}^n \left\langle x_r^t - 2x_{i,r+1}^t + (x_{i,r+1}^t + \eta\nabla f_i(x_{i,r+1}^t)), \, x_{r+1}^t - x_r^t \right\rangle + \frac{1}{2\eta} \|x_{r+1}^t - x_r^t\|^2$$

$$\overset{(a)}{=} \frac{1}{n} \sum_{i=1}^n \left[ f_i(x_{i,r+1}^t) + \left\langle \nabla f_i(x_{i,r+1}^t), \, x_r^t - x_{i,r+1}^t \right\rangle + \frac{1}{2\eta} \|x_{i,r+1}^t - x_r^t\|^2 \right]$$

$$+ \frac{1}{n\eta} \sum_{i=1}^n \left\langle x_r^t - \hat{x}_{i,r+1}^t, \, x_{r+1}^t - x_r^t \right\rangle + \frac{1}{2\eta} \|x_{r+1}^t - x_r^t\|^2, \tag{78}$$

where (a) uses Lemma 7, i.e., $y_{i,r+1}^t = x_{i,r+1}^t + \eta\nabla f_i(x_{i,r+1}^t)$ and $\hat{x}_{i,r+1}^t = 2x_{i,r+1}^t - y_{i,r+1}^t$.

Then, applying the server update rule from (17), i.e., $x_{r+1}^t = \frac{1}{n} \sum_{i=1}^n \hat{x}_{i,r+1}^t$, we get:

$$\mathcal{D}_{r+1}^t = \frac{1}{n} \sum_{i=1}^n \left[ f_i(x_{i,r+1}^t) + \left\langle \nabla f_i(x_{i,r+1}^t), \, x_r^t - x_{i,r+1}^t \right\rangle + \frac{1}{2\eta} \|x_{i,r+1}^t - x_r^t\|^2 \right] - \frac{1}{2\eta} \|x_{r+1}^t - x_r^t\|^2.$$

We now split the summation over $i \in [n]$ into the active and inactive sets at round $r$, i.e., $S_r^t$ and its complement:

$$
\mathcal{D}_{r+1}^t = \frac{1}{n} \sum_{i \in S_r^t} f_i(x_{i,r+1}^t) + \frac{1}{n} \sum_{i \in S_r^t} \langle \nabla f_i(x_{i,r+1}^t),\, x_{i,r}^t - x_{i,r+1}^t \rangle + \frac{1}{n} \sum_{i \in S_r^t} \langle \nabla f_i(x_{i,r+1}^t),\, x_r^t - x_{i,r}^t \rangle
$$

$$
+ \frac{1}{2n\eta} \sum_{i \in S_r^t} \|x_r^t - x_{i,r+1}^t\|^2 + \frac{1}{n} \sum_{i \notin S_r^t} f_i(x_{i,r}^t) + \frac{1}{n} \sum_{i \notin S_r^t} \langle \nabla f_i(x_{i,r}^t),\, x_r^t - x_{i,r}^t \rangle
$$

$$
+ \frac{1}{2n\eta} \sum_{i \notin S_r^t} \|x_r^t - x_{i,r}^t\|^2 - \frac{1}{2\eta} \|x_{r+1}^t - x_r^t\|^2. \tag{79}
$$

By the $L$-smoothness property of $f_i$, we have

$$
f_i(x_{i,r+1}^t) + \langle \nabla f_i(x_{i,r+1}^t),\, x_{i,r}^t - x_{i,r+1}^t \rangle \leq f_i(x_{i,r}^t) + \frac{L}{2} \|x_{i,r}^t - x_{i,r+1}^t\|^2. \tag{80}
$$

Substituting this into the previous expression, we get:

$$
\mathcal{D}_{r+1}^t \overset{(80)}{\leq} \frac{1}{n} \sum_{i \in S_r^t} \left[ f_i(x_{i,r}^t) + \langle \nabla f_i(x_{i,r}^t),\, x_r^t - x_{i,r}^t \rangle + \frac{1}{2\eta} \|x_r^t - x_{i,r+1}^t\|^2 \right]
$$

$$
+ \frac{L}{2n} \sum_{i \in S_r^t} \|x_{i,r}^t - x_{i,r+1}^t\|^2
$$

$$
+ \frac{1}{n} \sum_{i \notin S_r^t} \left[ f_i(x_{i,r}^t) + \langle \nabla f_i(x_{i,r}^t),\, x_r^t - x_{i,r}^t \rangle + \frac{1}{2\eta} \|x_r^t - x_{i,r}^t\|^2 \right] - \frac{1}{2\eta} \|x_r^t - x_{r+1}^t\|^2
$$

$$
+ \frac{1}{n} \sum_{i \in S_r^t} \langle \nabla f_i(x_{i,r+1}^t) - \nabla f_i(x_{i,r}^t),\, x_r^t - x_{i,r}^t \rangle. \tag{81}
$$

Moreover, we use the identity:

$$
\|x_r^t - x_{i,r+1}^t\|^2 = \|x_r^t - x_{i,r}^t\|^2 + 2 \langle x_r^t - x_{i,r}^t,\, x_{i,r}^t - x_{i,r+1}^t \rangle + \|x_{i,r}^t - x_{i,r+1}^t\|^2. \tag{82}
$$

Substituting (82) into the earlier bound, we obtain:

$$
\mathcal{D}_{r+1}^t \leq \frac{1}{n} \sum_{i=1}^n \left[ f_i(x_{i,r}^t) + \langle \nabla f_i(x_{i,r}^t),\, x_r^t - x_{i,r}^t \rangle + \frac{1}{2\eta} \|x_r^t - x_{i,r}^t\|^2 \right] - \frac{1}{2\eta} \|x_r^t - x_{r+1}^t\|^2
$$

$$
+ \frac{L}{2n} \sum_{i \in S_r^t} \|x_{i,r}^t - x_{i,r+1}^t\|^2 + \frac{1}{n} \sum_{i \in S_r^t} \langle \nabla f_i(x_{i,r+1}^t) - \nabla f_i(x_{i,r}^t),\, x_r^t - x_{i,r}^t \rangle
$$

$$
+ \frac{1}{n\eta} \sum_{i \in S_r^t} \langle x_r^t - x_{i,r}^t,\, x_{i,r}^t - x_{i,r+1}^t \rangle + \frac{1}{2n\eta} \sum_{i \in S_r^t} \|x_{i,r}^t - x_{i,r+1}^t\|^2
$$

$$
= \mathcal{D}_r^t - \frac{1}{2\eta} \|x_r^t - x_{r+1}^t\|^2 + \frac{1 + \eta L}{2n\eta} \sum_{i \in S_r^t} \|x_{i,r}^t - x_{i,r+1}^t\|^2
$$

$$
+ \frac{1}{n} \sum_{i \in S_r^t} \langle \nabla f_i(x_{i,r+1}^t) - \nabla f_i(x_{i,r}^t),\, x_r^t - x_{i,r}^t \rangle + \frac{1}{n\eta} \sum_{i \in S_r^t} \langle x_r^t - x_{i,r}^t,\, x_{i,r}^t - x_{i,r+1}^t \rangle. \tag{83}
$$

Now, for $i \in S_r^t$, using (71) and the $L$-smoothness of $f_i$, we have:

$$
\mathcal{D}_{r+1}^t \leq \mathcal{D}_r^t - \frac{1}{2\eta} \|x_r^t - x_{r+1}^t\|^2 + \frac{1 + \eta L}{2n\eta} \sum_{i \in S_r^t} \|x_{i,r}^t - x_{i,r+1}^t\|^2
$$

$$
+ \frac{\eta}{n\alpha} \sum_{i \in S_r^t} \|\nabla f_i(x_{i,r+1}^t) - \nabla f_i(x_{i,r}^t)\|^2 + \frac{1}{n\alpha} \sum_{i \in S_r^t} \langle \nabla f_i(x_{i,r+1}^t) - \nabla f_i(x_{i,r}^t),\, x_{i,r+1}^t - x_{i,r}^t \rangle
$$

$$
+ \frac{1}{n\alpha} \sum_{i \in S_r^t} \langle \nabla f_i(x_{i,r+1}^t) - \nabla f_i(x_{i,r}^t),\, x_{i,r}^t - x_{i,r+1}^t \rangle - \frac{1}{n\alpha\eta} \sum_{i \in S_r^t} \|x_{i,r+1}^t - x_{i,r}^t\|^2.
$$

Finally, using the $L$-smoothness of $f_i$ (i.e., $\|\nabla f_i(x) - \nabla f_i(y)\| \le L\|x - y\|$) and simplifying the inner products, we obtain:

$$\mathcal{D}_{r+1}^t \le \mathcal{D}_r^t - \frac{2 - \alpha - \alpha\eta L - 2\eta^2 L^2}{2n\alpha\eta} \sum_{i \in S_r^t} \|x_{i,r+1}^t - x_{i,r}^t\|^2, \tag{84}$$

where the final step uses the $L$-smoothness of $f_i$, and omits the nonnegative term $\|x_r^t - x_{r+1}^t\|^2$ to obtain a cleaner upper bound. $\square$

**Lemma 11.** *Let $\{(x_r^t, x_{i,r}^t, y_{i,r}^t, \hat{x}_{i,r}^t)\}$ be the sequence generated by Algorithm 1, and let $\mathcal{D}_r^t$ be the potential function defined in Definition 4. Then, the following expected descent inequality holds:*

$$\mathbb{E}_{(S_0^t,\ldots,S_r^t)}\left[\mathcal{D}_{r+1}^t\right] \le \mathbb{E}_{(S_0^t,\ldots,S_{r-1}^t)}\left[\mathcal{D}_r^t\right] - \frac{\alpha C \left(2 - \alpha - \alpha\eta L - 2\eta^2 L^2\right)}{2n^2\eta(1+\eta L)^2} \sum_{i=1}^n \|x_{i,r}^t - x_r^t\|^2. \tag{85}$$

*Proof.* From Lemma 8 and Lemma 10, we have

$$\begin{aligned}
\mathcal{D}_{r+1}^t &\le \mathcal{D}_r^t - \frac{2 - \alpha - \alpha\eta L - 2\eta^2 L^2}{2n\alpha\eta} \sum_{i \in S_r^t} \|x_{i,r+1}^t - x_{i,r}^t\|^2 \\
&\le \mathcal{D}_r^t - \frac{\alpha(2 - \alpha - \alpha\eta L - 2\eta^2 L^2)}{2n\eta(1+\eta L)^2} \sum_{i \in S_r^t} \|x_{i,r}^t - x_r^t\|^2.
\end{aligned} \tag{86}$$

Taking the expectation over $(S_0^t, \ldots, S_r^t)$, we obtain

$$\mathbb{E}_{(S_0^t,\ldots,S_r^t)}[\mathcal{D}_{r+1}^t] \le \mathbb{E}_{(S_0^t,\ldots,S_r^t)}[\mathcal{D}_r^t] - \frac{\alpha(2 - \alpha - \alpha\eta L - 2\eta^2 L^2)}{2n\eta(1+\eta L)^2}\mathbb{E}_{(S_0^t,\ldots,S_r^t)}\left[\sum_{i \in S_r^t} \|x_{i,r}^t - x_r^t\|^2\right]. \tag{87}$$

Note that $\mathcal{D}_r^t$ depends only on $(S_0^t, \ldots, S_{r-1}^t)$, so we have

$$\mathbb{E}_{(S_0^t,\ldots,S_r^t)}[\mathcal{D}_r^t] = \mathbb{E}_{(S_0^t,\ldots,S_{r-1}^t)}[\mathcal{D}_r^t]. \tag{88}$$

Using the law of total expectation, we compute:

$$\begin{aligned}
\mathbb{E}_{(S_0^t,\ldots,S_r^t)}\left[\sum_{i \in S_r^t} \|x_{i,r}^t - x_r^t\|^2\right] &= \mathbb{E}_{(S_0^t,\ldots,S_{r-1}^t)}\left\{\mathbb{E}_{S_r^t}\left[\sum_{i \in S_r^t} \|x_{i,r}^t - x_r^t\|^2 \,\Big|\, S_0, \ldots, S_{r-1}^t\right]\right\} \\
&\stackrel{(a)}{=} \mathbb{E}_{(S_0^t,\ldots,S_{r-1}^t)}\left[\frac{C}{n - rC} \sum_{i \notin (S_0^t,\ldots,S_{r-1}^t)} \|x_{i,r}^t - x_r^t\|^2\right],
\end{aligned} \tag{89}$$

where step (a) follows from the sampling lemma (Lemma 1 in Malinovsky et al. (2023b)).

We now write the expectation over all permutations explicitly:

$$\begin{aligned}
\mathbb{E}_{(S_0^t,\ldots,S_r^t)}\left[\sum_{i \in S_r^t} \|x_{i,r}^t - x_r^t\|^2\right] &= \frac{(C!)^r(n - rC)!}{n!} \sum_{(S_0^t,\ldots,S_{r-1}^t)}\left[\frac{C}{n - rC} \sum_{i \notin (S_0^t,\ldots,S_{r-1}^t)} \|x_{i,r}^t - x_r^t\|^2\right] \\
&= \frac{(C!)^r(n - rC)! \cdot C}{n!(n - rC)} \sum_{(S_0^t,\ldots,S_{r-1}^t)} \sum_{i \notin (S_0^t,\ldots,S_{r-1}^t)} \|x_{i,r}^t - x_r^t\|^2.
\end{aligned} \tag{90}$$

To simplify the final summation, we fix $i = k$ and consider the term $\|x_{k,r}^t - x_r^t\|^2$. This term appears in the summation if and only if $k \notin (S_0^t, \ldots, S_{r-1}^t)$. In other words, we can choose a permutation of length $rC$ from the set $\{0, 1, \ldots, k-1, k+1, \ldots, n-1\}$ to form $(S_0^t, \ldots, S_{r-1}^t)$, and then remove repetitions due to the internal ordering within each $S_i^t$ (each of size $C$).

The number of such permutations is $\frac{(n-1)!}{(n-rC-1)!}$, and the number of repetitions from permutations inside each round is $(C!)^r$. Therefore, the total number of times the term $\|x^t_{k,r} - x^t_r\|^2$ appears in the summation is

$$\frac{(n-1)!}{(n-rC-1)!(C!)^r}.$$

Thus, we compute:

$$
\begin{aligned}
\mathbb{E}_{(S^t_0,\ldots,S^t_r)}\left[\sum_{i\in S^t_r}\|x^t_{i,r} - x^t_r\|^2\right] &= \frac{(C!)^r(n-rC)!\cdot C}{n!(n-rC)}\sum_{(S^t_0,\ldots,S^t_{r-1})}\sum_{i\notin(S^t_0,\ldots,S^t_{r-1})}\|x^t_{i,r} - x^t_r\|^2 \\
&= \frac{(C!)^r(n-rC)!\cdot C}{n!(n-rC)}\sum_{i=1}^n\left[\frac{(n-1)!}{(n-rC-1)!(C!)^r}\|x^t_{i,r} - x^t_r\|^2\right] \\
&= \frac{C}{n}\sum_{i=1}^n\|x^t_{i,r} - x^t_r\|^2.
\end{aligned}
\tag{91}
$$

Finally, combining inequalities (86), (88), and (91), we conclude:

$$\mathbb{E}_{(S^t_0,\ldots,S^t_r)}[\mathcal{D}^t_{r+1}] \le \mathbb{E}_{(S^t_0,\ldots,S^t_{r-1})}[\mathcal{D}^t_r] - \frac{\alpha C\left(2-\alpha-\alpha\eta L-2\eta^2 L^2\right)}{2n^2\eta(1+\eta L)^2}\sum_{i=1}^n\|x^t_{i,r} - x^t_r\|^2. \tag{92}$$

$\square$

**Lemma 12.** *Let $\{(x^t_r, x^t_{i,r}, y^t_{i,r})\}$ be the sequence generated by Algorithm 1, and let $\mathcal{D}^t_r$ be defined as in Definition 4. Then, the following inequality holds:*

$$\frac{\alpha\eta(2-\alpha-\alpha\eta L-2\eta^2 L^2)}{2(1+\eta L)^4}\mathbb{E}_{(S^0,\ldots,S^{t-1})}\left[\frac{1}{R}\sum_{r=0}^{R-1}\|\nabla f(x^t_r)\|^2\right] \le \mathbb{E}_{(S^0,\ldots,S^{t-1})}[\mathcal{D}^t_0] - \mathbb{E}_{(S^0,\ldots,S^t)}[\mathcal{D}^{t+1}_0]. \tag{93}$$

*Proof.* By Lemma 11, we have

$$\frac{\alpha C(2-\alpha-\alpha\eta L-2\eta^2 L^2)}{2n^2\eta(1+\eta L)^2}\sum_{i=1}^n\|x^t_{i,r} - x^t_r\|^2 \le \mathbb{E}_{(S^t_0,\ldots,S^t_{r-1})}[\mathcal{D}^t_r] - \mathbb{E}_{(S^t_0,\ldots,S^t_r)}[\mathcal{D}^t_{r+1}]. \tag{94}$$

Summing the inequality (94) over $r = 0$ to $R-1$, we obtain

$$\frac{\alpha C(2-\alpha-\alpha\eta L-2\eta^2 L^2)}{2n^2\eta(1+\eta L)^2}\sum_{r=0}^{R-1}\sum_{i=1}^n\|x^t_{i,r} - x^t_r\|^2 \le \mathcal{D}^t_0 - \mathbb{E}_{(S^t_0,\ldots,S^t_{R-1})}[\mathcal{D}^t_R] \overset{(a)}{=} \mathcal{D}^t_0 - \mathbb{E}_{S^t}[\mathcal{D}^{t+1}_0], \tag{95}$$

where (a) follows from Definition 4.

Next, by Lemma 9 and the identity $R = \frac{n}{C}$, we have

$$\frac{\alpha\eta(2-\alpha-\alpha\eta L-2\eta^2 L^2)}{2(1+\eta L)^4}\cdot\frac{1}{R}\sum_{r=0}^{R-1}\|\nabla f(x^t_r)\|^2 \le \mathcal{D}^t_0 - \mathbb{E}_{S^t}[\mathcal{D}^{t+1}_0]. \tag{96}$$

Finally, taking expectation with respect to $(S^0,\ldots,S^{t-1})$, we obtain

$$\frac{\alpha\eta(2-\alpha-\alpha\eta L-2\eta^2 L^2)}{2(1+\eta L)^4}\mathbb{E}_{(S^0,\ldots,S^{t-1})}\left[\frac{1}{R}\sum_{r=0}^{R-1}\|\nabla f(x^t_r)\|^2\right] \le \mathbb{E}_{(S^0,\ldots,S^{t-1})}[\mathcal{D}^t_0] - \mathbb{E}_{(S^0,\ldots,S^t)}[\mathcal{D}^{t+1}_0]. \tag{97}$$

$\square$

***Proof of Theorem 2***. First, we define the constant $\mathcal{A}$ as follows:

$$\mathcal{A} := \frac{2(1 + \eta L)^4}{\alpha\eta(2 - \alpha - \alpha\eta L - 2\eta^2 L^2)}. \tag{98}$$

Moreover, we define the constant $\eta_1$ as follows:

$$\eta_1 = \frac{-\alpha L + \sqrt{(\alpha L)^2 + 4(2 - \alpha)(2L^2)}}{4L^2} > 0, \tag{99}$$

where $\eta_1$ is the larger root of the quadratic equation

$$(2L^2)\eta^2 + (\alpha L)\eta + (\alpha - 2) = 0.$$

The smaller root is given by

$$\eta_{1,-} = \frac{-(\alpha L) - \sqrt{(\alpha L)^2 + 4(2 - \alpha)(2L^2)}}{4L^2} < 0.$$

By the properties of quadratic functions, if $0 < \eta < \eta_1$, then $\eta_{1,-} < \eta < \eta_1$. Therefore,

$$(2L^2)\eta^2 + (\alpha L)\eta + (\alpha - 2) < 0,$$

i.e.,

$$2 - \alpha - \alpha\eta L - 2\eta^2 L^2 > 0. \tag{100}$$

Using Lemma 12, (98), and Definition 5, we obtain

$$\frac{1}{\mathcal{A}}\mathbb{E}_{(S^0,\ldots,S^{t-1})}[\mathcal{G}^t] \le \mathbb{E}_{(S^0,\ldots,S^{t-1})}[\mathcal{D}_0^t] - \mathbb{E}_{(S^0,\ldots,S^t)}[\mathcal{D}_0^{t+1}]. \tag{101}$$

Summing (101) over $t = 0$ to $T - 1$, we have

$$\frac{1}{\mathcal{A}}\sum_{t=0}^{T-1}\mathbb{E}_{(S^0,\ldots,S^{t-1})}[\mathcal{G}^t] \le \mathcal{D}_0^0 - \mathbb{E}_{(S^0,\ldots,S^T)}[\mathcal{D}_0^T]. \tag{102}$$

From the initialization of *Algorithm 1*, i.e., $x_{i,0}^0 = x^0$ and $x_0^0 = x^0$, we have

$$\begin{aligned}
\mathcal{D}_0^0 &= \frac{1}{n}\sum_{i=1}^n\left[f_i(x_{i,0}^0) + \left\langle\nabla f_i(x_{i,0}^0), x_0^0 - x_{i,0}^0\right\rangle + \frac{1}{2\eta}\|x_0^0 - x_{i,0}^0\|^2\right] \\
&= \frac{1}{n}\sum_{i=1}^n f_i(x^0) \stackrel{(1)}{=} f(x^0).
\end{aligned} \tag{103}$$

Moreover, using the $L$-smoothness of $f_i$, we obtain

$$\begin{aligned}
\mathcal{D}_r^t &= \frac{1}{n}\sum_{i=1}^n\left[f_i(x_{i,r}^t) + \left\langle\nabla f_i(x_{i,r}^t), x_r^t - x_{i,r}^t\right\rangle + \frac{1}{2\eta}\|x_r^t - x_{i,r}^t\|^2\right] \\
&\stackrel{(80)}{\ge} \frac{1}{n}\sum_{i=1}^n\left[f_i(x_r^t) - \frac{L}{2}\|x_r^t - x_{i,r}^t\|^2 + \frac{1}{2\eta}\|x_r^t - x_{i,r}^t\|^2\right] \\
&= \frac{1}{n}\sum_{i=1}^n\left[f_i(x_r^t) + \left(\frac{1}{2\eta} - \frac{L}{2}\right)\|x_r^t - x_{i,r}^t\|^2\right] \\
&\stackrel{(a)}{\ge} \frac{1}{n}\sum_{i=1}^n f_i(x_r^t) = f(x_r^t) \ge f^*,
\end{aligned} \tag{104}$$

where (a) follows from the condition $\eta L \le 1$, which is implied by (100):

$$2 - \alpha - \alpha\eta L - 2\eta^2 L^2 > 0 \ \Rightarrow\ 2 - 2\eta^2 L^2 > 0 \ \Rightarrow\ \eta L \le 1. \tag{105}$$

Therefore,

$$\mathbb{E}_{(S^0,\ldots,S^T)}[\mathcal{D}_0^T] \geq f^*. \tag{106}$$

Combining (101), (103), and (106), we get

$$\frac{1}{\mathcal{A}} \sum_{t=0}^{T-1} \mathbb{E}_{(S^0,\ldots,S^{t-1})}[\mathcal{G}^t] \leq f(x^0) - f^*, \tag{107}$$

which implies

$$\frac{1}{T} \sum_{t=0}^{T-1} \mathbb{E}[\mathcal{G}^t] \leq \frac{\mathcal{A}(f(x^0) - f^*)}{T}. \tag{108}$$

Finally, let $\tilde{x}$ be selected uniformly at random from the sequence $\{x_0^0, x_1^0, \ldots, x_{R-1}^0, x_0^1, \ldots, x_{R-1}^{T-1}\}$, as the output of *Algorithm 1*. Then,

$$\begin{aligned}
\mathbb{E}[\|\nabla f(\tilde{x})\|^2] &= \mathbb{E}\left[\frac{1}{T}\sum_{t=0}^{T-1}\frac{1}{R}\sum_{r=0}^{R-1}\|\nabla f(x_r^t)\|^2\right] \\
&\overset{(a)}{=} \frac{1}{T}\sum_{t=0}^{T-1}\mathbb{E}[\mathcal{G}^t] \leq \frac{\mathcal{A}(f(x^0) - f^*)}{T},
\end{aligned} \tag{109}$$

where (a) follows from Definition 5.

To ensure that $\tilde{x}$ is an $\epsilon$-approximate stationary point (as defined in Definition 2), we require

$$\mathbb{E}[\|\nabla f(\tilde{x})\|^2] \leq \epsilon^2.$$

This is achieved by choosing

$$T \geq \frac{\mathcal{A}(f(x^0) - f^*)}{\epsilon^2},$$

i.e., it suffices to choose lower bound as follows:

$$K := \left\lceil \frac{\mathcal{A}(f(x^0) - f^*)}{\epsilon^2} \right\rceil = O(\epsilon^{-2}).$$

$\square$

# D  DETAILED DISCUSSIONS: WHY NAIVE CLIENT SAMPLING STRATEGIES SUFFER FROM THE LOW CLIENT PARTICIPATION ISSUE

In FL with a large number of clients, it is often infeasible to involve all clients in every communication round due to computation, communication, and network constraints. A common approach in prior work is to select a subset of clients at each round using *naive sampling*, i.e., drawing clients independently *with replacement* from the entire client set. Each client $k$ is selected with a fixed probability $p_k$. However, this naive strategy leads to inefficiency and unfairness, and suffers from the low client participation issue. In particular, naive sampling methods face two main challenges: *the low-coverage issue* and *the imbalanced-selection issue*.

Suppose there are $K$ clients in total, and $\gamma K$ clients are sampled in each communication round, with each client $k$ being chosen with probability $p_k$ (uniform sampling corresponds to $p_k = 1/K$). Here, $\gamma$ denotes the client participation fraction, and $R$ denotes the total number of communication rounds. In a single communication round, the probability that client $k$ participates at least once is

$$r_k = 1 - (1 - p_k)^{\gamma K}.$$

Across $R$ rounds, the number of rounds in which client $k$ participates is

$$X_k \sim \text{Binomial}(R, r_k), \qquad \mathbb{E}[X_k] = Rr_k, \quad \text{Var}(X_k) = Rr_k(1 - r_k),$$

where $\text{Binomial}(R, r_k)$ denotes the binomial distribution with $R$ trials and success probability $r_k$.

**Low Coverage Issue.**  The low-coverage issue means that some clients remain uncovered in low-participation settings when the number of communication rounds is not very large and a naive sampling strategy is used.

The probability that a client $k$ is *never* selected in $R$ rounds is

$$\Pr[X_k = 0] = (1 - r_k)^R.$$

For uniform sampling with $p_k = 1/K$ and $K \gg 1$, we have

$$\Pr[X_k = 0] = \left(1 - \frac{1}{K}\right)^{\gamma K R} \approx e^{-\gamma R},$$

where $(1 - \frac{1}{K})^K \approx e^{-1}$ when $K \gg 1$.

Thus, the expected number of unique clients uncovered is

$$N = \mathbb{E}[\text{uncover}] = K\left(\left(1 - \tfrac{1}{K}\right)^{\gamma K R}\right) \approx Ke^{-\gamma R},$$

and the fraction of the expected number of unique clients uncovered is

$$\zeta = N/K \approx e^{-\gamma R}$$

We compute the results in Table 5 using different communication rounds $R$ and client pool sizes $K$. We find that in the high-participation setting, there is no low-coverage issue among clients. However, in the extreme low-participation setting, when the number of communication rounds is not very large ($R \sim 50$), about $8.21\%$ of clients remain uncovered under naive sampling, leading to a low-coverage issue. This demonstrates why many clients have low or no participation if client sampling strategies are not carefully designed.

The case $R \sim 50$ is particularly important because, from Tables 3 and 4, we observe that our method achieves the target when $R \sim 50$. In contrast, from Table 4, non-CR methods (e.g., FedDR) suffer from the low-participation issue and require more communication rounds to converge and achieve the target Top-1 accuracy.

Table 5: Number $N$ and fraction $\zeta$ of uncovered clients under different communication rounds $R$, client pool sizes $K$, and participation levels. Here, $\gamma$ denotes the client participation rate in each communication round.

| R | K | Extreme Low Participation ($\gamma = 5\%$) | | | Low Participation ($\gamma = 10\%$) | | | High Participation ($\gamma = 50\%$) | | |
|---|---|---|---|---|---|---|---|---|---|---|
| | | 50 | 500 | 50000 | 50 | 500 | 50000 | 50 | 500 | 50000 |
| 20 | $N$ | 18 | 184 | 18394 | 7 | 68 | 6767 | 0 | 0 | 2 |
| | $\zeta$ | 36.79% | 36.79% | 36.79% | 13.53% | 13.53% | 13.53% | 0.00% | 0.00% | 0.00% |
| 50 | $N$ | 4 | 41 | 4104 | 0 | 3 | 337 | 0 | 0 | 0 |
| | $\zeta$ | 8.21% | 8.21% | 8.21% | 0.67% | 0.67% | 0.67% | 0.00% | 0.00% | 0.00% |
| 200 | $N$ | 0 | 0 | 2 | 0 | 0 | 0 | 0 | 0 | 0 |
| | $\zeta$ | 0.00% | 0.00% | 0.00% | 0.00% | 0.00% | 0.00% | 0.00% | 0.00% | 0.00% |

**Imbalanced Selection Issue**   Although with large communication round naive sampling strategy mitigate the influence of low coverage issue, such naive strategy also suffer from imbalance selection issue. The imbalanced selection issue means that under naive sampling, some clients are selected far more often while others are rarely selected, leading to unfair participation.

We first use relative dispersion to briefly demonstrate the imbalance selection caused by naive sampling.

Since $X_k$ follows a binomial distribution, the relative dispersion of participation counts across clients is captured by the coefficient of variation

$$\mathrm{CV}(X_k) = \frac{\sqrt{\mathrm{Var}(X_k)}}{\mathbb{E}[X_k]} = \sqrt{\frac{1 - r_k}{Rr_k}} \approx \frac{e^{-\gamma/2}}{\sqrt{R(1 - e^{-\gamma})}} \approx \frac{1}{\sqrt{R\gamma}} \quad (\gamma \ll 1).$$

Thus, unless $R\gamma \gg 1$, participation counts vary widely. A large CV means some clients are selected far more often than average while others are selected far less (or not at all), which is precisely the *unfairness* induced by naive sampling.

Naive sampling yields $\mathrm{CV}(X_k) \approx 1/\sqrt{R\gamma}$, so fairness improves only at a $1/\sqrt{\cdot}$ rate: achieving balanced participation either demands many more rounds or much higher per-round participation—both at odds with low-participation constraints. This is the core of the *imbalanced selection issue* and motivates designs like client reshuffling that explicitly control coverage and balance.

One key point distinguishing the imbalance selection issue from the low coverage issue is the presence of large relative fluctuations among the "covered" clients. Even for clients that are selected at least once, the participation counts can be highly unequal. A standard Chernoff bound shows that, for any $\varepsilon \in (0, 1)$,

$$\Pr\left(\left|X_k - \mathbb{E}[X_k]\right| \geq \varepsilon\mathbb{E}[X_k]\right) \leq 2\exp\left(-\tfrac{\varepsilon^2}{3}\mathbb{E}[X_k]\right) = 2\exp\left(-\tfrac{\varepsilon^2}{3}R\gamma\right). \tag{110}$$

To make relative deviations small with high probability, one needs $R\gamma$ to be large. Table 6 lists the $R\gamma$ requirements for some common choices of relative fairness and probability. For instance, ensuring $\pm 20\%$ relative fairness with 95% probability requires $R\gamma \geq \frac{3}{0.2^2}\ln(40) \approx 277$. However, in low-participation settings this is impractical. For example, if $\gamma R = 0.1 \times 400 = 40$, the condition cannot be met. In advanced FL methods (e.g., SCAFFOLD, FedDC, FedCDR), the number of communication rounds is typically $R \sim 100$, which gives $\gamma R \sim 0.1 \times 100 = 10$. These results demonstrate that under low participation, $\gamma R$ is too small, leading to large relative fluctuations even among the "covered" clients. This unfairness gives rise to two key problems:

- **Biased and high-variance gradients** When some clients are selected disproportionately more often than others, their local updates dominate the aggregation. This induces a sampling bias in the global gradient estimate, especially under heterogeneous data, since over-represented clients skew the optimization trajectory toward their local data distribution. Moreover, the uneven participation amplifies the variance of the aggregated gradient, leading to slower and less stable convergence.

Table 6: $\gamma R$ requirements for common choices of relative fairness and probability. To achieve the given relative fairness and probability, $\gamma R$ should be greater than the value list in the table.

| relative fairness | 95% probability | 90% probability | 75% probability | 50% probability |
|---|---|---|---|---|
| 10% | 1106.7 | 898.7 | 623.8 | 415.9 |
| 20% | 276.7 | 224.7 | 156.0 | 104.0 |
| 30% | 123.0 | 99.9 | 69.3 | 46.2 |
| 50% | 44.3 | 35.9 | 25.0 | 16.6 |

- **Duplicate and wasted participation** In low-participation settings, a small set of clients may be repeatedly sampled across rounds, while many others are rarely or never selected. This redundancy does not provide new information to the server and results in wasted communication and computation resources. Ideally, participation should cover a diverse set of clients, but naive sampling with small $R\gamma$ fails to ensure such coverage.

Table 7: Communication and computation cost per client of different FL methods. The speedup is measured relative to FedAvg. Upload communication cost refers to the parameters sent from the client to the server, while download communication cost refers to the parameters sent from the server to the client.

| | FedAvg | FedProx | SCAFFOLD | FedDC | FedCDR |
|---|---|---|---|---|---|
| Download Comm. cost | 4d | 4d | 8d | 8d | 4d |
| Upload Comm. cost | 4d | 4d | 8d | $8d^2$ | 4d |
| Total Comm. cost | 8d | 8d | 16d | 16d | 8d |
| Comm. Speedup | – | $\times 1.0$ | $\times 0.5$ | $\times 0.5$ | $\times 0.5$ |
| Local Epoch | 5 | 10 | 5 | 5 | 10 |
| Total Comp. cost | 5NG | 10NG | 5NG | 5NG | 10NG |
| Comp. Speedup | – | $\times 0.5$ | $\times 1.0$ | $\times 1.0$ | $\times 0.5$ |
| Total Comm. $\times$ Comp. Speedup | – | $\times 0.5$ | $\times 0.5$ | $\times 0.5$ | $\times 0.5$ |

# E  DETAILED DISCUSSIONS: COMMUNICATION AND COMPUTATION OVERHEAD

We now compare the communication and computation overhead of our method (FedCDR) and the baselines (FedAvg, FedProx, SCAFFOLD, FedDC), arising from the specific design of each algorithm.

**FedAvg.**  FedAvg requires each selected client to perform $H$ local SGD updates per round and send its model parameters (or gradients) to the server. The server then averages the updates and broadcasts the global model. Communication per round involves one upload and one download of model parameters per participating client, while computation cost is proportional to $H$ local gradient evaluations. Thus, FedAvg is the most communication-efficient baseline but can suffer under heterogeneous data.

**FedProx.**  FedProx modifies the local objective with a proximal term $\frac{\mu}{2}\|w - w^t\|^2$ to stabilize updates under heterogeneity. This adds negligible computation overhead since the proximal term only affects gradient calculation. Communication overhead remains identical to FedAvg (upload and download per round). Therefore, FedProx balances stability and efficiency with minimal additional cost.

**SCAFFOLD.**  SCAFFOLD introduces client-side and server-side control variates to correct client drift. Each client must maintain its own control variate $c_k$ and communicate it with the server. In each round, clients send both model updates and control variates, and receive updated control variates from the server. Compared to FedAvg, communication overhead increases due to the extra transmission of control variates. Computation overhead also increases slightly since each local step requires applying control variate corrections.

**FedDC.**  FedDC further reduces client drift by maintaining a global correction term and requiring clients to align their updates accordingly. This design introduces communication of additional correction variables between the server and clients, and local updates require extra computation to apply these corrections. Hence, compared to FedAvg, FedDC has higher communication overhead (model parameters plus correction variables) and negligible computation overhead (gradient correction per local step).

To calculate the communication and computation cost per client, we assume the model parameter $x \in \mathbb{R}^d$ and that the computation cost of evaluating the gradient with one training sample is $G$. We use torch.float32 (4 bytes), and the average number of local training examples is $N$. For FedAvg, SCAFFOLD, and FedDC, we set the local epoch to $E = 5$, while for FedProx and FedCDR, we set the local epoch to $E = 10$. We summarize all results in Table 7.

---

[2]We use the rigorous implementation of FedDC; that is, each client uploads $g_i$ to the server and downloads $g_t$ from the server. By approximating $g_i$, the upload communication cost can be reduced from $8d$ to $4d$.

We find that the computation cost of proximal FL methods (FedProx, FedCDR) is twice that of non-proximal FL methods (FedAvg, SCAFFOLD, FedDC), due to the increased number of local epochs required to solve the proximal subproblem. However, SCAFFOLD and FedDC incur higher communication overhead because of the additional exchange of correction variables. To obtain the total cost, we multiply the communication cost by the computation cost. In terms of total cost, SCAFFOLD, FedDC, FedProx, and FedCDR are nearly the same, making it fair to compare their results under this setting. Moreover, proximal FL methods exhibit a communication-computation trade-off. If the local epoch is fixed at $E = 5$, the same as in non-proximal FL methods, then it is fair to also adjust the communication rounds of FedProx and FedCDR to 800 (while non-proximal FL methods use 400).

# F   DETAILED DISCUSSIONS: THE COMMUNICATION-COMPUTATION TRADE-OFF AND SYSTEMS HETEROGENEITY SIMULATION

We first analyze the communication-computation trade-off in FedCDR. Then we simulate system heterogeneity by varying the fraction of stragglers (0%, 50%, 90%) that perform fewer local epochs than the global target and evaluate our method under this setting.

## F.1   COMMUNICATION-COMPUTATION TRADE-OFF WITH A UNIFORM LOCAL EPOCH

The computation-communication trade-off concerns how the number of local epochs balances convergence speed (in communication rounds) and per-round client workload. With abundant bandwidth but limited computation, fewer local epochs with more communication rounds are preferable. Conversely, when communication is costly but computation is ample, more local epochs with fewer communication rounds save bandwidth. We use a uniform local epoch across all clients, so any change applies to all of the clients.

This trade-off hinges on the implementation of the proximal operator solver. An *exact* solver is unsuitable, as it requires a fixed and often large cost per round (a specially designed solver for the proximal operator). In contrast, an *inexact* solver naturally supports this trade-off. In our experiments, we use SGD as the approximate local solver.

**Theoretical Explanations**   From Theorem 1, the number of communication rounds satisfies

$$K \; := \; \left\lceil \frac{\mathcal{A}_1\big(f(x^0) - f^*\big)}{\epsilon^2} \; + \; \frac{(\mathcal{A}_1\mathcal{B}_1 + \mathcal{A}_1\mathcal{B}_2)\,M}{\epsilon^2} \right\rceil .$$

The amount of local computation determines the evaluation accuracy of the proximal operator in the local solver. In particular, using more local update steps yields higher accuracy (smaller $\epsilon_{t,i,r}$), which reduces the constant $M$ and thereby shrinks the second term above, leading to fewer communication rounds. Conversely, if only a small number of local steps (e.g., two) is permitted, $M$ increases and more communication rounds are required. When communication bandwidth is ample, one may choose smaller local workloads and compensate with additional rounds; our method still applies in this regime.

**Empirical Explanations**   To give the empirical explanations of the communication-computation trade-off in FedCDR, we vary the number of local epochs used by the inexact solver and examine the resulting communication and computation speedups. Table 3 shows the number of communication rounds needed to reach the target accuracy of $98\%$ on MNIST, together with the corresponding speedups. The results indicate that allocating more local epochs (i.e., heavier local computation) reduces communication cost, while fewer local epochs (i.e., lighter local computation) lead to higher communication cost.

## F.2   COMMUNICATION-COMPUTATION TRADE-OFF WITH DIFFERENT LOCAL EPOCHS

The computation–communication trade-off is useful not only in the uniform local epoch setting but also in adapting federated learning systems to heterogeneous communication bandwidths and computational resources. This trade-off becomes particularly important when clients have diverse capabilities and require individually tailored local epochs. In such cases, FedCDR provides a flexible mechanism to balance communication and computation.

A key feature of FedCDR is its ability to handle this trade-off more flexibly than existing baselines. Unlike FedAvg, SCAFFOLD, and FedDC, which fix the number of local epochs across all clients, our method allows clients to adopt different local epochs according to their computational capacity. Specifically, slower clients are assigned fewer local epochs (or are early-stopped), while faster clients perform more local work per round. Under such heterogeneous settings, many existing baselines suffer from degraded performance or even divergence due to inconsistent client progress. In contrast, FedCDR maintains stable updates, making it more robust to stragglers and device variability (*system heterogeneity*), which are common in realistic federated learning environments.

Table 8: Comparison of methods under different straggler fractions in two data heterogeneity settings. The values represent the number of communication rounds required for each method to achieve a target top-1 accuracy of $98\%$ on MNIST.

| Methods | $\alpha = 0.1$ | | | $\alpha = 1.0$ | | |
|---|---|---|---|---|---|---|
| | 0% | 50% | 90% | 0% | 50% | 90% |
| CR+FedAvg | 386 | $> 400$ | $> 400$ | 319 | $> 400$ | $> 400$ |
| CR+SCAFFOLD | 85 | 248 | $> 400$ | 75 | 136 | $> 400$ |
| CR+FedDC | 78 | 212 | $> 400$ | 57 | 94 | 379 |
| FedCDR | 60 | 82 | 116 | 49 | 76 | 104 |

## F.3 SYSTEM HETEROGENEITY SIMULATION

To simulate system heterogeneity (stragglers and device variability), we follow the experimental setup in Li et al. (2020a). We assume a global clock during training, where each participating device determines the amount of local work it can perform based on this clock cycle and its system constraints. In our simulations, we fix the maximum number of local epochs $E$ and force some devices to perform fewer than $E$ epochs of updates according to their constraints. Specifically, under different heterogeneous settings, at each round we assign $x$ epochs ($x$ is chosen uniformly at random from $\{1, 2, \cdots, E\}$) to $0\%$, $50\%$, or $90\%$ of the selected clients, respectively. The case where $0\%$ of clients perform fewer than $E$ epochs corresponds to a homogeneous system environment, while the case where $90\%$ of clients send partial updates corresponds to a highly heterogeneous system environment. Existing baselines simply drop these $0\%$, $50\%$, and $90\%$ stragglers once the global clock cycle ends, whereas FedCDR incorporates their partial updates.

Table 8 shows the impact of stragglers on convergence under two data heterogeneity settings. We observe that baseline methods degrade significantly as the straggler fraction increases, often failing to converge within 400 rounds. This is because they simply drop updates from more constrained clients once the global clock cycle ends, which leads to wasted computation and slower convergence. In contrast, FedCDR incorporates partial updates from stragglers and thus remains robust even when up to 90% of clients are stragglers. By leveraging clients with varying local epochs (i.e., different computation capabilities), FedCDR achieves more stable training and substantially faster convergence. Moreover, the degradation of baseline methods becomes more pronounced as the degree of data heterogeneity increases. This highlights the importance of leveraging clients' partial updates in highly heterogeneous settings, since such scenarios are more challenging and require broader data coverage. In such settings, clients hold data with highly diverse and often non-overlapping distributions, making global model training substantially more difficult. Leveraging clients' partial updates therefore becomes especially important, as it enables the aggregation process to incorporate a wider variety of data sources per round, thereby improving coverage of the heterogeneous data landscape and mitigating the adverse effects of skewed local distributions.

# G  EXPERIMENT DETAILS

## G.1  DETAILS OF DATASETS AND HETEROGENEOUS PARTITION STRATEGIES

### G.1.1  SYNTHETIC DATASETS

Synthetic datasets (Shamir et al., 2014) are artificially generated datasets in which all parameters can be explicitly controlled. We consider a simple 10-class classification problem:

$$y = \arg\max\left(\text{softmax}(Wx + b)\right), \tag{111}$$

where $x \in \mathbb{R}^{60}$ is the input feature vector, and $y$ is the predicted class label. The model parameters $W \in \mathbb{R}^{10\times 60}$ and $b \in \mathbb{R}^{10}$ represent the weight matrix and bias vector, respectively, and are fully controllable.

To adapt this dataset for federated learning and to introduce data heterogeneity, we adopt the heterogeneous partitioning strategy based on client-specific mean shift (Li et al., 2020a). Specifically, for each client $k$, we generate samples $(x, y)$ using a personalized model:

$$y = \arg\max\left(\text{softmax}(W_k x + b_k)\right),$$

where $W_k$ and $b_k$ denote the weight matrix and bias vector specific to client $k$. In the homogeneous setting, all clients share the same model, i.e., $W_1 = W_2 = \cdots = W_K = W$.

To simulate client model heterogeneity, we assume that each client has a distinct model. Specifically, we introduce variability by sampling the model parameters $W_k$ and $b_k$ from normal distributions with client-specific means:

$$W_k - W \sim \mathcal{N}(u_k, 1), \quad b_k - b \sim \mathcal{N}(u_k, 1),$$

where the client-specific mean $u_k$ is itself drawn from a higher-level distribution:

$$u_k \sim \mathcal{N}(0, \alpha).$$

Therefore, the hyperparameter $\alpha$ controls the variation of parameters between local models: smaller values of $\alpha$ correspond to more homogeneous models, while larger values introduce greater heterogeneity.

For each client $k$, we also generate local data pairs $(x, y)$. Under the independent and identically distributed (i.i.d.) setting, feature vectors are sampled from a shared Gaussian distribution:

$$x \sim \mathcal{N}(0, \Sigma),$$

where the covariance matrix $\Sigma$ is diagonal with entries $\Sigma_{j,j} = j^{-1.2}$.

To simulate data heterogeneity across clients, we assume that each client has its own distinct data distribution. Specifically, we introduce a client-specific shift in the feature distribution by sampling $x$ as:

$$x \sim \mathcal{N}(v_k, \Sigma), \quad v_k \sim \mathcal{N}(B_k, 1), \quad B_k \sim \mathcal{N}(0, \beta),$$

where $v_k$ represents a client-specific shift vector and is itself drawn from a higher-level distribution centered at $B_k$.

Therefore, the hyperparameter $\beta$ determines the degree of divergence between the local training data distributions across clients: smaller values correspond to more similar training data distributions, while larger values lead to greater divergence among the training data of clients.

To evaluate the performance of different methods under different degrees of data heterogeneity, we generate multiple synthetic datasets using various combinations of the heterogeneity parameters $(\alpha, \beta)$, denoted as Synthetic-$(\alpha, \beta)$. In our experiments, we consider three levels of heterogeneity: Synthetic-(0,0), Synthetic-(1,1), and Synthetic-(5,5). Each dataset consists of 500 clients.

The number of samples per client is drawn from a log-normal distribution:

$$n_k \sim \text{LogNormal}(\mu = 2, \sigma = 2) + 10,$$

where $\mu$ and $\sigma$ denote the mean and standard deviation of the underlying normal distribution, respectively. The log-normal distribution introduces additional variability in client data sizes, better capturing the non-uniformity observed in practical federated learning systems. To prevent extreme cases, we clip the maximum number of samples per client to 50.

### G.1.2 BENCHMARK DATASETS

We evaluate our method on three widely used benchmark datasets in federated learning: MNIST, CIFAR-10, and CIFAR-100 (Krizhevsky, 2009). The MNIST dataset is a widely used handwritten digit recognition benchmark. It consists of 10 classes (digits 0–9), with grayscale images of size $28 \times 28$. The dataset contains a total of 70,000 samples, including 60,000 training images and 10,000 test images. CIFAR-10 and CIFAR-100 are image classification datasets consisting of color images of size $32 \times 32$, with 10 and 100 classes, respectively. CIFAR-10 contains 50,000 training images and 10,000 test images, covering common object categories such as airplanes, automobiles, and animals. CIFAR-100 also has 50,000 training images and 10,000 test images, but with a more fine-grained label space organized into 100 classes grouped under 20 superclasses.

We adopt the Dirichlet-based partitioning scheme (Wang et al., 2020) to simulate heterogeneous client data distributions for all datasets. Specifically, we partition each dataset across clients using a Dirichlet-based distribution over class labels. In this setting, both the number of data points and the class proportions are imbalanced across clients. Specifically, we simulate a heterogeneous partition into $J$ clients by drawing class proportions from a Dirichlet distribution:

$$(p_{1,j}, p_{2,j}, \ldots, p_{C,j}) \sim \mathrm{Dir}(\alpha_1, \alpha_2, \ldots, \alpha_C), \tag{112}$$

where $p_{c,j}$ denotes the proportion of training instances of class $c$ assigned to client $j$, and $C$ is the total number of classes. The distribution $\mathrm{Dir}(\cdot)$ is the $C$-dimensional Dirichlet distribution. We set $\alpha_1 = \alpha_2 = \cdots = \alpha_C = \alpha$ to induce heterogeneity.

As a property of the Dirichlet distribution, when $\alpha_c < 1$, the sampled class proportions tend to concentrate near the corners and edges of the probability simplex. This leads to clients receiving data from only a few dominant classes, thereby simulating severe label imbalance and statistical heterogeneity—common characteristics of practical federated learning environments.

Based on the sampled proportions $\{p_{c,j}\}$, we allocate the training data to each client accordingly. For evaluation, we use the original test set from each dataset as a global test set to ensure a fair comparison across all methods.

Regarding the number of clients, we use 500 clients for MNIST, CIFAR-10, and CIFAR-100. To further simulate practical federated learning scenarios, we introduce sample size imbalance across clients. Specifically, for each dataset, the number of training samples per client $N$ is drawn from a log-normal distribution:

$$N \sim \mathrm{LogNormal}(\mu = 4,\ \sigma = 2) + 30,$$

where $\mu$ and $\sigma$ are the mean and standard deviation of the underlying normal distribution, respectively. The log-normal sampling introduces natural variability in client data volume. To avoid extreme outliers, we clip the number of samples per client to 500.

### G.2 DETAILS OF MODEL ARCHITECTURES AND TRAINING SCHEMES

#### G.2.1 RUNTIME ENVIRONMENT

We conduct our experiments on a Linux-based server running Ubuntu 20.04. Each experiment is executed on a compute node equipped with a 24-core 2.80 GHz Intel processor, an NVIDIA RTX 3090 GPU, and 100 GB of virtual memory allocated to the Python interpreter.

#### G.2.2 MODEL ARCHITECTURES

For all synthetic datasets, we use a fully connected neural network with architecture $60 \times 32 \times 10$, where the dimensions correspond to the input size, hidden layer size, and output size, respectively.

Table 9: Architecture of ResNet-20 with Group Normalization (GN) for CIFAR-10/CIFAR-100

| Stage | Output Size | Layers |
|-------|-------------|--------|
| Input | $32 \times 32$ | $3 \times 3$ conv, 16 filters, stride 1 |
| Stage 1 | $32 \times 32$ | $3 \times [\, 3 \times 3$ conv, 16 filters $+ \mathrm{GN}(G = 4) + \mathrm{ReLU}\,]$ |
| Stage 2 | $16 \times 16$ | $3 \times [\, 3 \times 3$ conv, 32 filters $+ \mathrm{GN}(G = 8) + \mathrm{ReLU}\,]$ |
| Stage 3 | $8 \times 8$ | $3 \times [\, 3 \times 3$ conv, 64 filters $+ \mathrm{GN}(G = 8) + \mathrm{ReLU}\,]$ |
| Output | $1 \times 1$ | Global average pooling, FC-10/FC-100, softmax |

For the MNIST dataset, we adopt a fully connected neural network of size $784 \times 128 \times 10$, following the architecture used in Li et al. (2020b). The input size reflects the flattened $28 \times 28$ grayscale images, and the output corresponds to the 10 classes.

For the CIFAR-10 and CIFAR-100 datasets, we employ ResNet-20 (He et al., 2016) as the backbone model for image classification. The batch normalization (BN) layers in ResNet-20 are replaced with group normalization (GN) layers, since BN performs poorly with heterogeneous data across workers (Wu & He, 2018). Table 9 presents the detailed architecture of ResNet-20 with group normalization.

### G.2.3 LOCAL EPOCH

For non-proximal methods (FedAvg, SCAFFOLD, FedDC), we set the local epochs of each client to 5. For proximal methods (FedProx, FedCDR), we set the local epochs to 10 to obtain an inexact solution to the proximal subproblem. A discussion of the communication and computation overhead caused by these methods and settings is provided in Appendix E.

### G.2.4 HYPERPARAMETERS

For each dataset, we tune and select the best-performing hyperparameters based on validation performance and report results using these selected values.

**Batch Size** For the synthetic dataset, we use a batch size of 16 for each local client. For the benchmark datasets (MNIST, CIFAR-10, CIFAR-100), we use a batch size of 32.

**Weight Decay** We use L2 weight decay, i.e., and apply SGD to the cross-entropy loss with L2 regularizer $\frac{\mu}{2}\|w\|_2^2$ of the model parameter $w \in \mathbb{R}^d$. In our experiments, we set $\mu = 5 \times 10^{-4}$.

**Client Learning Rate** The client learning rate $\gamma_0$ is selected from $\{0.01, 0.02, 0.05, 0.1\}$. We use an epochwise learning-rate schedule, defined in (113).

$$
\gamma_t = \begin{cases} \gamma_0, & 0 \leq t < \frac{T}{2}, \\ 0.1\,\gamma_0, & \frac{T}{2} \leq t < \frac{3T}{4}, \\ 0.01\,\gamma_0, & \frac{3T}{4} \leq t \leq T, \end{cases} \tag{113}
$$

where $T$ denotes the total number of communication rounds and $t$ denotes the current round.

**Optimizer and Momentum** We use stochastic gradient descent (SGD) with momentum ($\beta = 0.9$).

**Hyperparameters of FedProx** FedProx has a unique hyperparameter, the proximal coefficient $\mu$. We select $\mu$ from $\{10^{-5}, 10^{-4}, 10^{-3}, 10^{-2}\}$.

**Hyperparameters of FedDC** FedDC has a unique hyperparameter, the drift-regularization weight $\alpha$. We select $\alpha$ from $\{0.03, 0.1, 0.3\}$ on the Synthetic and MNIST datasets, and from $\{0.003, 0.01, 0.03\}$ on CIFAR-10 and CIFAR-100.

**Hyperparameters of FedCDR** FedCDR has two unique hyperparameters: the proximal coefficient $\eta$ and the relaxation coefficient $\alpha$. We select $\eta$ from $\{10, 10^2, 10^3, 10^4\}$ and $\alpha$ from $\{0.5, 1.0, 1.5, 1.99\}$.

### G.2.5 PREPROCESS

For preprocessing the images in the CIFAR-10 and CIFAR-100 datasets, we follow the standard data augmentation and normalization procedures. Specifically, we apply random cropping and horizontal flipping for data augmentation. For normalization, each color channel is standardized by subtracting the mean and dividing by the standard deviation of that channel. The channel-wise normalization statistics are as follows: $\mu_r = 0.4914$, $\mu_g = 0.4824$, $\mu_b = 0.4467$, and $\sigma_r = 0.2471$, $\sigma_g = 0.2435$, $\sigma_b = 0.2616$. This ensures that the input images have zero mean and unit variance per channel, which is a common practice to stabilize and accelerate training.

Table 10: Top-1 accuracy on synthetic datasets with additional baselines. Reported values are the mean ± standard deviation.

| Methods | Synthetic-(0,0) | Synthetic-(1,1) | Synthetic-(5,5) |
|---|---|---|---|
| FedAvg | 86.23 ± 0.32 | 74.73 ± 0.35 | 41.89 ± 1.15 |
| FedProx | 86.37 ± 0.45 | 77.55 ± 0.53 | 62.82 ± 0.71 |
| SCAFFOLD | 89.71 ± 0.21 | 88.14 ± 0.20 | 78.27 ± 0.29 |
| FedDC | 91.38 ± 0.17 | 90.20 ± 0.19 | 79.88 ± 0.24 |
| FedDR | 91.22 ± 0.37 | 91.78 ± 0.34 | 80.12 ± 0.37 |
| CR+FedAvg | 88.12 ± 0.21 | 77.54 ± 0.42 | 46.15 ± 1.32 |
| CR+FedProx | 88.45 ± 0.15 | 80.34 ± 0.24 | 65.80 ± 0.45 |
| CR+SCAFFOLD | 91.34 ± 0.25 | 88.15 ± 0.19 | 77.92 ± 0.32 |
| CR+FedDC | 92.25 ± 0.14 | 90.04 ± 0.21 | 79.78 ± 0.26 |
| FedCDR | **93.00 ± 0.24** | **92.02 ± 0.17** | **85.76 ± 0.37** |

Table 11: Top-1 accuracy on benchmark datasets with additional baselines. Reported values are the mean ± standard deviation.

| Methods | $\alpha = 0.1$ | | | $\alpha = 1.0$ | | |
|---|---|---|---|---|---|---|
| | MNIST | CIFAR10 | CIFAR100 | MNIST | CIFAR10 | CIFAR100 |
| FedAvg | 97.02 ± 0.12 | 67.89 ± 0.46 | 53.45 ± 0.56 | 97.56 ± 0.05 | 78.19 ± 0.47 | 64.88 ± 0.42 |
| FedProx | 97.17 ± 0.08 | 68.87 ± 0.52 | 55.97 ± 0.70 | 97.82 ± 0.04 | 79.00 ± 0.29 | 65.18 ± 0.45 |
| SCAFFOLD | 97.90 ± 0.06 | 73.12 ± 0.42 | 61.69 ± 0.35 | 98.10 ± 0.07 | 82.39 ± 0.35 | 68.72 ± 0.25 |
| FedDC | 98.04 ± 0.03 | 75.38 ± 0.30 | 64.80 ± 0.35 | 98.10 ± 0.03 | 83.40 ± 0.27 | 69.04 ± 0.36 |
| FedDR | 98.00 ± 0.05 | 75.24 ± 0.29 | 65.34 ± 0.34 | 98.05 ± 0.03 | 83.30 ± 0.28 | 69.00 ± 0.38 |
| CR+FedAvg | 97.45 ± 0.14 | 69.79 ± 0.70 | 55.68 ± 0.70 | 97.79 ± 0.07 | 78.29 ± 0.50 | 65.89 ± 0.41 |
| CR+FedProx | 97.68 ± 0.09 | 70.34 ± 0.56 | 58.34 ± 0.62 | 97.72 ± 0.05 | 79.34 ± 0.34 | 65.45 ± 0.43 |
| CR+SCAFFOLD | 98.06 ± 0.05 | 74.45 ± 0.40 | 63.47 ± 0.34 | 98.02 ± 0.06 | 82.35 ± 0.30 | 69.67 ± 0.25 |
| CR+FedDC | 98.00 ± 0.03 | 75.24 ± 0.25 | 64.56 ± 0.35 | 98.29 ± 0.03 | 83.56 ± 0.24 | 69.28 ± 0.34 |
| FedCDR | **98.17 ± 0.05** | **76.98 ± 0.31** | **67.89 ± 0.32** | **98.40 ± 0.04** | **85.29 ± 0.26** | **71.90 ± 0.28** |

## H  ADDITIONAL EXPERIMENTS FOR NON-CLIENT-RESHUFFLING BASELINES

To isolate the effect of the reshuffling protocol itself, we conduct additional experiments on baselines without client reshuffling, in line with prior work. We use FedAvg, FedProx, SCAFFOLD, FedDC, and FedDR as baselines and follow their original implementations. For each baseline, we also implement a client-reshuffling variant (denoted by the "CR+" prefix), and we further include our proposed FedCDR, which couples DR splitting with client reshuffling.

As shown in Tables 10, 11, and 12, we observe three main trends. First, client reshuffling consistently improves or at least maintains performance across all baselines: for FedAvg and FedProx the gains are clear on both synthetic and real datasets, while for stronger baselines such as SCAFFOLD and FedDC the differences are smaller but mostly positive and always within one standard deviation. This confirms that client reshuffling is a generally beneficial and safe enhancement on top of existing FL methods.

Second, DR-splitting–based methods benefit more from client reshuffling than traditional FedAvg-type schemes. On benchmark datasets, the improvement from FedDR to FedCDR is substantially larger than the improvement from FedDC to CR+FedDC, especially on the more heterogeneous and challenging settings (e.g., CIFAR10 and CIFAR100 with $\alpha = 0.1$, as well as the 10,000-client experiments). This indicates that DR splitting and client reshuffling are highly synergistic: reshuffling not only stabilizes DR splitting under non-IID data, but also unlocks significant accuracy gains.

Third, without client reshuffling, FedDR is comparable to (and sometimes slightly worse than) FedDC across datasets. However, after combining DR splitting with client reshuffling, FedCDR consistently outperforms even the strongest baselines with client reshuffling, including CR+FedDC. On both synthetic and benchmark datasets (and in the 10,000-client regime), FedCDR achieves the highest top-1 accuracies in almost all settings, often by a non-trivial margin over CR+FedDC. These results corroborate our theoretical findings: while client reshuffling provides a general improvement for many FL methods, it is particularly effective when coupled with DR splitting, and FedCDR leverages this combination to achieve state-of-the-art performance under severe data heterogeneity and large-scale participation.

Table 12: Top-1 accuracies on benchmark datasets with 10,000 clients. Reported values are mean ± standard deviation.

| Methods | $\alpha = 0.1$ | | | $\alpha = 1.0$ | | |
|---|---|---|---|---|---|---|
| | MNIST | CIFAR10 | CIFAR100 | MNIST | CIFAR10 | CIFAR100 |
| FedAvg | $96.06 \pm 0.20$ | $63.70 \pm 0.97$ | $52.03 \pm 1.12$ | $97.02 \pm 0.18$ | $75.67 \pm 0.78$ | $62.97 \pm 1.00$ |
| FedProx | $96.38 \pm 0.17$ | $65.60 \pm 0.93$ | $54.73 \pm 0.98$ | $97.13 \pm 0.13$ | $76.30 \pm 0.68$ | $63.78 \pm 0.94$ |
| SCAFFOLD | $96.90 \pm 0.16$ | $70.34 \pm 0.77$ | $62.67 \pm 0.80$ | $97.99 \pm 0.12$ | $80.05 \pm 0.63$ | $64.89 \pm 0.73$ |
| FedDC | $97.08 \pm 0.12$ | $71.54 \pm 0.64$ | $62.80 \pm 0.76$ | $97.89 \pm 0.09$ | $81.92 \pm 0.54$ | $65.35 \pm 0.68$ |
| FedDR | $97.00 \pm 0.16$ | $70.93 \pm 0.70$ | $62.34 \pm 0.85$ | $97.56 \pm 0.10$ | $81.30 \pm 0.61$ | $65.00 \pm 0.78$ |
| CR+FedAvg | $97.37 \pm 0.16$ | $65.25 \pm 0.75$ | $54.68 \pm 0.86$ | $97.48 \pm 0.12$ | $76.21 \pm 0.46$ | $63.24 \pm 0.82$ |
| CR+FedProx | $97.46 \pm 0.12$ | $66.37 \pm 0.73$ | $56.77 \pm 0.68$ | $97.57 \pm 0.09$ | $77.87 \pm 0.42$ | $64.07 \pm 0.74$ |
| CR+SCAFFOLD | $97.36 \pm 0.09$ | $72.98 \pm 0.60$ | $63.27 \pm 0.51$ | $98.00 \pm 0.08$ | $81.99 \pm 0.38$ | $68.37 \pm 0.61$ |
| CR+FedDC | $97.57 \pm 0.08$ | $73.16 \pm 0.46$ | $63.70 \pm 0.45$ | $98.03 \pm 0.06$ | $82.46 \pm 0.24$ | $68.88 \pm 0.50$ |
| FedCDR | $\mathbf{98.00 \pm 0.07}$ | $\mathbf{75.02 \pm 0.43}$ | $\mathbf{65.99 \pm 0.48}$ | $\mathbf{98.32 \pm 0.06}$ | $\mathbf{84.78 \pm 0.32}$ | $\mathbf{70.98 \pm 0.46}$ |

# I  ADDITIONAL EXPERIMENTS FOR SCALABILITY

To evaluate the scalability of our method, we further conduct experiments on a large-scale setting with 10,000 clients, as reported in Table 12. In each communication round, only $5\%$ of clients are sampled to participate, which mimics realistic large-scale FL. All other experimental configurations (models, total communication rounds, local epochs, and data partitioning via the Dirichlet distribution) are kept consistent with the 100-client experiments, and we re-tune the learning rates and regularization parameters for each method.

From Table 12, we highlight two key observations.

First, client reshuffling consistently reduces variance and improves robustness. For all baselines, the client-reshuffling variants (CR+FedAvg, CR+FedProx, CR+SCAFFOLD, CR+FedDC) achieve higher or comparable mean accuracy and typically smaller standard deviations compared to their non-reshuffling counterparts. This effect is particularly visible in the more heterogeneous case $\alpha = 0.1$ and on the harder CIFAR10/CIFAR100 datasets. These results confirm that client reshuffling not only improves average performance but also stabilizes training in large-scale, highly non-IID settings.

Second, DR splitting benefits more from client reshuffling, and FedCDR scales better than strong baselines. Without client reshuffling, FedDR is comparable to (and sometimes slightly better than) FedDC in the 10,000-client regime. However, once combined with client reshuffling, our FedCDR method achieves the best performance across all datasets and heterogeneity levels, with a noticeably larger margin over CR+FedDC than in the 100-client experiments (cf. Table 11). For example, on CIFAR10 with $\alpha = 0.1$, FedCDR improves over CR+FedDC by roughly 2 percentage points, and on CIFAR100 with $\alpha = 0.1$, the gap is above 2 percentage points. This indicates that DR splitting and client reshuffling are highly complementary: as the number of clients grows, FedCDR not only maintains stability but also amplifies its advantage over strong non-reshuffling and reshuffling baselines.

Table 13: Communication complexity of representative FL algorithms for reaching $\mathbb{E}\|\nabla f(x)\|^2 \leq \epsilon$ in the nonconvex setting. Here $\tilde{\mathcal{O}}(\cdot)$ hides logarithmic factors and problem-dependent constants.

| Method | Assumptions | Communication complexity and heterogeneity dependence |
|---|---|---|
| FedProx | $L$-smooth nonconvex local objective functions; bounded local dissimilarity $B_\epsilon$ or bounded gradient variance; possibly inexact local solves; partial participation. | $\tilde{\mathcal{O}}(1/\epsilon)$ communication rounds to reach $\mathbb{E}\|\nabla f(x)\|^2 \leq \epsilon$; the leading constant scales polynomially with the dissimilarity parameter $B_\epsilon$ and the number of local steps (Li et al., 2020a), so the rate **deteriorates as data heterogeneity increases** and does not yield an explicit linear speedup in $K$ and $S$. |
| SCAFFOLD | $L$-smooth local objectives; bounded-variance stochastic gradients; no similarity / dissimilarity assumptions on clients; partial participation with $S$ active clients and $K$ local steps per round. | From Theorem I in Karimireddy et al. (2020), the nonconvex rate is $R = \tilde{\mathcal{O}}(\frac{\sigma^2}{KS\epsilon^2} + \frac{(N/S)^{2/3}}{\epsilon})$ communication rounds to reach $\mathbb{E}\|\nabla f(x)\|^2 \leq \epsilon$. It has $\tilde{\mathcal{O}}(1/\epsilon^2)$ **dependence on** $\epsilon$, but enjoys (near-)linear speedup in the effective batch size $KS$ and its bound is **independent of gradient-dissimilarity parameters** such as $G$ or $B_\epsilon$. |
| FedDC | Nonconvex, $L$-smooth local objectives; $B$-local dissimilarity; bounded variance; partial participation with $C$ active clients per round. | Theorem 1 in the supplementary material of Gao et al. (2022) shows that in each round $L(w_t)$ decreases by at least $2p\|\nabla L(w_{t-1})\|^2$ with $p > 0$ depending on $B$, $C$, $\beta$, and the variance. This implies $\tilde{\mathcal{O}}(1/(p\epsilon))$ communication rounds to reach $\mathbb{E}\|\nabla f(x)\|^2 \leq \epsilon$, where $1/p$ **grows with the dissimilarity level** $B$ **and the ratio** $N/C$; i.e., the rate is asymptotically $\tilde{\mathcal{O}}(1/\epsilon)$ but with a **heterogeneity- and participation-dependent constant**. |
| **FedCDR** | Nonconvex, $L$-smooth local objectives; partial participation with $C$ active clients per round; client reshuffling across communication rounds; possibly inexact local proximal solves. | $\tilde{\mathcal{O}}(1/\epsilon)$ communication rounds to reach $\mathbb{E}\|\nabla f(x)\|^2 \leq \epsilon$; in contrast to FedProx and FedDC, the leading constant in our bound is **independent of any data-heterogeneity measure** and **does not blow up with the participation ratio** $N/C$. |

## J  COMPARISON OF NONCONVEX FL CONVERGENCE GUARANTEES

### J.1  ALIGNMENT OF ACCURACY CRITERIA

In the nonconvex setting, different works adopt slightly different accuracy criteria when reporting communication complexity. Our main convergence result is stated in terms of

$$\mathbb{E}\big[\|\nabla f(x_{\text{out}})\|^2\big] \leq \epsilon^2,$$

that is, we bound the squared gradient norm and use $\epsilon^2$ as the target tolerance. In contrast, many federated learning papers, including FedProx (Li et al., 2020a), SCAFFOLD (Karimireddy et al., 2020), and FedDC (Gao et al., 2022), use the criterion

$$\mathbb{E}\big[\|\nabla f(x_{\text{out}})\|^2\big] \leq \epsilon.$$

These two conventions are equivalent up to a reparameterization of $\epsilon$: if we rewrite our bound using a parameter $\tilde{\epsilon}$ defined by $\tilde{\epsilon} = \epsilon^2$, then a rate of $\tilde{\mathcal{O}}(1/\epsilon)$ under the first convention corresponds to $\tilde{\mathcal{O}}(1/\tilde{\epsilon}^{1/2})$ under the second, and vice versa. For a fair and transparent comparison, we therefore adopt the aligned criterion

$$\mathbb{E}\big[\|\nabla f(x_{\text{out}})\|^2\big] \leq \epsilon.$$

### J.2 COMPARISON OF METHODS FOR HANDLING DATA HETEROGENEITY

As summarized in Table 13, FedProx (Li et al., 2020a) attains a $\tilde{\mathcal{O}}(1/\epsilon)$ rate under a bounded local dissimilarity condition $B_\epsilon$, but the hidden constant depends polynomially on $B_\epsilon$ and on the number of local steps. Consequently, the convergence rate degrades as the data become more heterogeneous, and the theory does not yield a clean, explicit linear speedup in the number of local updates and participating clients.

SCAFFOLD (Karimireddy et al., 2020) achieves a convergence rate of order $\tilde{\mathcal{O}}\big(\frac{\sigma^2}{KS\epsilon^2} + \frac{(N/S)^{2/3}}{\epsilon}\big)$ under standard smoothness and bounded-variance assumptions, without any similarity or dissimilarity condition across clients. This bound explicitly exhibits (near-)linear speedup in the effective batch size $KS$, and it is robust to non-IID data in the sense that no gradient-dissimilarity parameter (such as $G$ or $B_\epsilon$) appears in the leading constant. However, the dependence on the target accuracy is $\tilde{\mathcal{O}}(1/\epsilon^2)$, which is asymptotically worse than $\tilde{\mathcal{O}}(1/\epsilon)$ when $\epsilon$ is small.

FedDC (Gao et al., 2022) also enjoys an $\tilde{\mathcal{O}}(1/\epsilon)$-type rate, but its analysis relies on a $B$-local dissimilarity assumption and yields a contraction factor $p > 0$ that depends on $B$, the number of active clients $C$, the smoothness constant $\beta$, and the variance. The resulting complexity is $\tilde{\mathcal{O}}(1/(p\epsilon))$, whose hidden constant grows with both the heterogeneity level $B$ and the participation ratio $N/C$. Thus, although FedDC matches the $\tilde{\mathcal{O}}(1/\epsilon)$ dependence on $\epsilon$, its guarantee deteriorates when the data become highly non-IID or when only a small fraction of clients participate in each round.

In contrast, FedCDR attains $\tilde{\mathcal{O}}(1/\epsilon)$ communication complexity under standard smoothness with partial participation and possibly inexact local proximal solves. Importantly, the leading constant in our bound is *independent* of any data-heterogeneity measure such as $G$, $B_\epsilon$, or $B$, and it does not worsen with the participation ratio $N/C$.

Therefore, FedCDR not only employs client reshuffling to meet practical requirements, but also achieves a state-of-the-art convergence rate, providing a nonconvex FL algorithm that simultaneously attains the best-known $\tilde{\mathcal{O}}(1/\epsilon)$ rate and a heterogeneity-insensitive convergence guarantee. Our method closes the gap between heterogeneity-robust methods like SCAFFOLD (which pay a $\tilde{\mathcal{O}}(1/\epsilon^2)$ price in accuracy) and $\tilde{\mathcal{O}}(1/\epsilon)$ methods like FedProx and FedDC (whose constants can explode with data heterogeneity and partial participation). Moreover, our method uses inexact local solvers, can adapt to stragglers, and is applicable to more complex practical environments.

### J.3 COMPARISON OF CLIENT-RESHUFFLING-BASED METHODS

As summarized in Table 14, RR–CLI focuses on convex and strongly convex objectives and achieves an accelerated $\tilde{\mathcal{O}}(1/T^2)$ rate in the number of meta-epochs $T$, but its analysis does not extend to general nonconvex objectives, which are standard in modern FL applications. Thus, RR–CLI is not directly comparable to our setting.

Clipped RR–CLI extends the reshuffling framework to nonconvex problems under generalized $(L_0, L_1)$-smoothness and obtains an $\tilde{\mathcal{O}}(1/\epsilon)$ rate for a gradient-type stationarity measure (Demidovich et al., 2024, Theorem 4 and Corollary 4). Specifically, Theorem 4 states that for any $T \geq 1$ and stepsizes $\{\gamma_t, \eta_t, \theta_t\}$ satisfying $\zeta/\hat{a}_t \leq \theta_t \leq 1/(4\hat{a}_t)$, the iterates of Clipped RR–CLI satisfy

$$\mathbb{E}\left[\min_{0 \leq t \leq T-1} \frac{\zeta}{8} \min\left(\frac{\|\nabla f(x_t)\|^2}{L_0}, \frac{\|\nabla f(x_t)\|}{L_1}\right)\right] \leq \frac{A_T}{T} \delta_0 + K_t(\eta_t^2 a_t \Delta_\star + \gamma_t^2 N\tilde{a}_t \Delta^\star + \eta_t^2 R a_t \Delta_\star), \tag{114}$$

where

$$A_T = \left(1 + K_t\left(\eta_t^2 a_t + \eta_t^2 R^2 \hat{a}_t + \gamma_t^2 N\tilde{a}_t + \eta_t^2 R a_t\right)\right)^T, \qquad K_t = 2\hat{a}_t \tilde{a}_t^2 + \frac{\hat{a}_t^3}{4\hat{a}_t^2},$$

and

$$\hat{a}_t = L_0 + L_1\|\nabla f(x_t)\|, \quad a_t = L_0 + L_1 \max_m \|\nabla f_m(x_t)\|, \quad \tilde{a}_t = L_0 + L_1 \max_{m,j} \|\nabla f_{mj}(x_t)\|.$$

The terms $\Delta_\star$ and $\Delta^\star$ are heterogeneity measures that quantify the mismatch between local and global minima and enter *explicitly* in the second term on the right-hand side of equation (114). To

Table 14: Comparison of convergence guarantees for client-reshuffling–based FL methods and Fed-CDR. Here $\tilde{\mathcal{O}}(\cdot)$ hides logarithmic factors and problem-dependent constants.

| Method | Assumptions | Convergence rate and heterogeneity dependence |
|---|---|---|
| RR–CLI | Convex or $\mu$-strongly convex and $L$-smooth local losses; partial participation via cohorts of size $C$. | In the $\mu$-strongly convex case, $f(x_T) - f^*$ decays linearly and yields an $\tilde{\mathcal{O}}(1/T^2)$ rate in terms of meta-epochs $T$ (Malinovsky et al., 2023a). The statistical term depends on variance quantities at $x^*$ that implicitly capture heterogeneity. The analysis is limited to (strongly) convex objectives and does not cover general nonconvex $f$. |
| Clipped RR–CLI | Nonconvex objectives with generalized asymmetric or symmetric $(L_0, L_1)$-smoothness; client reshuffling with partial participation via cohorts of size $C$. | For generalized-smooth nonconvex objectives, Theorem 4 and Corollary 4 give $T = \tilde{\mathcal{O}}(1/\epsilon)$ meta-epochs to reach a gradient-type criterion (Demidovich et al., 2024). The constants depend on heterogeneity measures $\Delta_\star, \Delta^\star$ and enter both the bound and stepsize conditions; when heterogeneity is large, they force extremely small learning rates or may fail to guarantee convergence. |
| **FedCDR** | Nonconvex, $L$-smooth local objectives; bounded-variance stochastic gradients; partial participation via cohorts of size $C$; client reshuffling across communication rounds; possibly inexact local proximal solves. | FedCDR requires $\tilde{\mathcal{O}}(1/\epsilon)$ communication rounds to reach $\mathbb{E}\|\nabla f(x_{\text{out}})\|^2 \leq \epsilon$. In contrast to Clipped RR–CLI, whose constants depend on heterogeneity measures such as $\Delta_\star, \Delta^\star$, the leading constant in our rate is **independent of any data-heterogeneity measure** and **does not deteriorate with the participation ratio** $N/C$. |

ensure the bound is at most $\epsilon$, Corollary 4 chooses $T \geq 72\delta_0/(\zeta\epsilon)$ and requires $\gamma_t, \eta_t$ to be "small enough", which in view of equation (114) amounts to conditions of the form

$$K_t\left(\eta_t^2 a_t \Delta_\star + \gamma_t^2 N \tilde{a}_t \Delta^\star + \eta_t^2 R a_t \Delta_\star\right) \leq \epsilon.$$

Thus, when $\Delta_\star$ or $\Delta^\star$ are large, the admissible stepsizes must scale on the order of $\sqrt{\epsilon}/\sqrt{\Delta_\star}$ and $\sqrt{\epsilon}/\sqrt{\Delta^\star}$, respectively; otherwise the right-hand side of equation (114) cannot be made small and the theorem no longer guarantees convergence. In this sense, the "constants" involving $\Delta_\star, \Delta^\star$ are not harmless multiplicative factors in the usual big-$\tilde{\mathcal{O}}(\cdot)$ notation, but structural quantities that directly restrict the learning rate in highly heterogeneous regimes.

In contrast, FedCDR operates under the standard $L$-smooth nonconvex model and client reshuffling, and still attains an $\tilde{\mathcal{O}}(1/\epsilon)$ rate for the usual nonconvex criterion $\mathbb{E}\|\nabla f(x_{\text{out}})\|^2 \leq \epsilon$. Crucially, our leading constant is independent of any explicit data-heterogeneity measure and remains stable under partial participation, while additionally allowing inexact local proximal solves. This positions FedCDR as a practically relevant client-reshuffling method that combines a sharp nonconvex rate with robustness to data heterogeneity and system variability.

Table 15: Notation table

| Symbol | Description |
|---|---|
| $n$ | Total number of clients |
| $t$ | Meta-epoch index ($t = 0, 1, 2, \dots$) |
| $R$ | Number of communication rounds in each meta-epoch |
| $r$ | Communication round index within meta-epoch $t$, $r = 1, \dots, R$ |
| $S^t$ | Permutation of all clients in meta-epoch $t$ |
| $S_r^t$ | Batch of clients selected in communication round $r$ of meta-epoch $t$ |
| $x_r^t$ | Global model at communication round $r$ in meta-epoch $t$ |
| $x_{i,r}^t$ | Local model on client $i$ at communication round $r$ in meta-epoch $t$ |
| $\eta$ | Proximal coefficient in FedCDR |
| $f_i$ | Local objective function of client $i$ |
| $f$ | Global objective $f(x) = \frac{1}{n} \sum_{i=1}^n f_i(x)$ |

## K    NOTATION

To clarify the notation, we summarize the main symbols used in FedCDR in Table 15. The iterations are organized hierarchically. A meta-epoch $t$ corresponds to one full reshuffling of all $n$ clients and is composed of $R$ communication rounds indexed by $r = 1, \dots, R$. In communication round $r$ of meta-epoch $t$, only the clients in the batch $S_r^t$ participate: they receive the current global model $x_r^t$ and perform several local proximal-gradient iterations starting from $x_r^t$ to obtain their local models $x_{i,r}^t$. The server then aggregates $\{x_{i,r}^t : i \in S_r^t\}$ to form the next global model $x_{r+1}^t$. Thus, meta-epochs $t$ group multiple communication rounds $r$, and within each communication round the local iterations are the inner optimization steps carried out on each selected client.

## L    LIMITATIONS

Our experiments are constrained by limited computational resources, which prevent us from verifying the scalability of our algorithm on large datasets like ImageNet or testing its effectiveness in the federated training of large language models (LLMs).

## M    FUTURE WORKS

For future work, we are interested in studying the theoretical convergence of our method under generalized smoothness conditions. We are also interested in analyzing the last-iterate convergence of the proposed method.

## N    THE USE OF LLM

Our paper only uses LLMs to correct grammar.

## O    REPRODUCIBILITY STATEMENT

We have taken extensive steps to ensure the reproducibility of our results. All theoretical claims are stated with explicit assumptions, and complete proofs are provided in the appendix (see Appendix B and C). Our algorithm is described in detail in the main text (*Algorithm 1*), with additional derivation included in Appendix A. For empirical results, we report full experimental settings and complete experimental details in Section 6, Appendix F, and Appendix G. Together, these resources ensure that both our theoretical and empirical contributions can be independently reproduced.

