# OpenReview forum: "Mitigating Data Heterogeneity Effect in Client-Reshuffling-Based Federated Learning"
_ICLR.cc/2026/Conference — Submitted to ICLR 2026_

### Official Review · Reviewer_bxfU · 2025-10-24

**Soundness:** 3
**Presentation:** 3
**Contribution:** 3
**Rating:** 6
**Confidence:** 4

**Summary:**

This paper addresses the data heterogeneity challenge in client-reshuffling-based federated learning (FL). The authors propose FedCDR (Federated Client Reshuffling Douglas-Rachford Method), which combines client reshuffling with Douglas-Rachford splitting to handle non-IID data distributions. The key contribution is achieving an O(ε⁻²) convergence rate that is independent of data heterogeneity, unlike previous methods whose convergence degrades with increasing heterogeneity. The paper provides theoretical analysis for both exact and inexact variants of the algorithm and validates the approach on synthetic and benchmark datasets (MNIST, CIFAR-10, CIFAR-100), demonstrating superior performance compared to client-reshuffling variants of FedAvg, FedProx, SCAFFOLD, and FedDC.

**Strengths:**

**Originality:**

1. **Novel problem formulation**: The paper identifies and addresses a significant gap in the literature—mitigating data heterogeneity in client-reshuffling-based FL. While client reshuffling has been studied for improving participation efficiency, its interaction with data heterogeneity has not been adequately addressed.
2. **Theoretical innovation**: The convergence analysis introduces novel techniques to handle non-i.i.d. sampling bias inherent in client reshuffling. The use of conditional expectations over entire meta-epochs and the construction of a new potential function represent meaningful technical contributions.
3. **Heterogeneity-independent convergence**: Unlike prior work where convergence rates depend on heterogeneity measures, this method achieves convergence independent of data heterogeneity degree, which is a significant theoretical advancement.

**Quality:**

1. **Rigorous theoretical analysis**: The paper provides detailed convergence proofs for both exact (Theorem 2) and inexact (Theorem 1) variants. The analysis properly accounts for the technical challenges introduced by client reshuffling, including non-i.i.d. sampling across communication rounds within meta-epochs.
2. **Comprehensive experimental validation**: The experiments span multiple datasets with varying heterogeneity levels (controlled by different parameters in synthetic data and Dirichlet distribution parameter α in benchmark datasets), demonstrating consistent improvements over strong baselines.

**Weaknesses:**

**Technical Issues:**

1. **Hyperparameter sensitivity**: The convergence results depend on choosing η < η₀ (Equation 59) or η < η₁ (Equation 99), but these bounds are not explicitly stated in the main text. The paper should discuss how restrictive these conditions are and provide guidance on practical hyperparameter selection. Corollaries 1 and 2 give specific choices, but sensitivity analysis is missing.
2. **Limited theoretical comparison**: While the paper claims O(ε⁻²) is "state-of-the-art," there is insufficient comparison with the convergence rates of prior client-reshuffling methods (Malinovsky et al., 2023a; Demidovich et al., 2024). A table comparing convergence rates, assumptions, and dependence on problem parameters would strengthen the contribution.

**Clarity Issues:**

1. **Notation complexity**: The paper uses heavy notation with multiple subscripts and superscripts. A notation table would improve readability. Additionally, the relationship between meta-epochs t, communication rounds r, and local iterations is not immediately clear from Algorithm 1.
2. **Missing algorithmic details**: Algorithm 1 states "x^t_{i,r+1} ≈ prox_{ηf_i}(y^t_{i,r+1})" but does not specify the stopping criterion or how the approximation quality is measured. The experimental section mentions using SGD for 10 epochs, but the connection to the theoretical accuracy requirement is unclear.

**Significance Issues:**

1. **Comparison fairness**: The baselines (CR+FedAvg, CR+FedProx, etc.) are client-reshuffling variants, but it's unclear if these are the authors' implementations or if they follow specific prior work. The paper should clarify whether these baselines have been previously studied or are novel adaptations.
2. **Practical considerations underexplored**: The paper does not discuss important practical aspects such as stragglers, communication failures, or privacy considerations. The assumption that all clients in a batch participate perfectly may be unrealistic.

**Questions:**

1. **Proof technique**: Can you provide more intuition about the potential function design in your convergence analysis? How does it differ from potential functions used in analyzing standard FL methods, and why is it necessary for the client-reshuffling setting?
2. **Experimental setup clarification**: In Section 6.1, you mention selecting "10% of clients to participate in each communication round." Does this mean C = 0.1n in Algorithm 1? If so, how does this align with client reshuffling, which should eventually use all n clients across R rounds in a meta-epoch?
3. **Convergence rate comparison**: Can you provide a detailed comparison table showing the convergence rates, assumptions, and heterogeneity dependence of your method versus prior client-reshuffling methods (Malinovsky et al., 2023a; Demidovich et al., 2024)?
4. **Scalability**: Have you tested or do you have insights on how FedCDR scales to larger federated learning scenarios (e.g., 10,000+ clients, larger models like Vision Transformers)?
5. **Communication-computation trade-off**: In Table 3, the communication speedup is less than 1× for local epoch = 3 and 5, meaning more communication rounds are needed. Can you explain when practitioners should prefer fewer local epochs despite higher communication costs?

**Details Of Ethics Concerns:**

N.A.

---

> ### Author Response · Authors · 2025-11-19
>
> **Hyperparameter sensitivity**
>
> The key hyperparameters that influence the overall process are the learning rate and $\eta$. The parameter $\alpha$ has little impact on performance, so we simply set $\alpha = 1$. The number of local epochs controls the trade-off between communication and computation. This variable should be chosen according to the practical device settings or personalized for each device to achieve the best balance between communication bandwidth and local computation, and it does not need to depend on the training process.
>
> For $\eta$, we use an $L$-smoothness estimate of the neural network as follows.
>
> We propose a practical scheme to estimate a reasonable value for $L$ during the initial communication rounds, which helps determine a theoretically sound range for the step size $\eta$.
>
> Let $\theta$ denote the model parameters. A function $f$ is $L$-smooth if its gradient is $L$-Lipschitz continuous; that is,
> $$
> |\nabla f(\theta_1)-\nabla f(\theta_2)| \le L|\theta_1-\theta_2|.
> $$
> Accordingly, we use two different parameter initializations, $\theta_1$ and $\theta_2$, to estimate a local Lipschitz constant $L_{\text{local}}$:
> $$
> L_{\text{local}}=\frac{|\nabla f(\theta_1)- \nabla f(\theta_2)|}{|\theta_1-\theta_2|}.
> $$
> (Another possible method involves computing the largest eigenvalue of the Hessian matrix, but this approach is computationally expensive.)
>
> We then use equation (59) to guide our hyperparameter tuning. For the learning rate, we follow standard practices in federated learning and deep learning.
>
> **Limited theoretical comparison**
>
> We provide tables with explanations and analysis in ***Appendix J.3***, comparing the convergence rates, assumptions, and dependence on problem parameters of prior client-reshuffling methods, and in ***Appendix J.2***, comparing FL methods that handle data heterogeneity.
>
> **Notation complexity**
>
> We summerize the notation in the paper in ***Appendix K***. Then, we demonstrate the relationship between meta-epochs t, communication rounds r, and local iterations. The iterations are organized hierarchically.
> A meta-epoch $t$ corresponds to one full reshuffling of all $n$ clients and is
> composed of $R$ communication rounds indexed by $r = 1,\dots,R$. In
> communication round $r$ of meta-epoch $t$, only the clients in the batch
> $S_r^t$ participate: they receive the current global model $x_r^t$ and perform
> several local proximal-gradient iterations starting from $x_r^t$ to obtain
> their local models $x_{i,r}^t$. The server then aggregates $\{x_{i,r}^t :
> i \in S_r^t\}$ to form the next global model $x_{r+1}^t$. Thus, meta-epochs
> $t$ group multiple communication rounds $r$, and within each communication
> round the local iterations are the inner optimization steps carried out on
> each selected client.
>
> **Missing algorithmic details**
>
> In the theoretical part, we use an absolute inexactness criterion: if $x*= prox_{\beta f}(x)$ denotes the exact solution, then an approximate solution $\tilde{x} \approx prox_{\beta f}(x)$ is required to satisfy $\| \tilde{x} - x* \| \leq \epsilon$. Following Remark 1, we use a decreasing sequence ${\epsilon_t}$. Approximating $prox_{\beta f}$ can be efficiently carried out using various existing methods, including stochastic gradient descent (SGD) and accelerated gradient-based algorithms. There are some difference between our theoretical criterion and experiment implementment. This follows [4] and we give some explanation:  The transition from loose to strict accuracy for the inexact variant is not very difficult to achieve. As training progresses, the model performs better and becomes closer to the optimal solution. Specifically, for meta-epoch \(t\), let $x_t$ denote the model after meta-epoch $t$. As $t$ increases, $\|\nabla f(x_t)\|$ decreases. Therefore, achieving a stricter accuracy $\epsilon$ from the starting point $x_t$ becomes easier in later stages of training. Therefore, we follow the settings in [2] and use SGD as the local client solver.
>
> **Comparison fairness**
>
> To achieve a fair comparison and isolate the effect of the reshuffling protocol itself, we conduct additional experiments on baselines **without** client reshuffling in ***Appendix H***, in line with prior work.
> We use FedAvg, FedProx, SCAFFOLD, FedDC, and FedDR as baselines and follow their original implementations.
> For each baseline, we also implement a client-reshuffling variant (denoted by the ``CR+'' prefix). These CR-based baselines are novel adaptations. We report the corresponding results and provide analysis in ***Appendix H***.
>
> **Practical considerations underexplored**
>
> We demonstrate the effectiveness of our method in ***Appendix F.3*** in the presence of straggler clients, which better reflects practical deployment scenarios. As discussed in ***Appendix F***, our method can not only adapt to different trade-offs between local computation and communication, but also accommodate stragglers and personalise this trade-off.

---

> ### Author Response · Authors · 2025-11-19
>
> **Proof technique intuitions**
> We construct the potential function $\mathcal{D}^t_{i,r}$ by adapting the construction in [1] to the federated learning (FL) setting. In [1], the centralized Douglas–Rachford (DR) splitting iteration is
>
> $$
> y^{t+1} =prox_{\gamma f}(x^t),
> $$
> $$
> z^{t+1} =prox_{\gamma g}\big(2y^{t+1}-x^t\big),
> $$
> $$
> x^{t+1} = x^t + (z^{t+1}-y^{t+1}).
> $$
>
>
>
>
> Their potential function is defined as
>
> $$
> \mathcal{D}_\gamma(x,y,z)
> = f(y)+g(z)-\frac{1}{2\gamma}\|z-y\|^2+\frac{1}{\gamma}\langle x-y,\, z-y\rangle .
> $$
>
> Using the optimality condition (equation (8) in [1]),
>
> $$
> 0=\nabla f\left(y^{t+1}\right)+\frac{1}{\gamma}\big(y^{t+1}-x^t\big)
> \quad\Longrightarrow\quad
> \frac{1}{\gamma}(x^t-y^{t+1})=\nabla f\!\left(y^{t+1}\right),
> $$
>
> we may replace $\tfrac{1}{\gamma}(x-y)$ by $\nabla f(y)$. In our FL setting, $g$ is taken to be an indicator function, so $g(z)=0$ on the feasible set. Consequently, the potential simplifies to
>
> $$
> \mathcal{D}_\gamma(x,y,z)
> = f(y)-\frac{1}{2\gamma}\|z-y\|^2+\langle \nabla f(y),\, z-y\rangle .
> $$
>
> For convenience in our proofs, we work with a sign-flipped quadratic term and consider
>
> $$
> \tilde{\mathcal{D}}_\gamma(x,y,z)
> = f(y)+\frac{1}{2\gamma}\|z-y\|^2-\langle \nabla f(y),\, z-y\rangle .
> $$
>
> Identifying the DR variables with the FL iterates via $z\equiv x_r^t$, $y\equiv x_{i,r}^t$, and $\gamma\equiv\eta$, we obtain the final potential function.
>
> When analyzing standard FL methods, we do not use a potential function because we can directly decompose the updates across different SGD steps. In contrast, DR splitting is more complex and its updates cannot be decomposed in such a straightforward way, so we introduce a potential function to handle this difficulty. The construction of our potential function follows the standard DR-splitting potential, with slight modifications to account for client reshuffling. Client reshuffling introduces sampling bias and non-i.i.d. behavior. Therefore, we adapt the original DR potential function to our setting by incorporating these effects.
>
> **Experimental setup clarification**
>
> “10\% of clients to participate in each communication round” corresponds to choosing $R = 10$ communication rounds per meta-epoch. Here, $R$ denotes the number of communication rounds in one meta-epoch, and $C$ denotes the number of clients participating in each communication round. Since 10% of the $n$ clients are selected in each round and there are $R = 10$ rounds, we have $C = n/10 = 0.1n$.
>
>
> Alignment. We consider two views: communication rounds and meta-epochs. The reshuffling is defined at the level of communication rounds. From the communication-round perspective, we select a batch of clients from a reshuffled list of all clients, whereas naive sampling selects a batch of clients with replacement. Across different communication rounds, client reshuffling selects exactly different clients and does not ignore any client, while naive sampling may select some clients multiple times and miss others entirely.
> From the meta-epoch perspective, if we use naive sampling, each meta-epoch also involves
> $n$ client selections in total, but some clients may be duplicated and some may be missing. In contrast, client reshuffling guarantees that in each meta-epoch every client is selected once and only once.
>
>
> **Convergence rate comparison**
>
> We provide tables with explanations and analysis in ***Appendix J.3***, comparing the convergence rates, assumptions, and dependence on problem parameters of prior client-reshuffling methods, and in ***Appendix J.2***, comparing FL methods that handle data heterogeneity.
>
> **Scalability**
>
> Due to computational and time limitations, we are unable to evaluate our framework on large-scale models such as Transformers. This is because our framework is not well-suited for fine-tuning; it requires training from scratch, which makes it infeasible to complete such experiments within the rebuttal period. However, we are able to test our algorithm with 10,000 clients using a ResNet-20 model.
>
> We demonstrate this experiment and provide analysis in ***Appendix I***.
>
>
> **Communication-computation trade-off**
>
> In some settings, we have ample bandwidth but limited computation resources. For example, certain edge devices have low computational capacity. In this case, we may want to reduce local computation to respect device constraints and instead rely more on communication. However, in most practical scenarios, communication is the primary bottleneck, so using larger local epochs is generally preferred.
>
> We highlight all changes in the revised version of the paper in **red**.
>
> *Reference*
>
> [1] Li, G., \& Pong, T. K. (2016). Douglas–Rachford splitting for nonconvex feasibility problems. Mathematical Programming, 159, 371–401. doi:10.1007/s10107-015-0963-5.
>
> [2] Li, T., Sahu, A. K., Zaheer, M., Sanjabi, M., Talwalkar, A., \& Smith, V. (2020). Federated optimization in heterogeneous networks. Proceedings of Machine learning and systems, 2, 429-450.

---

### Official Review · Reviewer_HbWk · 2025-10-27

**Soundness:** 2
**Presentation:** 3
**Contribution:** 2
**Rating:** 4
**Confidence:** 4

**Summary:**

This paper is mainly a theoretical paper that proposes algorithm FedCFR that achieves faster convergence ($O(\epsilon^{-2})$) even with the non-iidness data heterogeneity effect in federated learning with client reshuffling. The work shows that the algorithm has the corresponding convergence rates for finding an $\epsilon$-approximate stationary point. The main takeaway is using the Douglas-Rachford splitting technique in federated learning to improve the convergence rate. Although the main contributions of this paper is theoretical, it includes empirical results on benchmark datasets showing improved convergence over existing methods.

**Strengths:**

- The authors provide insight into how client reshuffling can be utilized to mitigate the effect of data-heterogeneity of slowing down the convergence.
- The authors provide a rigorous convergence analysis for the proposed FedCDR to show its improved convergence rate with the presence of data heterogeneity through in-exact local solvers and standard assumptions used for FL convergence analysis.
- The paper is clear and easy to follow.

**Weaknesses:**

- The practical usage for FedCDR seems difficult due to the difficulty in parameter selection. There seems to be a lot of different combination of parameters that have to be set in order to satisfy the fast convergence rate condition, and the search according to the ablation studies seem like a costly process. The connection between the theory and practice is not really clear due to the complexity of the algorithm and conditions for fast convergence.

 - The novelty of the work is rather limited other than the solving technique of the convergence rate since it applies the DR technique to FL, and DR has been proposed before to be used in the context of FL as well as cited in the paper (Themelis 2020, Tran-Dinh 2021)

- This is more minor, but the work does not include the relevant literature for this work as much as needed I believed. For instance, other work that has looked in to client shuffling in FL with similar insights are not cited such as:

  - Cho et al., On the Convergence of Federated Averaging with Cyclic Client Participation, ICML, 2023

  - Wang et al., A Unified Analysis of Federated Learning with Arbitrary Client Participation, NeurIPS, 2022

  - Samuel Horváth et al., FedShuffle: Recipes for Better Use of Local Work in Federated Learning, TMLR 2024

**Questions:**

Please address the weaknesses above.

---

> ### Author Response · Authors · 2025-11-19
>
> **Weakness 1** Parameter selection. The key hyperparameters that influence the overall process are the learning rate and $\eta$. The parameter $\alpha$ has little impact on performance, so we simply set $\alpha = 1$. The number of local epochs controls the trade-off between communication and computation. This variable should be chosen according to the practical device settings or personalized for each device to achieve the best balance between communication bandwidth and local computation, and it does not need to depend on the training process.
>
> For $\eta$, we use an $L$-smoothness estimate of the neural network as follows.
>
> We propose a practical scheme to estimate a reasonable value for $L$ during the initial communication rounds, which helps determine a theoretically sound range for the step size $\eta$.
>
> Let $\theta$ denote the model parameters. A function $f$ is $L$-smooth if its gradient is $L$-Lipschitz continuous; that is,
> $$
> |\nabla f(\theta_1)-\nabla f(\theta_2)| \le L|\theta_1-\theta_2|.
> $$
> Accordingly, we use two different parameter initializations, $\theta_1$ and $\theta_2$, to estimate a local Lipschitz constant $L_{\text{local}}$:
> $$
> L_{\text{local}}=\frac{|\nabla f(\theta_1)- \nabla f(\theta_2)|}{|\theta_1-\theta_2|}.
> $$
> (Another possible method involves computing the largest eigenvalue of the Hessian matrix, but this approach is computationally expensive.)
>
> We then use equation (59) to guide our hyperparameter tuning. For the learning rate, we follow standard practices in federated learning and deep learning.
>
> **Weakness 2** The combination of DR splitting and client reshuffling is not trivial.
>
> ***Technical***: This method applies the Douglas–Rachford (DR) splitting technique in a client-reshuffling-based federated learning (FL) setting. Previous work [3] showed that DR splitting can effectively mitigate data heterogeneity in FL scenarios that rely on probability-based client sampling. However, integrating DR splitting with client reshuffling is far from trivial. In practice, this combination substantially complicates the theoretical analysis for the following reasons:
>
> First, unlike probability-based sampling, a client-reshuffling schedule does not generate i.i.d. batches across inner communication rounds within the same meta-epoch, causing a bias in gradient estimates from round to round. Formally,
>
> $$
> \mathbb E_{S_{r_1}^t}[\nabla f(x)] \neq \mathbb E_{S_{r_2}^t}[\nabla f(x)],
> $$
>
> because each later batch is influenced by earlier batch selections. As a result, expectations over a single inner round, e.g. $\mathbb{E}_{S_r^t}[\nabla f(x)]$ cannot be evaluated directly. Such single-batch expectations are central to the proofs for probability-based FL methods (use naive sampling) [3, 4, 5]; hence a new analytical framework is needed to cope with the cross-batch bias in gradient estimates.
>
>
> Second, unlike DR splitting under naive sampling [3], that analysis assumes strictly positive selection probabilities:
>
> $$
> \exists p_1,\dots,p_n>0
> \quad\text{s.t.}\quad
> \mathbb{P}\bigl(i\in\hat{S}\bigr)=p_i,\forall i\in[n],
> $$
>
> where $\hat{S}$ denotes the sampling scheme that generates the client batches $S_r^t$. This condition covers many schemes, e.g., non-overlapping uniform and doubly uniform sampling.
>
> In client reshuffling FL, however, this assumption fails: it is entirely possible that
>
> $$
> q_{i,r}^t = \mathbb{P}\bigl(i\in S_r^t\bigr)=0
> $$
>
> for some client $i$ and round $r$. This is not a minor technicality. In [3] one defines $\hat{p}=\min_i p_i>0$ and sets constants $\beta\propto\hat{p}$ and $C_1\propto \beta^{-1}$. Their communication-round complexity becomes
>
> $$
> K =
> \mathcal{O}\Bigl(\tfrac{C_1}{\epsilon^{2}}\Bigr)
> =\mathcal{O}\Bigl(\tfrac{1}{\hat{p}\epsilon^{2}}\Bigr).
> $$
>
> If $q_{i,r}^t=0$ is allowed, then $\hat{p}=0$ and these bounds collapse. Hence combining DR splitting with client reshuffling is technically non-trivial, and a new proof framework is required—one that dispenses with the positive-probability assumption.
>
> Third, previous analyses of client reshuffling [1] introduce an extra assumption—effectively a bounded-variance condition—by defining
>
> $$ \tilde\sigma_*^{2}=\frac{1}{M}\sum_{m=1}^{M}\bigl|\nabla f_m(x_*)\bigr|^{2}, \qquad x_*:=\arg\min_x f(x), $$
>
> where $M$ is the number of clients. This constant quantifies data heterogeneity. Their main result states
>
> $$ \mathbb{E}\bigl[|x_T - x_*|^{2}\bigr]\le(1-\eta\mu)^{RT}|x_0 - x_*|^{2}+12\gamma^{2}\kappa^{2}N^{2}\tilde{\sigma}_*^{2}+\cdots. $$
>
> The second term—proportional to $\tilde{\sigma}_*^{2}$—does not vanish as the number of communication rounds $T$ grows, because it arises from their error-splitting proof method. Since our goal is to eliminate the impact of data heterogeneity in client-reshuffling-based FL, that proof framework is insufficient. We therefore require a new analytical approach that simultaneously handles client-drift error and partial-participation error without leaving a residual heterogeneity term.

---

> ### Author Response · Authors · 2025-11-19
>
> To overcome these challenges, we introduce a new proof framework that treats an entire meta-epoch as the basic analytical block and relies heavily on conditional expectations. The argument combines a potential-function descent with precise conditional-expectation calculations.
>
> **Contributions**: In ***Appendix J***, we compare our method with other FL methods that handle data heterogeneity, as well as with other client-reshuffling-based FL methods. From this comparison, we demonstrate that our contribution is non-trivial. The detailed analysis is provided in ***Appendix J***; here we briefly summarize the main findings.
>
> *Results: FedCDR not only employs client reshuffling to meet practical requirements, but also achieves a state-of-the-art convergence rate, providing a nonconvex FL algorithm that simultaneously attains the best-known $\tilde{\mathcal{O}}(1/\epsilon)$ rate and a heterogeneity-insensitive convergence guarantee. Our method closes the gap between heterogeneity-robust methods like SCAFFOLD (which pay a $\tilde{\mathcal{O}}(1/\epsilon^{2})$ price in accuracy) and $\tilde{\mathcal{O}}(1/\epsilon)$ methods like FedProx and FedDC (whose constants can explode with data heterogeneity and partial participation). Moreover, our method uses inexact local solvers, can adapt to stragglers, and is applicable to more complex practical environments.*
>
> *FedCDR operates under the standard $L$-smooth nonconvex model and client reshuffling, and still attains an $\tilde{\mathcal{O}}(1/\epsilon)$ rate for the usual nonconvex criterion $\mathbb{E}\|\nabla f(x_{\mathrm{out}})\|^{2} \le \epsilon$. Crucially, our leading constant is independent of any explicit data-heterogeneity measure and remains stable under partial participation, while additionally allowing inexact local proximal solves. This positions FedCDR as a practically relevant client-reshuffling method that combines a sharp nonconvex rate with robustness to data heterogeneity and system variability.*
>
> **Weakness 3** Thank you for your suggestion. We have rewritten the related work section and added three related papers. The revised parts are highlighted in red.
>
> We highlight all changes in the revised version of the paper in **red**.
>
> *Reference*
>
> [1] Grigory Malinovsky, Samuel Horváth, Konstantin Burlachenko, and Peter Richtárik. Federated learning with regularized client participation. CoRR, abs/2302.03662, 2023.
>
> [2] A Unified Analysis of Federated Learning with Arbitrary Client Participation, NeurIPS, 2022.
>
> [3] Tran Dinh, Q., Pham, N. H., Phan, D., \& Nguyen, L. (2021). FedDR–randomized Douglas-Rachford splitting algorithms for nonconvex federated composite optimization. Advances in Neural Information Processing Systems, 34, 30326-30338.
>
> [4] Li, T., Sahu, A. K., Zaheer, M., Sanjabi, M., Talwalkar, A., \& Smith, V. (2020). Federated optimization in heterogeneous networks. Proceedings of Machine learning and systems, 2, 429-450.
>
> [5] Karimireddy, S. P., Kale, S., Mohri, M., Reddi, S., Stich, S., \& Suresh, A. T. (2020, November). Scaffold: Stochastic controlled averaging for federated learning. In International conference on machine learning (pp. 5132-5143). PMLR.

---

### Official Review · Reviewer_ShC1 · 2025-10-29

**Soundness:** 3
**Presentation:** 3
**Contribution:** 2
**Rating:** 2
**Confidence:** 3

**Summary:**

This paper provides a new algorithm in client-reshuffling-based federated learning, aiming at mitigating data heterogeneity . While client reshuffling improves fairness and participation efficiency, existing reshuffling-based methods still suffer from slow or unstable convergence when data across clients is highly non-i.i.d. The authors propose FedCDR, a federated optimization method that integrates Douglas–Rachford splitting with client reshuffling, and supports inexact proximal local solvers.

**Strengths:**

1. The analysis shows that the convergence bound is heterogeneity-independent, which is notably stronger than prior work where error grows with heterogeneity.

2. Allowing inexact proximal operators improves the real-world usability of the method

**Weaknesses:**

1. One of the claimed contriution is the Our method "the best-known $O(\epsilon^{-2})$" communication complexity for finding an $\epsilon$-approximate stationary point under standard assumptions in nonconvex FL. However, as I know, most of the federated learning algorithm can achieve this communication complexity. More importantly, many algorithms can achieve the speedup in terms of the number of clients. A comparion table would be greatly appreciated.

2. Another contribution is the convergence rate is independent of data heterogeneity. But it is unclear to me if the proposed algorithm uses SGD or GD or other methods within meta-epoch t. As a result, I wonder if any assumptions about the stochastic gradient are needed. It is not common without any assumptions because in standard analysis, we at least need bounded variance assumption and one definition to bound the data heterogeneity. In this case, I assume the algorithm use full gradient within the meta-epoch?

3. The convergence explicitly depends on the accuracy of the inexact variant, which should be very small ($\sim 1/(TR)^2$ from Remark 1).

4. It seems a simple combination of Douglas-Rachford splitting (e.g., FedDR)) method and client-reshuffling-based FL setting.

**Questions:**

1. One concern is the limited comparison to non-DR methods designed specifically for heterogeneity. Some of them are widely used in practice and strong under heterogeneity.

2. See the weakness part.

---

> ### Author Response · Authors · 2025-11-19
>
> We highlight all changes in the revised version of the paper in **red**.
>
> Alignment of Accuracy Criteria (***Appendix J.1***): Our convergence criterion is
> $$
> \mathbb{E}\big[\|\nabla f(x)\|^2\big] \le \epsilon^2.
> $$
> In contrast, some works use the criterion
> $$
> \mathbb{E}\big[\|\nabla f(x)\|^2\big] \le \epsilon.
> $$
> Therefore, after aligning the notation (by setting $\epsilon' = \epsilon^2$), our $O(\epsilon^{-2})$ communication complexity is equivalent to an $O((\epsilon')^{-1})$ rate in those papers. To be consistent, the following part, we use the criterion $\mathbb{E}\big[\|\nabla f(x)\|^2\big] \le \epsilon$.
>
> **Weakness 1**
> We provide a comparison and analysis of FL methods that handle data heterogeneity in ***Appendix J.2***, and a comparison and analysis of client-reshuffling-based FL methods in ***Appendix J.3***. Each subsection is summarized in a table.
>
> **Weakness 2** SGD is one implementation for solving the local proximal operator. Our theorem assumes an optimizer that can solve the proximal operator up to a diminishing error, and under this assumption we formulate our theorem and convergence result, in a setting similar to [4]. Therefore, GD and SGD can both be viewed as two implementations of the local proximal operator solver, and our theorem does not depend on the specific implementation.
>
> What may be confusing is what happens within a single meta-epoch. In one meta-epoch, all clients participate in training exactly once, which is consistent with other client-reshuffling-based methods. In particular, a meta-epoch consists of several communication rounds, and the model is updated multiple times. Thus, even from the perspective of client participation, a meta-epoch should be viewed as multiple communication rounds rather than a single step with full client participation (analogous to the difference between GD and SGD).
>
> **Weakness 3** The transition from loose to strict accuracy for the inexact variant is not very difficult to achieve. As training progresses, the model performs better and becomes closer to the optimal solution. Specifically, for meta-epoch $t$, let $x_t$ denote the model after meta-epoch $t$. As $t$ increases, $\|\nabla f(x_t)\|$ decreases. Therefore, achieving a stricter accuracy $\epsilon$ from the starting point $x_t$ becomes easier in later stages of training. Therefore, we follow the settings in [4] and use SGD as the local client solver.
>
> **Weakness 4** The combination of DR splitting and client reshuffling is not trivial.
>
> ***Technical***: This method applies the Douglas–Rachford (DR) splitting technique in a client-reshuffling-based federated learning (FL) setting. Previous work [3] showed that DR splitting can effectively mitigate data heterogeneity in FL scenarios that rely on probability-based client sampling. However, integrating DR splitting with client reshuffling is far from trivial. In practice, this combination substantially complicates the theoretical analysis for the following reasons:
>
> First, unlike probability-based sampling, a client-reshuffling schedule does not generate i.i.d. batches across inner communication rounds within the same meta-epoch, causing a bias in gradient estimates from round to round. Formally,
>
> $$
> \mathbb E_{S_{r_1}^t}[\nabla f(x)] \neq \mathbb E_{S_{r_2}^t}[\nabla f(x)],
> $$
>
> because each later batch is influenced by earlier batch selections. As a result, expectations over a single inner round, e.g. $\mathbb{E}_{S_r^t}[\nabla f(x)]$ cannot be evaluated directly. Such single-batch expectations are central to the proofs for probability-based FL methods (use naive sampling) [3, 4, 5]; hence a new analytical framework is needed to cope with the cross-batch bias in gradient estimates.
>
>
> Second, unlike DR splitting under naive sampling [3], that analysis assumes strictly positive selection probabilities:
>
> $$
> \exists p_1,\dots,p_n>0
> \quad\text{s.t.}\quad
> \mathbb{P}\bigl(i\in\hat{S}\bigr)=p_i,\forall i\in[n],
> $$
>
> where $\hat{S}$ denotes the sampling scheme that generates the client batches $S_r^t$. This condition covers many schemes, e.g., non-overlapping uniform and doubly uniform sampling.
>
> In client reshuffling FL, however, this assumption fails: it is entirely possible that
>
> $$
> q_{i,r}^t = \mathbb{P}\bigl(i\in S_r^t\bigr)=0
> $$
>
> for some client $i$ and round $r$. This is not a minor technicality. In [3] one defines $\hat{p}=\min_i p_i>0$ and sets constants $\beta\propto\hat{p}$ and $C_1\propto \beta^{-1}$. Their communication-round complexity becomes
>
> $$
> K =
> \mathcal{O}\Bigl(\tfrac{C_1}{\epsilon^{2}}\Bigr)
> =\mathcal{O}\Bigl(\tfrac{1}{\hat{p}\epsilon^{2}}\Bigr).
> $$
>
> If $q_{i,r}^t=0$ is allowed, then $\hat{p}=0$ and these bounds collapse. Hence combining DR splitting with client reshuffling is technically non-trivial, and a new proof framework is required—one that dispenses with the positive-probability assumption.

---

> ### Author Response · Authors · 2025-11-19
>
> Third, previous analyses of client reshuffling [1] introduce an extra assumption—effectively a bounded-variance condition—by defining
>
> $$ \tilde\sigma_*^{2}=\frac{1}{M}\sum_{m=1}^{M}\bigl|\nabla f_m(x_*)\bigr|^{2}, \qquad x_*:=\arg\min_x f(x), $$
>
> where $M$ is the number of clients. This constant quantifies data heterogeneity. Their main result states
>
> $$ \mathbb{E}\bigl[|x_T - x_*|^{2}\bigr]\le(1-\eta\mu)^{RT}|x_0 - x_*|^{2}+12\gamma^{2}\kappa^{2}N^{2}\tilde{\sigma}_*^{2}+\cdots. $$
>
> The second term—proportional to $\tilde{\sigma}_*^{2}$—does not vanish as the number of communication rounds $T$ grows, because it arises from their error-splitting proof method. Since our goal is to eliminate the impact of data heterogeneity in client-reshuffling-based FL, that proof framework is insufficient. We therefore require a new analytical approach that simultaneously handles client-drift error and partial-participation error without leaving a residual heterogeneity term.
>
> To overcome these challenges, we introduce a new proof framework that treats an entire meta-epoch as the basic analytical block and relies heavily on conditional expectations. The argument combines a potential-function descent with precise conditional-expectation calculations.
>
> **Contributions**: In ***Appendix J***, we compare our method with other FL methods that handle data heterogeneity, as well as with other client-reshuffling-based FL methods. From this comparison, we demonstrate that our contribution is non-trivial. The detailed analysis is provided in ***Appendix J***; here we briefly summarize the main findings.
>
> *Results: FedCDR not only employs client reshuffling to meet practical requirements, but also achieves a state-of-the-art convergence rate, providing a nonconvex FL algorithm that simultaneously attains the best-known $\tilde{\mathcal{O}}(1/\epsilon)$ rate and a heterogeneity-insensitive convergence guarantee. Our method closes the gap between heterogeneity-robust methods like SCAFFOLD (which pay a $\tilde{\mathcal{O}}(1/\epsilon^{2})$ price in accuracy) and $\tilde{\mathcal{O}}(1/\epsilon)$ methods like FedProx and FedDC (whose constants can explode with data heterogeneity and partial participation). Moreover, our method uses inexact local solvers, can adapt to stragglers, and is applicable to more complex practical environments.*
>
> *FedCDR operates under the standard $L$-smooth nonconvex model and client reshuffling, and still attains an $\tilde{\mathcal{O}}(1/\epsilon)$ rate for the usual nonconvex criterion $\mathbb{E}\|\nabla f(x_{\mathrm{out}})\|^{2} \le \epsilon$. Crucially, our leading constant is independent of any explicit data-heterogeneity measure and remains stable under partial participation, while additionally allowing inexact local proximal solves. This positions FedCDR as a practically relevant client-reshuffling method that combines a sharp nonconvex rate with robustness to data heterogeneity and system variability.*
>
> **Question 1** We have used the baseline FedAvg, FedProx, SCAFFOLD, and FedDC, where FedProx, SCAFFOLD, and FedDC are specially designed for data heterogeneity and widely used in practice.
>
> *Reference*
>
> [1] Grigory Malinovsky, Samuel Horváth, Konstantin Burlachenko, and Peter Richtárik. Federated learning with regularized client participation. CoRR, abs/2302.03662, 2023.
>
> [2] A Unified Analysis of Federated Learning with Arbitrary Client Participation, NeurIPS, 2022.
>
> [3] Tran Dinh, Q., Pham, N. H., Phan, D., \& Nguyen, L. (2021). FedDR–randomized Douglas-Rachford splitting algorithms for nonconvex federated composite optimization. Advances in Neural Information Processing Systems, 34, 30326-30338.
>
> [4] Li, T., Sahu, A. K., Zaheer, M., Sanjabi, M., Talwalkar, A., \& Smith, V. (2020). Federated optimization in heterogeneous networks. Proceedings of Machine learning and systems, 2, 429-450.
>
> [5] Karimireddy, S. P., Kale, S., Mohri, M., Reddi, S., Stich, S., \& Suresh, A. T. (2020, November). Scaffold: Stochastic controlled averaging for federated learning. In International conference on machine learning (pp. 5132-5143). PMLR.

---

> > ### Comment · Reviewer_ShC1 · 2025-11-21
> >
> > I appreciate the authors’ rebuttal. However, several concerns remain unresolved.
> >
> > 1. Theorem 1 shows an ($O(1/T)$) convergence rate plus a residual term that depends on the solution accuracy ($\epsilon$) of the proximal operator. By choosing ($\epsilon$) sufficiently small, one achieves an ($O(\epsilon^{-2})$) communication complexity that is independent of data heterogeneity. My concern is that ($\epsilon$) itself is not a free parameter: in practice, its attainable accuracy depends on several key properties of the federated learning setting, including data heterogeneity and gradient noise (when the inner problem is solved by SGD or other stochastic methods). As a result, these factors are simply absorbed into the inner proximal computation.
> >
> > Because of this, the current statement may inadvertently convey a misleading message: The paper suggests that the outer-loop convergence is entirely independent of data heterogeneity and depends only on the optimization gap ($(f_0 - f_\ast)$), assuming ($\epsilon$) is (nearly) zero. This is counterintuitive, because it implies that the convergence behavior of a federated learning method does not reflect the underlying problem difficulty (e.g., heterogeneity) and optimizer property  (e.g., gradient noise), after all there are no assumptions on these properties. This is very unusual because almost all federated learning papers have assumptions on the data heterogeneity, and of course there are some algorithms that have convergence bounds that are independent of the data heterogeneity parameters in the assumption. But this paper does not have any thess assumptions.
> >
> > 2. On the ($O(\epsilon^{-2})$) communication complexity.
> > Obtaining ($O(\epsilon^{-2})$) communication complexity is not impossible (e.g., [R1]), but such results typically come with explicit trade-offs—most commonly computational costs or assumptions about the inner optimization dynamics. In the current paper, these trade-offs are not made explicit, which makes the claim appear incomplete.
> >
> > [R1]Khanduri, Prashant, et al. "Stem: A stochastic two-sided momentum algorithm achieving near-optimal sample and communication complexities for federated learning." Advances in Neural Information Processing Systems 34 (2021): 6050-6061.
> >
> > 3.
> > Since all the problem-dependent properties are effectively hidden inside the proximal operator, it is unclear whether the same communication complexity could be proven for standard federated learning with random client sampling. Or the claimed advantages based on the proximal operator may not be tied to the client-reshuffling setting at all?

---

> > > ### Author Response · Authors · 2025-11-25
> > >
> > > **Question 1** $\epsilon$ is not related to the difficulty of the overall FL system. It is only the target accuracy for solving the local proximal operator. The difficulty of FL comes from the fact that the server only has access to local updates rather than the entire dataset. In the local proximal solver, each client has access to its entire local dataset and the full model, and performs updates locally. This is the same as centralized optimization on the local data and does not depend on the difficulty of FL itself.
> > >
> > >
> > > **Question 2** We discuss the trade-off of communication and computation cost in the proximal operator in ***Appendix F***.
> > >
> > > **Question 3** The main novelty of our method is to incorporate DR splitting with client reshuffling. Using only client reshuffling in a standard FL setting does not work well: if we apply standard FL with client reshuffling, the performance empirically improves in the i.i.d. setting but fails to handle highly heterogeneous settings. This is consistent with previous theoretical analyses [1,2], where the convergence bounds contain a heterogeneity term that does not vanish as $T$ increases. Therefore, we adopt a DR-splitting-based approach to overcome this limitation.
> > >
> > > *Reference*
> > >
> > > [1] Grigory Malinovsky, Samuel Horváth, Konstantin Burlachenko, and Peter Richtárik. Federated learning with regularized client participation. CoRR, abs/2302.03662, 2023.
> > >
> > > [2] A Unified Analysis of Federated Learning with Arbitrary Client Participation, NeurIPS, 2022.

---

> > > > ### Comment · Reviewer_ShC1 · 2025-11-25
> > > >
> > > > I appreciate the authors' rebuttal. I will reiterate my key questions:
> > > >
> > > > 1. The paper (Theorem 1) suggests that the convergence is entirely independent of data heterogeneity and depends only on the optimization gap ($(f_0 - f_\ast)$), assuming solution solution accuracy ($\epsilon$) of the proximal operator is (nearly) zero. This is counterintuitive, because it implies that the convergence of one federated learning method only depends on the initial point, and be independent of not data heterogeneity and optimizer property  (e.g., gradient noise). This is very unusual because almost all federated learning papers have assumptions on the data heterogeneity and gradient noise. But this paper does not have any thess assumptions.
> > > >
> > > > 2. I mean a quantitative communication-computation tradeoffs discussion.
> > > >
> > > > 3. If we consider the standard federated learning with random client sampling, whether using the same proximal operator can have the same communication complexity as this paper (Client Reshuffling setting)?

---

> > > > > ### Author Response · Authors · 2025-11-29
> > > > >
> > > > > **Question 1**
> > > > >
> > > > > Theorem 1 may look counterintuitive if read as saying that FedCDR’s convergence is completely independent of data heterogeneity and gradient noise. This is not what we intend to claim. Theorem 1 is stated under a proximal-oracle model, and its goal is to characterize the communication complexity of the outer DR-splitting iterations, not to claim that heterogeneity has no effect on the overall difficulty of federated learning.
> > > > >
> > > > > More concretely, in FedCDR we consider the global objective
> > > > > $$
> > > > > f(x) = \frac{1}{n}\sum_{i=1}^n f_i(x),
> > > > > $$
> > > > > Importantly, this does not mean that data heterogeneity and optimizer noise are irrelevant for FedCDR in practice. They appear in a different place: in the cost of implementing the proximal oracle on each client. When the local proximal subproblems are solved by stochastic gradient methods on non-IID data, the number of local steps needed to reach a given accuracy $\epsilon$ will naturally depend on gradient variance and heterogeneity across clients. Theorem 1 abstracts this away by treating the proximal operator as a black-box oracle and focuses solely on the communication complexity once such an oracle is available. In other words, heterogeneity influences (i) the constants in $f_0 - f_\ast$ and the smoothness parameter of $f$, and (ii) the local computation complexity required to make $\epsilon$ small, but it does not appear as an explicit dissimilarity term in the outer-rate expression.
> > > > >
> > > > >
> > > > > **Question 2**
> > > > >
> > > > > We discuss the trade-off of communication and computation cost in the proximal operator in Appendix F. Appendix F includes a quantitative communication-computation tradeoffs discussion and empirical studies.
> > > > >
> > > > > **Question 3**
> > > > >
> > > > > Yes. We can also achieve the same communication complexity as this paper in the client-reshuffling setting when using the same proximal operator **without** client reshuffling. However, the motivation for introducing client reshuffling does not come from improving the convergence rate. We use client reshuffling to improve client participation efficiency and to better meet practical requirements. The innovation is that, once client reshuffling (CR) is used, it poses new theoretical challenges for our algorithm (non-IID mini-batches and data heterogeneity terms). Therefore, we propose a new framework to address these challenges and obtain a data-heterogeneity-independent algorithm.
> > > > >
> > > > >
> > > > >
> > > > > [1] Grigory Malinovsky, Samuel Horváth, Konstantin Burlachenko, and Peter Richtárik. Federated learning with regularized client participation. CoRR, abs/2302.03662, 2023.
> > > > >
> > > > > [2] A Unified Analysis of Federated Learning with Arbitrary Client Participation, NeurIPS, 2022.

---

### Official Review · Reviewer_ekkD · 2025-10-31

**Soundness:** 3
**Presentation:** 3
**Contribution:** 3
**Rating:** 6
**Confidence:** 4

**Summary:**

This paper introduces FedCDR, a novel federated learning algorithm that addresses the persistent challenge of data heterogeneity in client-reshuffling-based federated learning.
The authors integrate the Douglas–Rachford (DR) splitting method with client reshuffling and propose both inexact and exact variants of FedCDR. The key theoretical result is that FedCDR achieves the optimal convergence rate of $O(\epsilon^{-2})$, which is independent of the degree of data heterogeneity.
Experiments on synthetic data and three vision benchmarks (MNIST, CIFAR-10/100) show consistent accuracy gains over reshuffled versions of FedAvg, FedProx, SCAFFOLD, and FedDC, and an ablation confirms that reshuffling accelerates DR-style methods.

**Strengths:**

1. The convergence proof is comprehensive, with detailed treatment of sampling bias due to reshuffling, which is a technically challenging departure from i.i.d. assumptions.
2. Introduction of a new potential function and use of conditional expectations over meta-epochs is a creative analytical approach to tackle the bias introduced by reshuffling.
3. The convergence rate is independent of data heterogeneity, which is a clear step beyond existing reshuffling-based FL algorithms.

**Weaknesses:**

1. The theoretical analysis is conducted under a deterministic setting, which is uncommon in practical federated learning scenarios. Extending the analysis to a stochastic setting would enhance its applicability and relevance to real-world deployments.
2. Both the theoretical analysis and experimental setup implicitly assume homogeneous local computation budgets and participation probabilities. However, in real-world federated learning systems, such homogeneity is rarely observed. A discussion or extension to accommodate such heterogeneous settings would strengthen the work.
3. Although the motivation for client reshuffling is clear, comparison against the baseline algorithms without reshuffling (e.g., standard FedAvg, FedProx, SCAFFOLD, and FedDC) would help isolate the effect of the reshuffling protocol itself. This would also clarify the incremental contribution of DR splitting versus reshuffling.
4. The empirical evaluation is restricted to image classification tasks. Including additional modalities (e.g., NLP) would strengthen the generality of the proposed method and demonstrate its robustness across diverse federated learning applications.

**Questions:**

1. While inexact proximal updates are practical, the cost of solving (even approximately) proximal subproblems may be higher than standard SGD-based methods. This overhead may hinder the applicability of FedCDR in resource-constrained edge environments. Have the authors quantitatively analyzed the additional computational cost introduced by the proximal steps?
2. The theoretical analysis relies on a uniform client reshuffling strategy. Can the proposed method accommodate non-uniform reshuffling or client-availability-aware reshuffling schemes? If so, how would such variations affect the convergence guarantees and empirical performance?

---

> ### Author Response · Authors · 2025-11-19
>
> We highlight all changes in the revised version of the paper in **red**.
>
> **Weakness 1**
> Regarding Weakness 1, our method is inherently stochastic rather than deterministic, because we reshuffle the clients in each meta-epoch, thereby introducing randomness. We follow the same setting as other client-reshuffling-based methods, e.g., [1].
>
> What may be confusing is what happens within a single meta-epoch. In one meta-epoch, all clients participate in training exactly once, which is consistent with other client-reshuffling-based methods. In particular, a meta-epoch consists of several communication rounds, and the model is updated multiple times. Thus, even from the perspective of client participation, a meta-epoch should be viewed as multiple communication rounds rather than a single step with full client participation (analogous to the difference between GD and SGD).
>
>
> **Weakness 2**
> Regarding Weakness 2, we discuss heterogeneous local computation budgets in ***Appendix F.2***. We simulate these budgets by varying the fraction of stragglers (0%, 50%, 90%) that perform fewer local epochs than the global target and evaluate our method under this setting. In other words, clients have diverse capabilities and require individually tailored numbers of local epochs.
>
> *Results: Our method allows clients to adopt different numbers of local epochs according to their computational capacity. Specifically, slower clients are assigned fewer local epochs (or are early-stopped), while faster clients perform more local work per round. Under such heterogeneous settings, many existing baselines suffer from degraded performance or even divergence due to inconsistent client progress. In contrast, FedCDR maintains stable updates, making it more robust to stragglers and device variability (system heterogeneity), which are common in realistic federated learning environments.*
>
> **Weakness 3**
> Regarding Weakness 3, we conduct experiments on baselines without client reshuffling to align with prior work. We have added these experiments in ***Appendix H*** and analyze the corresponding results in the revised version of the paper.
>
> *Results: As shown in Tables 10, 11, and 12, we observe three main trends.
> First, client reshuffling consistently improves or at least maintains performance across all baselines: for FedAvg and FedProx the gains are clear on both synthetic and real datasets, while for stronger baselines such as SCAFFOLD and FedDC the differences are smaller but mostly positive and always within one standard deviation.
> This confirms that client reshuffling is a generally beneficial and safe enhancement on top of existing FL methods.*
>
> *Second, DR-splitting–based methods benefit more from client reshuffling than traditional FedAvg-type schemes.
> On benchmark datasets, the improvement from FedDR to FedCDR is substantially larger than the improvement from FedDC to CR+FedDC, especially on the more heterogeneous and challenging settings (e.g., CIFAR10 and CIFAR100 with $\alpha = 0.1$, as well as the 10{,}000-client experiments).
> This indicates that DR splitting and client reshuffling are highly synergistic: reshuffling not only stabilizes DR splitting under non-IID data, but also unlocks significant accuracy gains.*
>
> *Third, without client reshuffling, FedDR is comparable to (and sometimes slightly worse than) FedDC across datasets.
> However, after combining DR splitting with client reshuffling, FedCDR consistently outperforms even the strongest baselines with client reshuffling, including CR+FedDC.
> On both synthetic and benchmark datasets (and in the 10{,}000-client regime), FedCDR achieves the highest top-1 accuracies in almost all settings, often by a non-trivial margin over CR+FedDC.
> These results corroborate our theoretical findings: while client reshuffling provides a general improvement for many FL methods, it is particularly effective when coupled with DR splitting, and FedCDR leverages this combination to achieve state-of-the-art performance under severe data heterogeneity and large-scale participation.*
>
> **Weakness 4**
> Due to the limited rebuttal time, we were unable to perform these scalable experiments, and we leave this investigation for future work.

---

> > ### Author Response · Authors · 2025-11-19
> >
> > **Question 1**
> > Regarding Question 1, we quantitatively analyze the additional computational cost introduced by the proximal steps in ***Appendix E***.
> >
> > *Results: The computational cost of proximal FL methods (FedProx, FedCDR) is approximately twice that of non-proximal FL methods (FedAvg, SCAFFOLD, FedDC), due to the increased number of local epochs required to solve the proximal subproblem. However, SCAFFOLD and FedDC incur higher communication overhead because of the additional exchange of correction variables. To obtain the total cost, we multiply the communication cost by the computation cost. In terms of total cost, SCAFFOLD, FedDC, FedProx, and FedCDR are nearly the same, making it fair to compare their results under this setting.*
> >
> > **Question 2**
> > Our theory relies on uniform sampling strategies to complete the proof. We leave the exploration of non-uniform sampling to future work.
> >
> > [1] Grigory Malinovsky, Samuel Horváth, Konstantin Burlachenko, Peter Richtárik:
> > Federated Learning with Regularized Client Participation. CoRR abs/2302.03662 (2023).

---

### Meta-Review · Area_Chair_mbFY · 2026-01-07

**Summary:**

I will focus here on reviewer comments that remained unresolved and informed my recommendation.

1) The lack of dependence of convergence rate to heterogeneity (ShC1) needs to be elaborated on, in light of how this is so prominent in other methods.

2) Quantitative communication-computation tradeoffs should also be addressed (ShC1)

3) Heterogeneous local computation budgets (ekkD) would appear in practice.

4) Several reviewers asked for an indication of how bounds would look like under incrementally different modeling assumptions. E.g., ekkD asked for a comparison to standard algorithms with client reshuffling, and ShC1 asked for the same with random client sampling.

5) There are a lot of hyperparameters involved (HbWk), limiting the practical application of the proposed scheme.

6) Novelty appears to be limited, main techniques already exist (HbWk).

**Reviewer Concerns:**

1) The authors claim that heterogeneity is somewhat hidden in the performance of the proximal operator. This needs to be further elaborated on, indicating how final bounds compared to other bounds under heterogeneity, making this an apples-to-apples comparison.

2) The quantitative results on communication-computation tradeoffs in Appendix F should be expanded.

3) Heterogeneity in computation should be explored beyond the effect of stragglers.

4) The requested comparisons would help highlight how different modeling assumptions components affect the system.

5) $\eta# still seems hard to compute in practice.

6) The authors provide an explanation as to why combining DR splitting and client reshuffling is not trivial: makes experiments of the type requested in 4 all the more important.

**Reviewer Scores:**

I do not think any of the reviewers would raise their scores.

---

### Decision · Program_Chairs · 2026-01-26

Reject